# Multi-omic profiling of clear cell renal cell carcinoma identifies metabolic reprogramming associated with disease progression

Junyi Hu [1,14], Shao-Gang Wang [1,14], Yaxin Hou [1,14], Zhaohui Chen [2,14], Lilong Liu[1], Ruizhi Li[3], Nisha Li[3,4], Lijie Zhou[5], Yu Yang[6], Liping Wang[7], Liang Wang[2], Xiong Yang[2], Yichen Lei[8], Changqi Deng[2], Yang Li [1], Zhiyao Deng[1], Yuhong Ding[1], Yingchun Kuang[1], Zhipeng Yao[1], Yang Xun[1], Fan Li[1], Heng Li[1], Jia Hu[1], Zheng Liu[1], Tao Wang[1], Yi Hao[9], Xuanmao Jiao[10], Wei Guan [1]✉, Zhen Tao [11,12]✉, Shancheng Ren [13]✉ & Ke Chen [1]✉

Clear cell renal cell carcinoma (ccRCC) is a complex disease with remarkable immune and metabolic heterogeneity. Here we perform genomic, transcriptomic, proteomic, metabolomic and spatial transcriptomic and metabolomic analyses on 100 patients with ccRCC from the Tongji Hospital RCC (TJ-RCC) cohort. Our analysis identifies four ccRCC subtypes including De-clear cell differentiated (DCCD)-ccRCC, a subtype with distinctive metabolic features. DCCD cancer cells are characterized by fewer lipid droplets, reduced metabolic activity, enhanced nutrient uptake capability and a high proliferation rate, leading to poor prognosis. Using single-cell and spatial trajectory analysis, we demonstrate that DCCD is a common mode of ccRCC progression. Even among stage I patients, DCCD is associated with worse outcomes and higher recurrence rate, suggesting that it cannot be cured by nephrectomy alone. Our study also suggests a treatment strategy based on subtype-specific immune cell infiltration that could guide the clinical management of ccRCC.

Renal cell carcinoma (RCC), one of the top ten most prevalent malignancies worldwide, predominantly manifests as clear cell RCC (ccRCC)[1–3]. Metabolic deregulation is a key characteristic of ccRCC[4]. To date, most clinical metabolomic studies have focused solely on metabolomic profiling[5,6], leaving the relationship between genomic, epigenomic or other alterations and metabolic disorders largely unexplored. Recent single-cell and spatial sequencing technologies have provided a more intuitive map of the tumor microenvironment (TME) within ccRCC[7–9]. However, due to limited sample sizes, correlating these single-cell datasets with existing genomic or epigenomic molecular subtypes has been challenging. Therefore, a comprehensive dataset encompassing multi-omic data could bridge previous findings.

## Results

### Multi-omics characterization of Tongji Hospital RCC (TJ-RCC) cohort

In total, 100 treatment-naive ccRCC samples and 50 paired normal adjacent tissues (NATs) were profiled using whole-exon sequencing

(WES), whole-transcriptome sequencing (WTS), proteome and untargeted metabolomics (both liquid chromatography–mass spectrometry (LC–MS) and gas chromatography–mass spectrometry (GC–MS)) to identify the general molecular characteristics of the TJ-RCC cohort (Fig. 1a). Patient characteristics are summarized in Supplementary Table 1.

## Genomic landscape

Previous studies identified *VHL*, *PBRM1*, *BAP1*, *SETD2* and *KDM5C* as the most frequently mutated genes in ccRCC. Moreover, a mutation signature associated with aristolochic acid (AA) exposure, characterized by T>A transversions, has been identified in Romanian and Chinese cohorts[10–12]. The most frequently mutated genes in this study align with previous studies (Fig. 1b and Extended Data Fig. 1a)[11–13]. Co-occurrence analysis identified the following two mutually exclusive somatic mutation pairs: *PBRM1-BAP1* and *BAP1-KDM5C* (Extended Data Fig. 1b). Non-negative matrix factorization (NMF) analysis on a meta-cohort[11,12] identified seven somatic mutational signatures. All of these signatures were found in the TJ-RCC cohorts. Overall, 49 tumors were confirmed to have the AA-associated mutation signature (SBS22; Extended Data Fig. 1c–e). These tumors had more somatic mutations (Extended Data Fig. 1f), in agreement with previous studies[10–12]. However, frequently mutated genes did not show a significantly higher mutation rate in the AA-associated group (Fig. 1c). Consistently, the AA-associated signature was largely associated with mutations in genes excluding *VHL*, *PBRM1*, *SETD2*, *BAP1* and *KDM5C* in ccRCC (Extended Data Figs. 1g,h). As major ccRCC driver mutations are not apparently associated with AA-dependent mutagenesis, our data suggest that the role of AA in ccRCC warrants reassessment.

Chromosome 3p loss was the most frequent event, followed by chromosome 5q gain, 14q loss, 7 gain and 9 loss, consistent with other ccRCC cohorts[11–14] (Extended Data Fig. 1i).

## Transcriptomic, metabolomic and proteomic analysis

Both transcriptomic and proteomic data correlated closely with somatic copy number alteration (SCNA) status (Extended Data Fig. 1j). Principal component analysis (PCA) demonstrated significant discrimination between tumors and NATs (Fig. 1d). Pathway analysis indicated activation of angiogenesis, glycolysis and immune-related pathways in ccRCC tissues, alongside inhibition of several metabolic pathways (Fig. 1e,f and Supplementary Table 2). The metabolome analysis results correlated with those from our transcriptomic and proteomic analyses (Fig. 1g), with exceptions such as glycerolipid metabolism and glutathione (GSH) metabolism pathway.

To provide an overview of the metabolic dysregulation in ccRCC, we generated an integrated map of metabolites and metabolic enzymes (Supplementary Fig. 1a). We found that the expression of enzymes involved in fatty acid biosynthesis was not changed in tumor tissues, which indicates that the accumulation of lipid droplets (LDs), a hallmark of ccRCC[15], was potentially caused by decreased β-oxidation activity.

## ccRCC can be classified into four immune subtypes

An immune subtype classifier was previously defined in ref. 13 to predict the prognosis of patients with ccRCC. This study and others used the signature matrices offered by xCell[16] or CIBERSORTx[17] to deconvolute the cell composition of tumor tissues and did not consider tissue-specific expression patterns, especially signatures from macrophages[18,19] and tumor cells. To overcome this, we generated a ccRCC-specific signature matrix from a public scRNA-seq dataset[20] and identified 25 different cell types. This matrix was used to divide tumor samples into four immune subtypes (IM1–IM4; Extended Data Figs. 2a–c). We also performed xCell analysis and found that the two deconvolution pipelines generally produced similar output (Fig. 2a and Extended Data Figs. 2d,e).

IM1 was characterized by enriched endothelial (Fig. 2a) and stromal cell signatures with a lack of immune cell signatures suggestive of immune exclusion. IM2 tumors also had enriched endothelial signatures and were depleted of immune and stromal cell signatures. Among the four subgroups, IM3 tumors had the lowest level of endothelial and stromal cell signatures but showed increased T cells and tumor-associated macrophages (TAMs) scores. IM4 tumors exhibited the highest level of stromal and TAM scores, together with intermediate T cell scores. Pathway analysis of differentially expressed genes (DEGs) revealed that acute inflammatory response and complement cascade had the highest activity in IM4 tumors (Extended Data Fig. 2f). No correlation was observed between IM subgroups and the AA-associated mutational signature (Extended Data Fig. 2g). Immunohistochemistry (IHC) staining on selected samples from the four subgroups validated these TME features (Extended Data Fig. 2h).

Next, we investigated whether genomic and clinical characteristics correlated with immune subtypes. As shown in Fig. 2b, IM4 harbored the most arm-level SCNAs, followed by IM3, while somatic mutation burden (that is, single-nucleotide variants) showed no difference across four groups (Fig. 2b,c). Subgroups with more SCNAs were associated with higher tumor grade and more advanced-stage disease (Fig. 2d). These results were validated in the Cancer Genome Atlas kidney renal clear cell carcinoma (TCGA KIRC) cohort (Extended Data Fig. 3a–c).

## Correlations between the IM subtypes and survival

Kaplan–Meier (KM) analysis revealed that IM2 patients had the best overall survival (OS) and progression-free survival (PFS), while IM4 tumors were associated with poorer outcomes. Surprisingly, although IM3 contained more advanced-stage tumors, the prognosis of the IM3 group was comparable to that of the IM1 group. Similarly, although IM3 and IM4 groups in TCGA had comparable numbers of advanced-stage tumors, IM3 tumors were associated with better prognosis (Extended Data Fig. 3d).

## Immune cell populations underpinning IM1–IM4

To further investigate the cell composition heterogeneity underlying immune subtypes, we performed single-nucleic RNA-seq (snRNA-seq) using 10 samples (4 IM1, 2 IM2, 2 IM3 and 2 IM4) and 10×-multiome (both snRNA-seq and single-nucleus ATAC sequencing (snATAC-seq)) on another 10 samples (1 IM1, 3 IM2, 4 IM3 and 2 IM4). After quality control, 97,978 single nuclei were classified into 5 major cell types and 41 subclusters (Fig. 2e,f, Extended Data Fig. 3e and Supplementary Fig. 2). The distribution of immune cell clusters varied across immune subtypes, whereas endothelial cell subpopulations exhibited less variability (Fig. 2g and Extended Data Fig. 3f). Specifically, monocytes, *LILRB5*+ macrophages (MØ03–LILRB5), terminally differentiated effector memory or effector cells (T_EMRA) and activated NK (aNK) cells predominated in IM1 and IM2 tumors. We also observed increased

---

**Fig. 1 | Multi-omics characterization of RCC cohort of TJ-RCC cohort.**
**a**, Study design and workflow of TJ-RCC. Sample selection for single nucleic and spatial profiling was based on the result of molecular subtyping. **b**, Genomic profiles of 100 TJ-RCC tumors. The entries in each column show the genomic profiles of the 100 TJ-RCC tumors. **c**, Mutation rate of frequently mutated genes in AA (*n* = 49) and non-AA (*n* = 441) groups. Mutation types are colored the same as those in **b**. *P* values were calculated by one-sided Fisher's exact test and adjusted by the BH algorithm. **d**, PCA of global transcriptomic, proteomic and metabolomic difference between 100 tumors and 50 NATs. Shaded areas denote different groups. **e,f**, Bar plot showing ssGSEA-based pathway differences between 100 tumors and 50 NATs in transcriptome (**e**) and proteome (**f**). *T* values were calculated by a linear algorithm in Limma. **g**, Pathway-level differences in metabolite abundance between 100 tumors and 50 NATs. Metabolite pathways are collected in the KEGG pathway database. Size of each dot represents the total number of metabolites in the pathway.

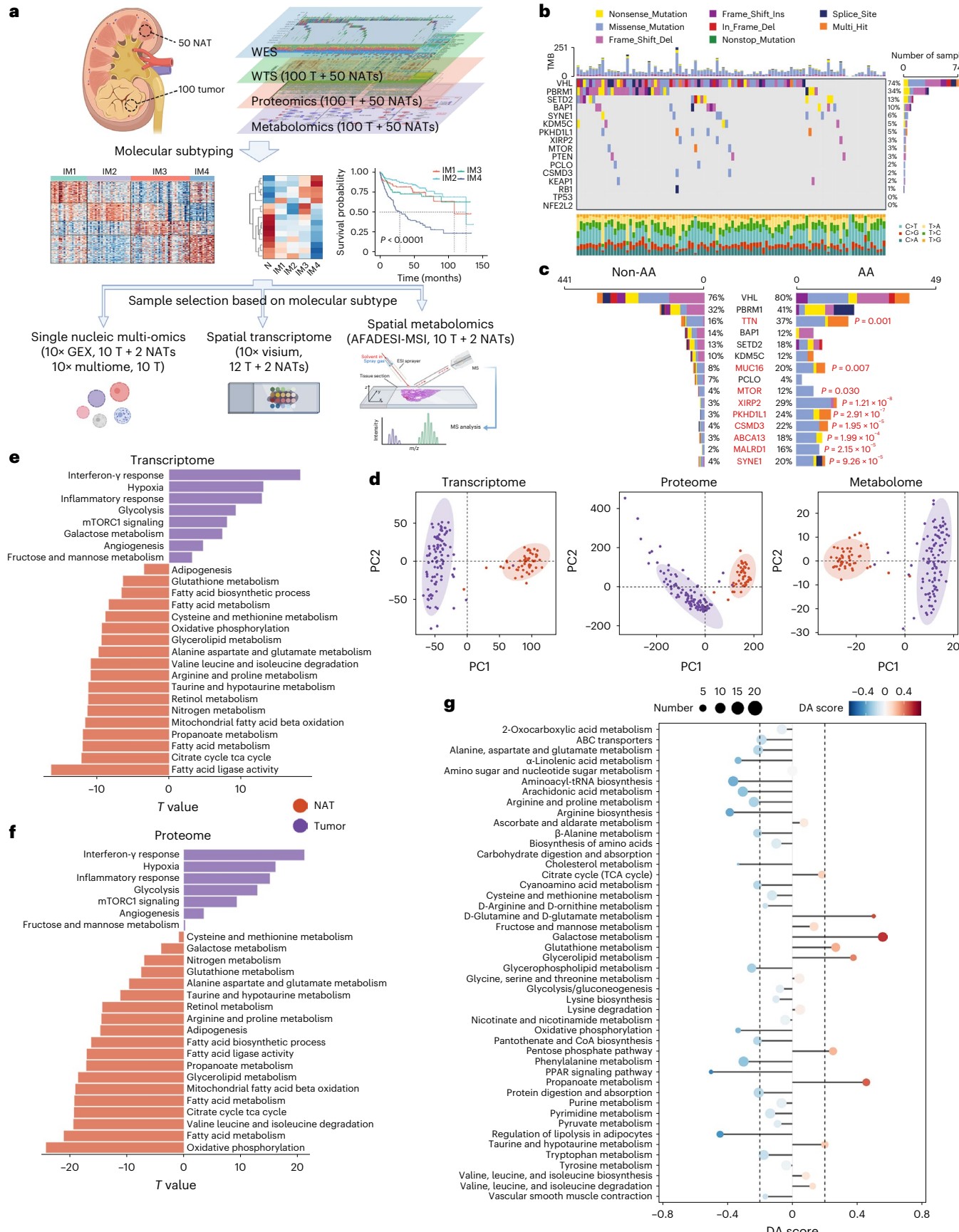

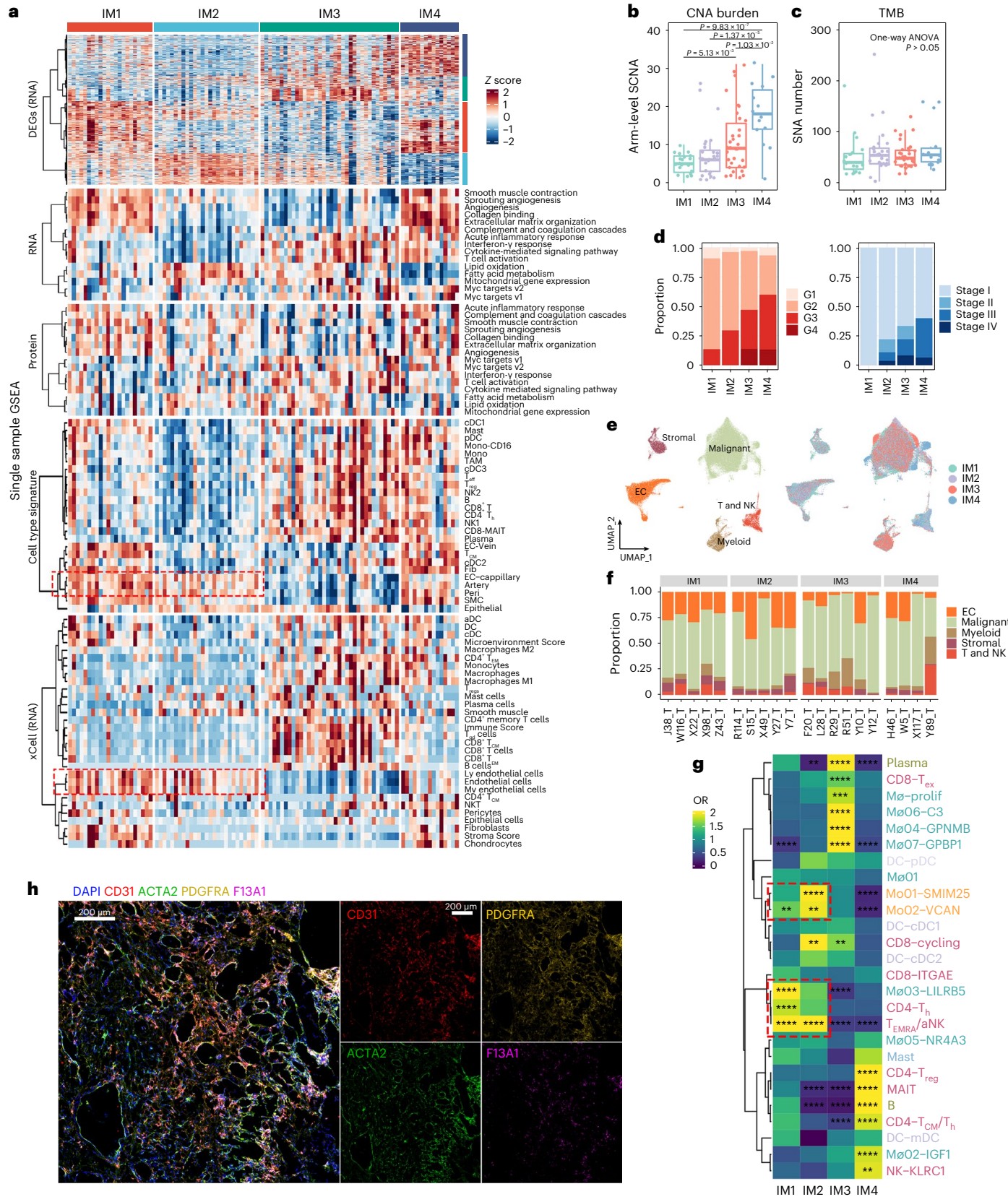

**h** DAPI CD31 ACTA2 PDGFRA F13A1

expression levels of *FOS*, *FOSB* and *JUND* in the endothelial cells of IM1 and IM2 tumors (Extended Data Fig. 3g), and these genes can activate endothelial cell proliferation[21]. snATAC-seq data confirmed that *FOS* and *FOSB* had higher chromatin accessibility in endothelial cells from IM1 and IM2 samples (Extended Data Fig. 3h).

Additionally, IM3 tumors had increased CD8[+] T cell infiltration. The proportion of terminal exhausted T cells was also higher in IM3 tumors, followed by IM4 tumors, which was confirmed by bulk-seq data (Extended Data Fig. 4a,b). Substantial differences were observed in TAMs of IM3 and IM4 tumors. Proliferating macrophages (Mø−prolif),

**Fig. 2 | TME-based molecular subtyping of ccRCC. a,** Integrative classification of 100 tumor samples. The heatmap shows DEGs, mRNA and protein level pathway scores calculated by the ssGSEA algorithm, ssGSEA-predicted cell abundance and xCell-predicted cell abundance, sequentially. DEGs were defined as genes differentially expressed in one IM subtype versus all other IM subtypes. Each matrix of 100 tumor samples was transformed into row $z$ score before visualization, respectively. **b,c,** Arm-level SCNAs and nonsynonymous mutational burdens in immune subtypes. $P$ values were calculated by two-sided Tukey's test. IM1, $n = 22$; IM2, $n = 27$; IM3, $n = 36$ and IM4, $n = 15$. The central line of the box represents the median value, box limits indicate the IQRs and the whiskers extend to 1.5× IQR. **d,** Difference in tumor grade and stage among the four immune subtypes. **e,** UMAP plot of snRNA-seq data colored by major cell types (left) and immune subtypes (right). **f,** Bar plot of the proportions of major cell types per sample. **g,** Heatmap of OR showing differences in cell abundance between immune subtypes. $P$ value was calculated by one-sided Fisher's exact test and corrected by BH algorithm. OR > 1.5 and adjusted $P$ value ($P_{adj}$) < 0.05 indicated that this subpopulation was more likely to distribute in this immune subtype, whereas OR < 0.5 and adjusted $P$ value < 0.05 indicated that it was preferable not to distribute in this immune subtype. Innate immune subclusters enriched in IM1/IM2 are marked with red boxes. **h,** Multicolor fluorescence on W5_T. CD31 marks blood vessels, PDGFRA marks fibroblasts, ACTA2 marks smooth muscle and pericytes and F13A1 marks IGF1+ macrophages. The experiment was repeated three times. TMB, tumor mutation burden; aNK, activated natural killer; cDC, conventional DC; DC, dendritic cell; MAIT, mucosal-associated invariant T cells; mDC, mature DC; Mo, monocyte; Mø, macrophage; pDC, plasmacytoid DC; $T_H$, T helper cell; $T_{ex}$, exhausted T cell; $T_{gd}$, gamma delta T cell; IQR, interquartile range.

---

*GPNMB*+ macrophages (Mø04–GPNMB), *C3*+ macrophages (Mø06–C3) and *GBP1*+ macrophages (Mø07–GBP1) were also enriched in IM3 tumors (Fig. 2g). Pathway analysis revealed enhanced interferon-γ response and regulation of T cell cytotoxicity in these clusters (Extended Data Fig. 4c).

*IGF1*+ macrophages were more abundant in IM4 tumors. Pathway enrichment analysis demonstrated that the growth factor complex pathway was enriched in this subpopulation (Fig. 2g and Extended Data Fig. 4c). *IGF1* has been reported to promote the survival and migration of fibroblasts[22], potentially contributing to fibroblast accumulation in ccRCC (Extended Data Fig. 4d–f). Interestingly, the abundance of *IGF1*+ macrophages was positively correlated with the fibroblast score but not with other stromal cells, such as pericytes (Extended Data Fig. 4d,f,g). As shown in Extended Data Fig. 4h, there was an obvious spatial boundary between the fibroblast-enriched and pericyte/smooth muscle-enriched regions. Subsequent immunofluorescence staining revealed that *IGF1*+ macrophages (*F13A1*+) mainly localized in fibroblast (*PDGFRA*+)-enriched regions, supporting our hypothesis that *IGF1*+ macrophages promote the accumulation of fibroblasts (Fig. 2h and Extended Data Fig. 4i) in IM4 tumors. In contrast, although *ACKR1*+ endothelial and *GJA5*+ arterial endothelial also showed a positive correlation with fibroblasts in bulk RNA-seq, these cells were scattered on the immunofluorescence or IHC slices (Extended Data Fig. 4f,i–k).

### Correlating immune and metabolic heterogeneity in TME

To investigate the role of metabolism in immune subtype stratification, we evaluated metabolic-related Gene Ontology (GO) terms per sample. Unsupervised clustering revealed four different clusters of metabolic gene expression (Extended Data Fig. 5a). Cluster 1 involves collagen and proteoglycan metabolism, potentially associated with extracellular matrix remodeling. Cluster 2 is related to steroid hormone metabolism. The activity of both cluster 1 and cluster 2 mainly arose from interstitial and endothelial cells (Extended Data Fig. 5b). Both cluster 1 and 2 were enriched in IM1 and IM4 tumors. Cluster 3 is related to fatty acid and amino acid metabolism and was enriched in IM2 and IM3.

Cluster 4 is related mainly to nucleoside metabolism (Extended Data Fig. 5a). Nucleoside metabolism alone could distinguish IM3 samples from other subtypes (Fig. 3a and Extended Data Fig. 5c). Conjoint analysis revealed enhanced cytidine, pyrimidine and other nucleoside metabolism-related pathways in IM3 samples at both the transcriptome and proteome levels (Fig. 3b and Supplementary Table 3). Subsequently, correlation analysis between the single sample gene set enrichment analysis (ssGSEA) score of $T_{eff}$ cells and pyrimidine metabolism in the TCGA KIRC dataset suggested that increased pyrimidine/cytidine metabolism activity was correlated with increased CD8+ T cell infiltration and poor prognosis, aligning with our findings in the TJ-RCC cohort (Fig. 3c,d). Additionally, increased pyrimidine derivates in IM3 samples were observed in the metabolome data (Fig. 3e).

However, although IM3 is a CD8+ T cell-infiltrated subgroup, IM3 tumors still have a better prognosis than IM4 tumors. These results suggest that CD8+ T cell infiltration may limit tumor progression in ccRCC.

To substantiate our findings, we performed spatial transcriptomics (ST) and metabolomics profiling on 12 tumor sections (4 IM1, 4 IM2, 3 IM3 and 1 IM4) and 2 NAT controls. High guanine and hypoxanthine signals were detected in a CD8+ T cell-infiltrated sample (R29_T) that belongs to the IM3 group (Fig. 3f,g and Extended Data Fig. 5d). In contrast, in R51_T, also an IM3 sample with focal lymphocytic infiltration (Extended Data Fig. 5e), only weak signals of guanine and hypoxanthine were detected in the non-TIL-infiltrated area (Fig. 3f,g). These findings indicated the intratumoral heterogeneity (ITH) of ccRCC and a potential correlation between pyrimidine derivates and TILs. This hypothesis was further confirmed in another sample with focal CD8+ T cell infiltration (Fig. 3h–l).

### IM4 tumors show distinctive metabolic features and have poor outcomes

Cluster 3, which is downregulated in the IM4 subgroup (Extended Data Fig. 5a), includes various metabolic pathways, involving fatty acid and amino acid metabolism. These pathways are typically downregulated in ccRCC[13,14,23], suggesting enhanced metabolic deregulation in IM4

---

**Fig. 3 | Correlation between immune and metabolic heterogeneity in the TME. a,** Heatmap of ssGSEA-based pyrimidine-related pathway scores. Pathway scores were calculated using a TPM matrix of bulk RNA-seq data from 100 tumor samples. **b,** Visualization of differential GO terms (ssGSEA) between IM3 tumors and other tumors of TJ-RCC. $T$ values were calculated by a linear model of Limma and extracted to visualize the relationship between transcriptome and proteome. Each dot represents a single pathway. GO terms associated with cytidine, pyrimidine or nucleoside are labeled in specific colors. **c,** KM plot of survival data stratified by higher or lower pyrimidine metabolism activity in TCGA KIRC. Pathway scores of bulk RNA-seq data were estimated per sample by ssGSEA algorithm. $P$ value was calculated by the log-rank algorithm. **d,** Pyrimidine and cytidine metabolism activity are positively associated to $T_{eff}$ cell abundance. Pyrimidine and cytidine metabolism pathway scores were calculated in the TPM matrix of TCGA KIRC using the ssGSEA algorithm. $T_{eff}$ abundance was obtained from the results of xCell. Correlation coefficient was calculated using Pearson's correlation algorithm. $P$ values were from two-sided Student's $t$ test. The red line shows the linear interpolation of the data. **e,** Heatmap showing enrichment of given metabolites in IM3. $P$ values were calculated by comparing all expression values of each metabolite in a subgroup to all other values of that metabolite using two-sided Wilcoxon $t$ test and then adjusted using the BH algorithm. ****$P_{adj}$ < 0.0001, ***$P_{adj}$ < 0.001, **$P_{adj}$ < 0.01, *$P_{adj}$ < 0.05. **f,** Expression of *CD8A* in ST data of given slices. **g,j,** Spatial metabolome slice colored by metabolites abundance. **h,** Immunohistochemical staining of CD8A and CD3D on Y7_T. Experiment was repeated three times. **i,k,** Spatial transcriptome of Y7_T colored by *CD8A* expression level (**i**) and Seurat clusters (**k**). **l,** Pathway enrichment of spatial transcriptome shows that clusters 3 and 4 have higher pyrimidine metabolism activity. Pathway score was calculated by the ssGSEA algorithm. The density plot is colored by clusters on the ST profile in **k**.

samples. To systemically investigate these differences, we identified DEGs between IM4 and other tumor subgroups (|log₂ fold change (FC)| >1, adjusted $P$ < 0.05). In total, 853 DEGs were clustered into four gene modules using unsupervised analysis (Fig. 4a).

The expression of genes in modules 1 and 2 was higher in IM4 tumors than in other tumor samples belonging to IM1–IM3.

Notably, NATs had the highest expression level of module 1 genes. Analysis of snRNA-seq data of NATs revealed that module 1 genes were expressed in tubular cells except for proximal tubular cells, and module 2 signatures were derived mainly from collecting ducts (Extended Data Fig. 6a–d), elucidating the recent finding that a collecting duct signature was related to poor prognosis[7].

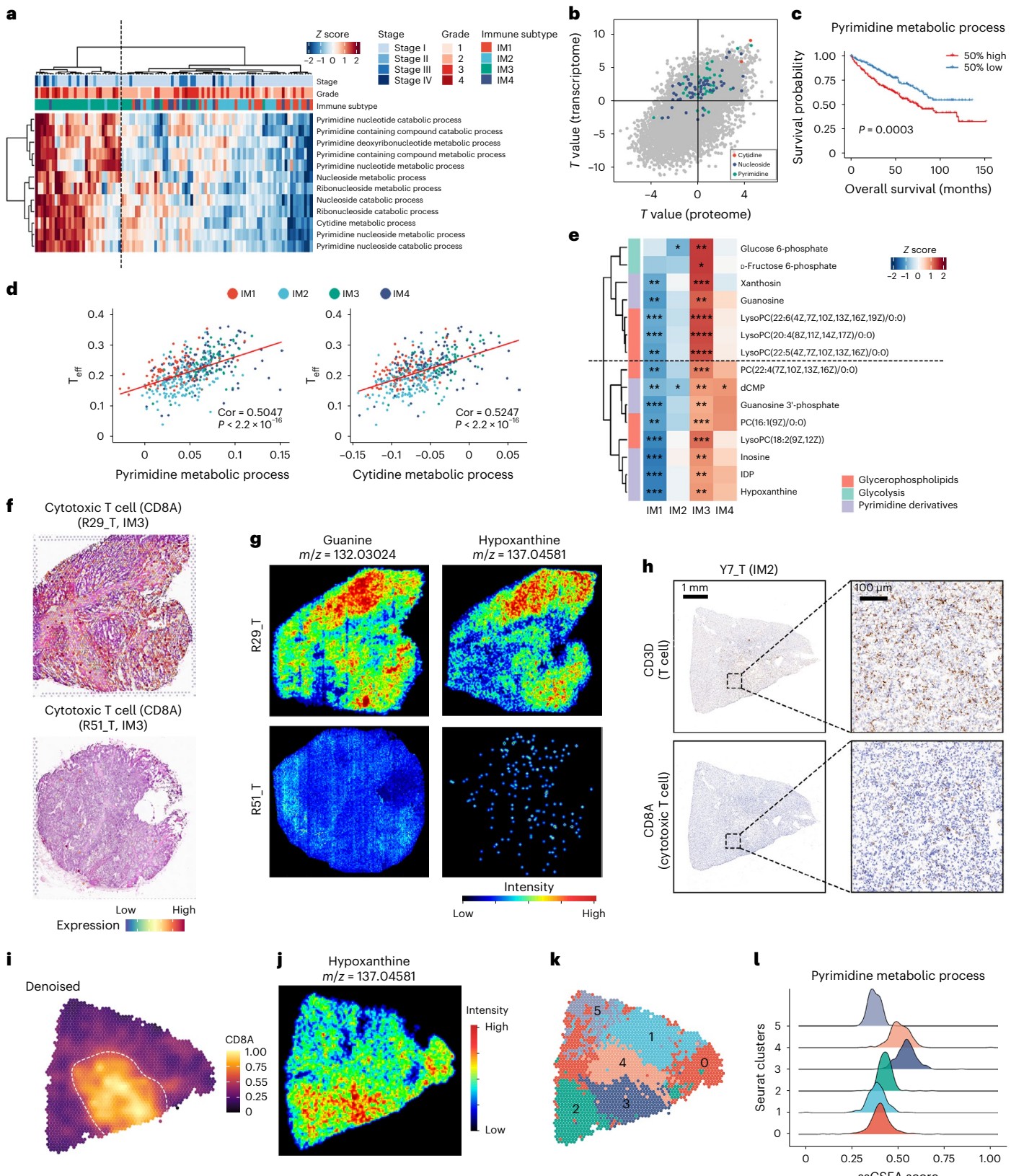

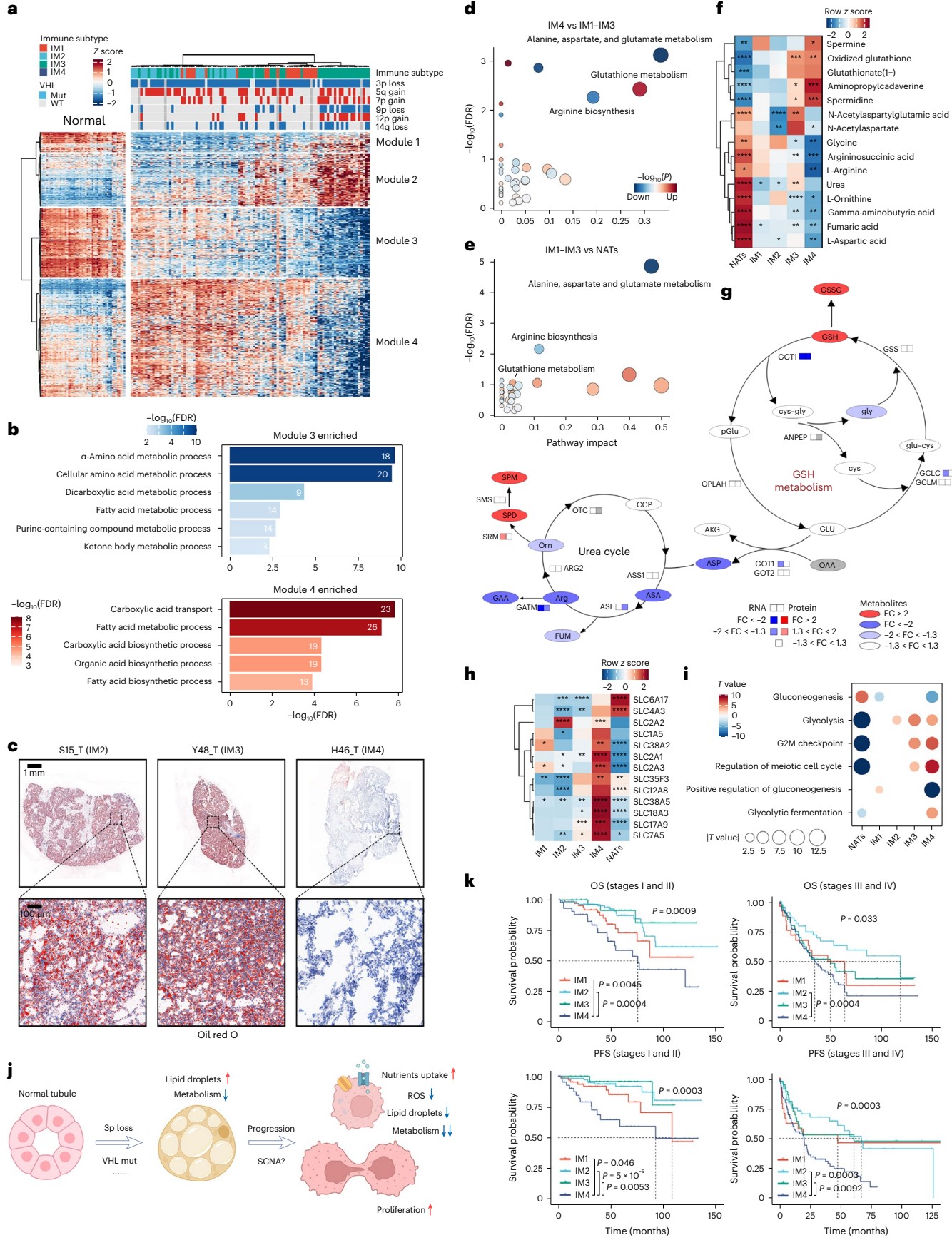

**Fig. 4 | The DCCD subtype exhibits dramatic metabolic disorders and worse outcomes in ccRCC. a**, Unsupervised clustering of upregulated and downregulated genes in DCCD tumors. $Z$ scores were calculated based on the mean value and s.d. of the TPM matrix across 100 tumors and 50 NATs. **b**, Enriched GO terms in module 3 and module 4 genes. Gene number involved in each GO term is labeled in each bar. **c**, ORO staining on given tumor samples to visualize LDs. **d,e**, Enrichment analysis of differential metabolites in IM4 versus IM1–IM3 (**d**) and IM1–IM3 versus NATs (**e**). The $x$ axis represented the pathway impact calculated by MetaboAnalyst. Dot size represents the pathway impact. $P$ values from the global test were adjusted by the BH algorithm. **f**, Metabolites enriched or absent in IM4. $P$ values were calculated by comparing the selected subgroup to all the other samples using two-sided Wilcoxon $t$ test and then adjusted using the BH algorithm. ****$P_{adj} < 0.0001$, ***$P_{adj} < 0.001$, **$P_{adj} < 0.01$, *$P_{adj} < 0.05$. **g**, Summary of key metabolic changes in IM4 versus IM1–IM3. Abbreviations of metabolites are detailed in Supplementary Table 4. **h**, Upregulated transporters at mRNA level in IM4. $P$ values were calculated by comparing all expression values of each gene in a subgroup to all other values of that gene with two-sided Wilcoxon $t$ test and then adjusted using the BH algorithm. ****$P_{adj} < 0.0001$, ***$P_{adj} < 0.001$, **$P_{adj} < 0.01$, *$P_{adj} < 0.05$. **i**, Pathway enrichment analysis of bulk transcriptome using the ssGSEA algorithm. $T$ value was calculated with the Limma package by comparing one subtype to all other samples. A negative $T$ value implies a downregulated pathway score. The dot size is defined by the absolute value of $T$ value. **j**, Ideogram of key changes that participate in carcinogenesis and tumor progression. **k**, KM in early (left) and advanced (right) ccRCC, grouped by immune subtypes. The global $P$ value was calculated by the log-rank algorithm. Pairwise comparison was performed when the global $P$ value < 0.05. Square brackets show the results of pairwise comparison.

Module 3 genes, prominently expressed in normal proximal tubules and downregulated in non-IM4 tumors, were further decreased in IM4 tumors (Fig. 4a). These genes are enriched in the metabolism of fatty acids, amino acids and carbohydrates (Fig. 4b). Decreased activity of these pathways has been described as a feature of ccRCC[13,14,23]. In contrast to the genes in module 3, genes in module 4 were upregulated in non-IM4 tumors but downregulated in IM4 tumors to below NAT levels (Fig. 4a). The ccRCC oncogene, *HIF2A*, is also included in this module. These genes are enriched in multiple biosynthesis pathways, including the fatty acid biosynthesis pathway (Fig. 4b), leading to LD accumulation in tumor cells[15]. Notably, module 4-related metabolic dysregulations were described as a feature of ccRCC and also correlated with tumor progression[4,24]. Because these features are largely inhibited in IM4 tumors, module 4-related features may have dual effects on ccRCC. Validation via Oil red O (ORO) staining of 20 randomly selected TJ-RCC cohort samples revealed LD accumulation in non-IM4 samples exhibiting a 'clear cell' phenotype, but almost no LDs were observed in IM4 samples, aligning with transcriptomic features (Fig. 4c).

Metabolic reprogramming in IM4 tumors was also confirmed by metabolomic analyses. Levels of metabolites related to alanine, aspartate and glutamate metabolism and arginine biosynthesis were lower in IM1–IM3 tumors than in NATs. These biosynthesis pathways were also downregulated (albeit to a lesser extent) in the IM4 group, corresponding to our transcriptomic findings (Fig. 4d,e). Limited arginine biosynthesis indicated a limited urea cycle in IM4 tumors (Fig. 4f,g). Notably, urea cycle dysfunction in IM4 tumors did not seem to depend on enzyme dysregulation, suggesting unknown mechanisms (Fig. 4g).

In contrast, GSH metabolism was activated in IM4. Both GSH and oxidized GSH (GSSG) are highly enriched in IM4 tumors. These molecules could consume reactive oxygen species (ROS) in tumor cells, leading to a higher proliferation rate and drug resistance[25]. Unlike GSH-high triple-negative breast cancer[26], IM4 tumors do not show elevated levels of substrates of GSH biosynthesis (Fig. 4f,g and Extended Data Fig. 6e). In addition, the majority of enzymes involved in GSH biosynthesis were not dysregulated in IM4 tumors. Only *GGT1*, the enzyme that catalyzes the transfer of the glutamyl moiety of GSH, was downregulated (Fig. 4g). Although the levels of ornithine, which make up part of the urea cycle, were decreased in IM4 tumors, levels of two ornithine derivates, spermine and spermidine, were still largely increased (Fig. 4f,g); both have been associated with cancer cell proliferation[27,28].

Given the downregulation of metabolic energy generation pathways and accelerated proliferation rate, IM4 tumors likely have alternative energy sources. The expression of genes encoding glucose transporters (*SLC2A1* and *SLC2A3*), glutamine transporters (*SLC1A5* and *SLC38A5*), a branched-chain amino acid transporter (*SLC7A5*) and a thiamine transporter (*SLC35F3*) was upregulated (Fig. 4h), potentially leading to increased nutrient uptake in IM4 tumors. The level of a new nicotinamide mononucleotide transporter, *SLC12A8* (refs. 29,30), was also increased in IM4 tumors (Fig. 4h). Although IM4 tumors may uptake additional glucose, only the glycolytic fermentation process was enhanced, while gluconeogenesis was inhibited in this group (Fig. 4i).

In summary, the characteristics of IM4 tumors include increased nutrient uptake, decreased levels of ROS and LDs, low metabolic activity and a higher proliferation rate (Fig. 4j). Because ccRCC is characterized by LD accumulation in cancer cells, we named this process de-clear cell differentiation (DCCD) of ccRCC. Furthermore, as IM4 tumors have completed this transformation process, we termed them DCCD-ccRCC.

Because patients with primary tumors diagnosed at early stages (stages I and II) do not receive postoperative drug treatment[31], we asked whether patients with early-stage DCCD tumors had poorer prognosis. Strikingly, among patients with ccRCC restricted to the kidney (stages I and II), those with primary tumors stratified as DCCD tumors have significantly worse OS and PFS (Fig. 4k). This indicated that surgery alone is unlikely to cure these patients. Our data suggest that patients with localized DCCD-ccRCC should be identified and offered further treatment after surgery.

## IM subtype is associated with treatment response

To identify potential treatment plans for patients with DCCD (IM4) and non-DCCD (IM1–IM3) ccRCC, we analyzed bulk RNA-seq data from three clinical trials. Our aim was to establish whether these subgroups were associated with response to specific treatments.

In the IMmotion 151 trial[32], the combination of atezolizumab (programmed cell death 1 ligand 1 (PD-L1) inhibitor) and bevacizumab (vascular endothelial growth factor (VEGF) inhibitor) improved the prognosis of IM3 and DCCD (IM4) groups compared to sunitinib (receptor tyrosine kinase inhibitor (TKI); Fig. 5a–c). We could not clarify whether patients benefited from either atezolizumab, bevacizumab or both.

**Fig. 5 | IM subtype is associated with treatment response. a,d,g**, Heatmap showing expression level of signature genes in IMmoton 151 (**a**), JAVELIN (**d**) and CheckMate (**g**). Expression were normalized into row $z$ score based on the mean value and s.d. across four IM subgroups. **b,e,h**, KM plot of immune subtypes in given cohorts IMmotion 151 (**b**), JAVELIN (**e**) and CheckMate (**h**). The global $P$ value was calculated by the log-rank algorithm. Pairwise comparison was performed when the global $P$ value < 0.05. $P$ values of pairwise comparison were corrected by the BH algorithm. ****$P_{adj} < 0.0001$, ***$P_{adj} < 0.001$, **$P_{adj} < 0.01$, *$P_{adj} < 0.05$. atezo_bev, atezolizumab plus bevacizumab; combo, avelumab plus axitinib. **c,f**, Forest plot for PFS hazard ratios in patients treated with combined therapy versus sunitinib; IMmoton 151 (**c**) and JAVELIN (**f**). The numbers next to the forest plot indicate the number of participants in each group. Centers for the error bars represent the hazard ratio and error bands represent 95% confidence intervals (CIs) of hazard ratios. **i**, Forest plot for OS and PFS hazard ratios in patients treated with nivolumab versus everolimus (left, OS and right, PFS). The numbers next to the forest plot indicated the number of participants in each group. Centers for the error bars represent the hazard ratio and error bands represent 95% CIs of hazard ratios.

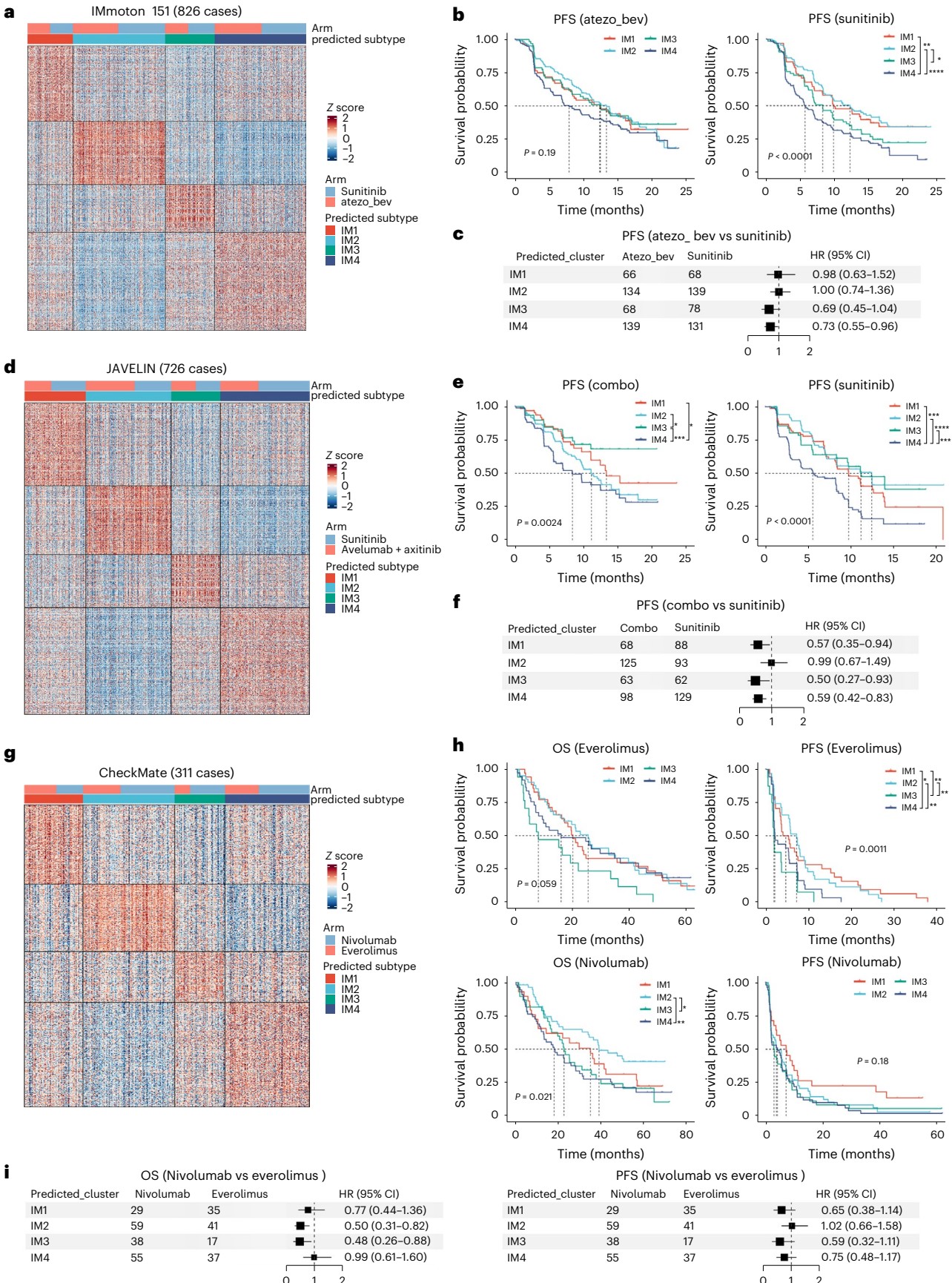

In another phase III clinical trial, JAVELIN[33], DCCD patients exhibited the poorest prognosis, independent of therapy type, which replicates the observation from IMmotion 151 (Fig. 5d,e). Compared to sunitinib alone, combination therapy with avelumab (PD-L1 inhibitor) and axitinib (VEGFR inhibitor) improved PFS in IM1, IM3 and IM4 groups (Fig. 5f). The IM1 group, which was characterized by high expression of angiogenesis-related genes, could therefore benefit from avelumab plus axitinib treatment, but not atezolizumab and bevacizumab (Fig. 5c,f). Notably, both IMmotion 151 and JAVELIN revealed that the combination of anti-angiogenic therapies with immune checkpoint blockade did not benefit IM2 patients.

In a meta-checkmate cohort with combined RNA-seq data of CheckMate 009, 010 and 025 (ref. 34), patient stratification impacted OS but not PFS in nivolumab-treated metastatic RCC (mRCC) patients (Fig. 5g,h). Notably, in the everolimus (mechanistic target of rapamycin kinase (mTOR) inhibitor)-treated group, patients with IM3 tumors had the shortest OS and PFS. Comparatively, treatment with nivolumab (PD-1 inhibitor) extended OS times versus everolimus (mTOR inhibitor) in IM2 and IM3 patients (Fig. 5i). Compared to everolimus, nivolumab may be a better choice for second-line treatment for IM2 and IM3 patients.

### Subcluster level shift from non-DCCD to DCCD in ccRCC

ssGSEA analysis revealed intermediate IM2 and IM4 scores in IM1 and IM3 samples (Extended Data Fig. 7a,b), which was replicated in TCGA KIRC samples, where signatures were inversely correlated across IM1 and IM3 samples (Pearson's $r = -0.725$; Fig. 6a), supporting the hypothesis of continuous DCCD in ccRCC. Therefore, we named these samples IM2-like or IM4-like according to the dominant gene signature and defined the DCCD score as the $D$ value of IM4 and IM2 scores (Fig. 6a,b). KM analysis demonstrated that patients with IM4-like tumors had better OS and PFS than those with IM4 but poorer than those with IM2 and IM2-like tumors, which had similar outcomes (Fig. 6c and Extended Data Fig. 7c). Despite significant TME differences between IM1 and IM3 tumors (Fig. 2a and Extended Data Fig. 3a), their outcomes were comparable (Extended Data Fig. 3d). Given that IM2 and IM2-like groups primarily contained low-grade, early-stage tumors (Extended Data Fig. 7d), we analyzed stage I cases from TCGA KIRC. Even among these early-stage cases, a higher IM4 score was associated with poor prognosis (Extended Data Fig. 7e), suggesting that the degree of DCCD progression might determine outcomes for patients with ccRCC.

We also extended this adjusted classification strategy to clinical trial datasets. In patients treated with TKI alone, this classification showed similar but relatively weaker predictive ability than that in TCGA (in both IMmotion[32] and JAVELIN[33] cohorts), possibly because these cohorts included only patients with metastatic RCC (Extended Data Fig. 8a,c), and a similar phenomenon was observed in advanced TCGA cases (Extended Data Fig. 7e). However, in immune checkpoint blockade-treated groups, it failed to improve predictive value over the initial IM classification (Extended Data Fig. 8a–g), possibly because the IM classification system considered signatures from the entire microenvironment while adjusted classification focused on features of cancer cells.

Subsequent analysis of snRNA-seq data revealed the same inverse correlation between IM2 and IM4 scores observed in the bulk-seq data (Fig. 6d,e and Extended Data Fig. 9a–d). Therefore, we divided single cancer cells into IM2-like or IM4-like phenotypes based on the DCCD score. Interestingly, the proportion of IM2-like or IM4-like cells correlated with the DCCD score at the bulk-seq level (Fig. 6f and Extended Data Fig. 9e), indicating that the DCCD process reflects the accumulation of IM4-like cancer cells inside tumors.

Because DCCD-ccRCC commonly contains more arm-level SCNAs than IM1–IM3 tumors (Fig. 4a), we asked whether SCNAs drive the DCCD process. In both the TJ-RCC and TCGA KIRC cohorts, the number of SCNA events exhibited the same trend—IM4>IM4-like>IM2-like≈IM2. Progression-related SCNA events occurred in up to 75% of IM4 tumors (Extended Data Fig. 10a,b). In snRNA-seq data, subclonal SCNAs could be found only in two tumors with part of malignant cells exhibiting DCCD features (partial DCCD; Fig. 6g and Extended Data Fig. 10c). Among 12 ccRCC samples with spatial profiles, partial DCCD was found in two samples (Fig. 6h,i and Extended Data Fig. 10d,e). Notably, Y7_T, exhibiting IM2 features in both bulk-seq and snRNA-seq data (Fig. 6f), showed a subclonal DCCD shift in the visium slice, reflecting DCCD-induced ITH. The non-DCCD region exhibited a classical 'clear cell' phenotype (Fig. 6h and Extended Data Fig. 10d), corresponding to that observed via ORO staining (Fig. 4c). Loss of chromosome 9 could be found in the DCCD region of X98_T, consistent with snRNA-seq data (Fig. 6g,j and Extended Data Fig. 10h). In contrast, no subclonal SCNAs could be observed in Y7_T (Extended Data Fig. 10f,g). Taken together, these results indicated that there was no absolute correlation between SCNA events and DCCD. Spatial metabolome analysis showed fewer fatty acids, especially long-chain fatty acids, in DCCD regions, regardless of the presence of subclonal SCNAs (Fig. 6k,l and Extended Data Fig. 10i,j), leading to decreased LD accumulation in these regions.

### Establishing the trajectory toward DCCD

To gain deeper insight into the transformation process toward DCCD, we constructed a trajectory using snRNA-seq data. IM4-like single cells were centered at the end of the trajectory. An increase in the IM4 score and a decrease in the IM2 score were observed throughout the pseudotime (Fig. 7a,b). Subsequently, we extracted differentially expressed transcription factors at both the bulk and snRNA levels, resulting in 83 differentially expressed transcription factors grouped into two clusters (Fig. 7c). *HNF1A*, *HNF1B*, necessary for renal tubule development[35,36] and associated with ccRCC oncogenesis[37], and *HNF4A*, a regulator of *HNF1A*[38], along with *PPARA*, the primary transcription factor maintaining metabolic features of normal kidney proximal tubules[39], were involved. Their downregulation alongside the trajectory may be associated with the inhibition of proximal tubule-specific metabolic features in DCCD tumors (Fig. 4a). The androgen receptor is also downregulated in IM4-like single cells. Its loss in ccRCC is related to a higher lymph node metastasis rate[40]. *HIF1A*, a conventional tumor suppressor of RCC[41], is significantly elevated during the later stages of differentiation toward

**Fig. 6 | Subcluster level shift from non-DCCD to DCCD could be observed in ccRCC. a,** Heatmap of IM4 and IM2 signature genes in IM1 and IM3 tumors. Samples are ranked by DCCD score. **b,** IM1 and IM3 groups have intermediate DCCD scores, and *P* values were calculated by Tukey's test. IM1, *n* = 93; IM2, *n* = 159; IM3, *n* = 74 and IM4, *n* = 116. In boxplots, the central line represents the median value, box limits indicate the IQRs and the whiskers extend to 1.5× IQR. **c,** KM plot of survival data from TCGA KIRC. IM1 and IM3 samples were classified into IM4-like or IM2-like according to a positive or negative DCCD score. *P* value was calculated by the log-rank algorithm. *P* values of pairwise comparisons were corrected by the BH algorithm. ****$P_{adj}$ < 0.0001, ***$P_{adj}$ < 0.001, *$P_{adj}$ < 0.05. **d,** UMAP plot of snRNA-seq of malignant cells colored by sample IDs. **e,** UMAP plot of snRNA-seq colored by IM2 score (top) and IM4 score (bottom). **f,** Bar plot showing proportions of IM2-like and IM4-like cancer cells per sample. All malignant cells were mapped to IM2-like and IM4-like cells according to the DCCD score. **g,** Single-nucleic level SCNA in malignant cells of X98_T. Loss of chr 9 could be observed only in IM4-like malignant cells of X98_T. **h,** H&E staining of X98_T. The slice is used for 10× visium profiling. **i,** Spatial signature scores in visium data of X98_T. **j,** Visualization of the chr 9 score on the visium slide shown in 6 h. Dots are colored by chr 9 score. **k,** Differential metabolites between IM2-like and IM4-like regions in Y7_T. Metabolites belonging to carboxylic acids, fatty acyls and glycerolipids were colored according to the metabolite class. The *m/z* values were labeled to differentially distributed fatty acyls. **l,** Spatial distribution of lipid metabolites in Y7_T. Chr, chromosome; H&E, hematoxylin and eosin.

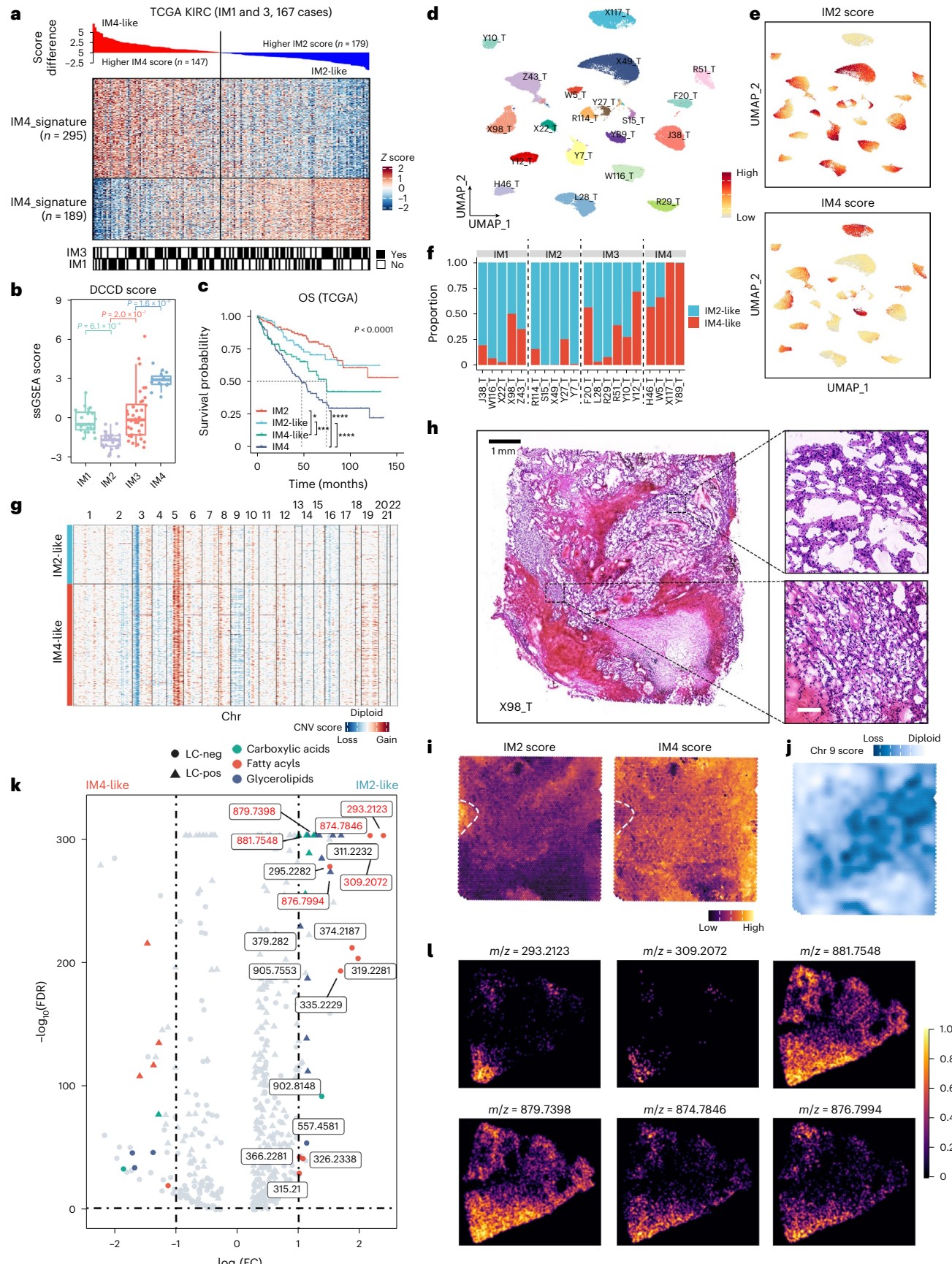

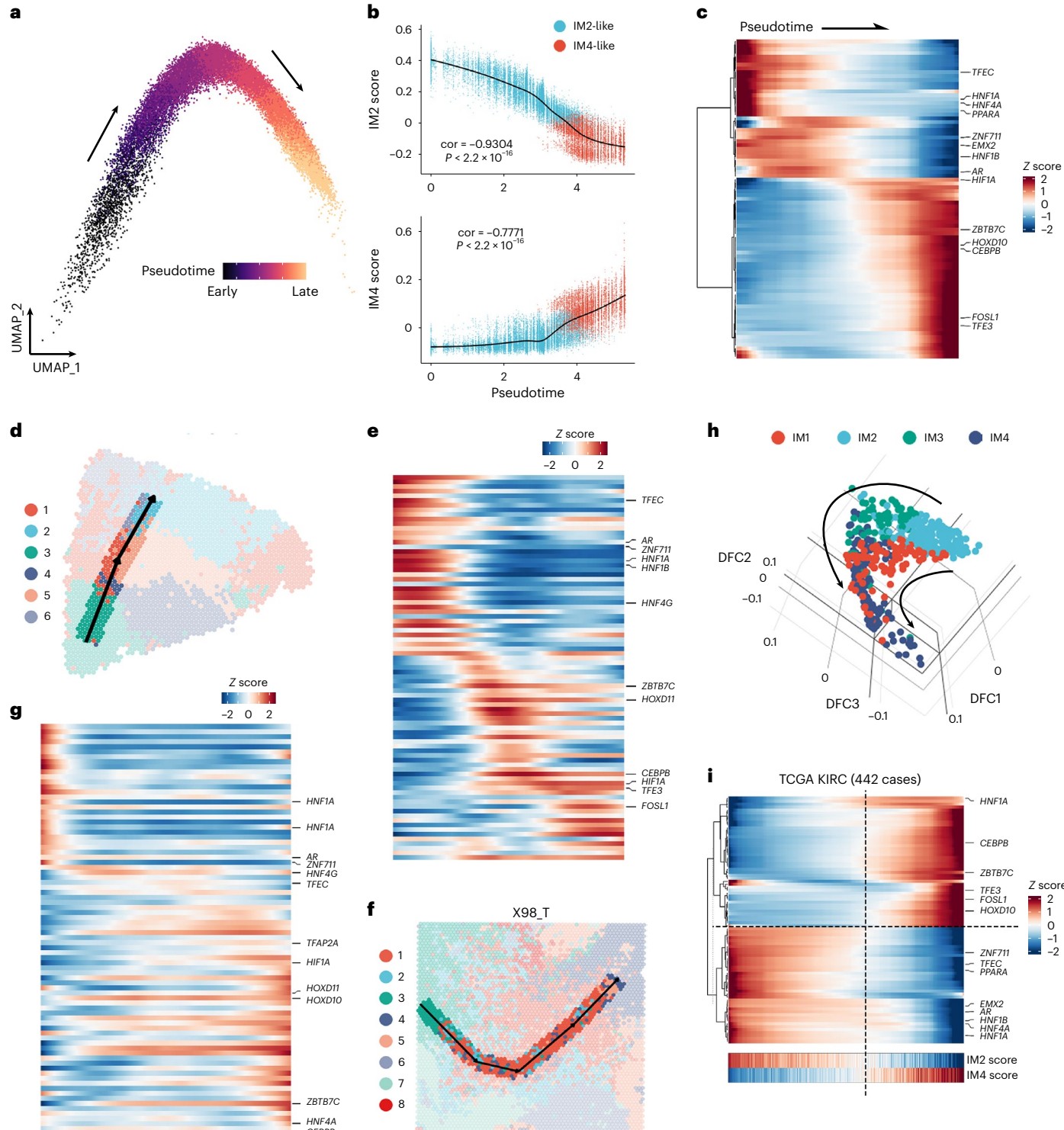

**Fig. 7 | Establishing the trajectory of the DCCD with multi-omics data.**
**a**, Diffusion map of single malignant cells colored by pseudotime. **b**, IM2 and
IM4 scores change throughout pseudotime. Each dot was colored by IM2-like or
IM4-like identity per cell. *P* values were calculated by two-sided Student's *t* test.
**c**, Heatmap showing expression of transcription factors (TFs) gradually changes
throughout pseudotime in snRNA-seq data. Intersected differentially expressed
TFs between IM4-like and IM2-like malignant cells and IM4 and IM4-like and IM2
and IM2-like samples in bulk RNA-seq were used to visualize the heatmap.

**d,f**, Spatial trajectory of transformation from IM2-like to IM4-like patterns in
Y7_T (**d**) and X98_T (**f**). The plots are colored by Seurat clusters, and the arrows
show the transformation direction. **e,g**, Expression of TFs changes along the
spatial trajectory in Y7_T (**e**) and X98_T (**g**). **h**, Three-dimensional diffusion map
of TCGA KIRC samples. The arrow shows the transformation direction from
IM2 to IM4. **i**, Smoothened heatmap of DCCD-related TFs in TCGA KIRC.
Samples are ranked by DCCD score. DFC, diffusion component.

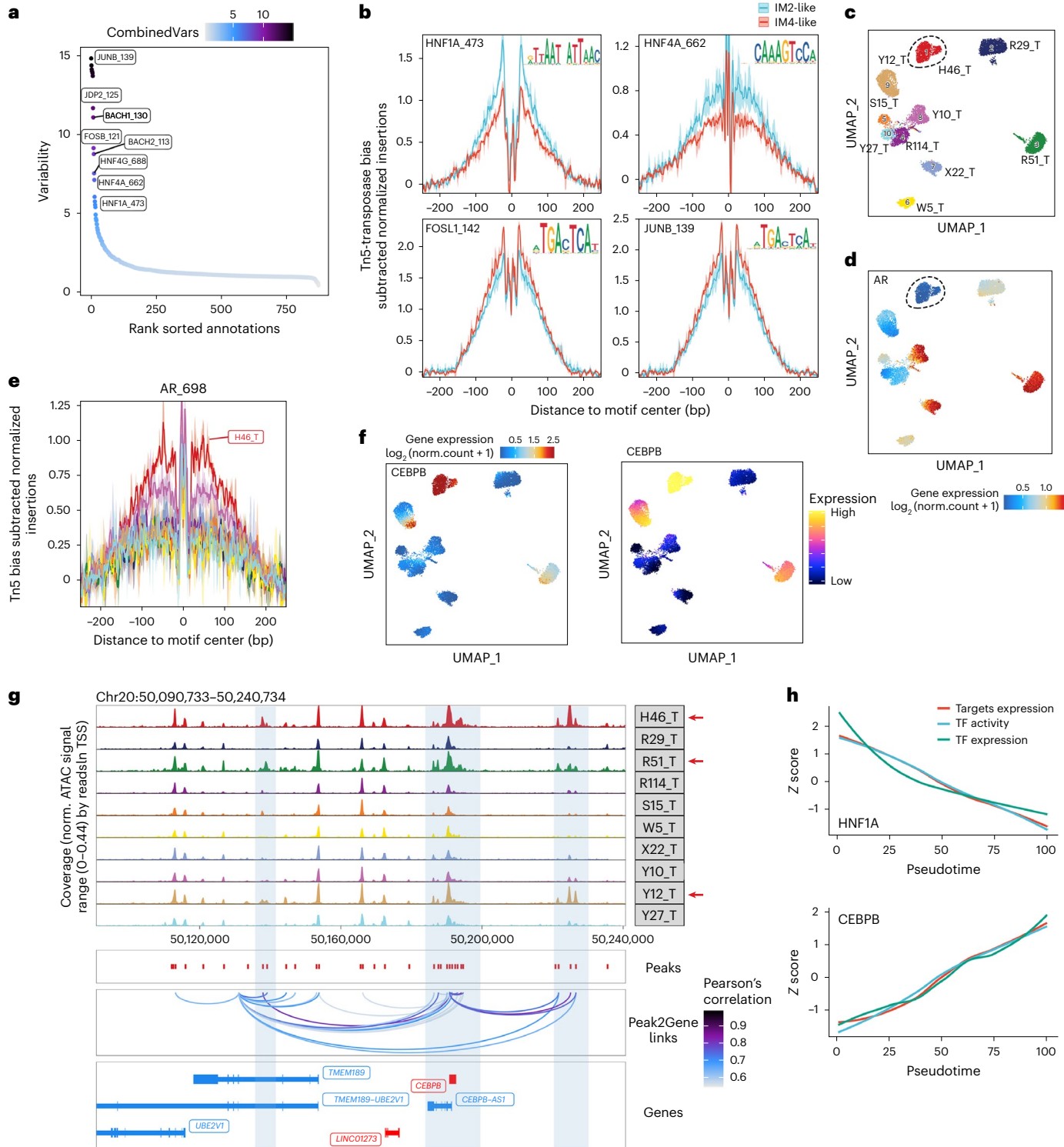

**Fig. 8 | snATAC-seq reveals transcriptional factors involved in DCCD. a,** Motif deviation between IM4-like and IM2-like single nuclei. **b,** Tn5 bias-adjusted footprints of given TFs grouped by IM2-like and IM4-like cancer cells. Cells were grouped based on IM2-like or IM4-like identity to create pseudobulk ATAC-seq profiles. Normalization was performed to subtract the Tn5 bias from the footprinting signal. Data are presented as normalized read depth ± s.e.m. **c,d,** UMAP of snATAC data of cancer cells colored by sample names (**c**) and mRNA expression level of AR (**d**). **e,** Tn5 bias-adjusted footprints of AR in cancer cells grouped by sample names. Samples are colored the same as **c**. Data are presented as normalized read depth ± s.e.m. **f,** UMAP of cancer cells colored by mRNA (left) and predicted gene scores (right). Predicted gene scores represent the predicted expression level of each gene calculated by snATAC-seq data. **g,** Genome accessibility track visualization of CEBPB with peak-to-gene links. Genes translated from 5′ to 3′ are colored in red, while others are colored in blue. **h,** Gradual changes in target expression, TF activity and TF expression alongside pseudotime.

DCCD. Upregulation of *HIF1A* has been linked to lower drug response rates in mRCC[42]. The spatial trajectory of the two samples aligns with the results from the snRNA-seq data (Fig. 7d–g).

Because partial-DCCD samples are mixtures of IM2-like and IM4-like cells, it is possible to establish a trajectory with bulk-seq data based on the DCCD signature score. Therefore, we generated the trajectory using Monocle 2 with the DCCD score as pseudotime. Continuous changes in key transcription factors involved in DCCD were highly consistent between bulk-seq and snRNA-seq data (Fig. 7h,i), suggesting that DCCD is a widespread, continuous biological process in ccRCC.

ChromVAR deviation analysis of snATAC-seq data revealed the most differentially activated transcription factors between DCCD and non-DCCD populations. As shown in Fig. 8a, AP-1 transcription factor subunits were ranked at the top, followed by *BACH1*, *BACH2* and members of the hepatocyte nuclear factor family. These findings were confirmed with motif footprinting analysis (Fig. 8b). Integrated multi-omics data also revealed asynchronous phenomena between the transcriptome and chromatin availability. For example, H46_T is an androgen receptor low-expressing (*AR*[low]) tumor according to both bulk-seq and snRNA-seq. However, motif footprinting analysis revealed that the androgen receptor binding site of H46_T showed the highest availability across all samples (Fig. 8c–e).

Despite the sample-specific peak-to-gene linkage features (Supplementary Fig. 3a), several elements were observed across multiple samples. We identified a peak located approximately 30k upstream from the promoter of *LRP2* that was positively correlated with the expression of *LRP2* (Supplementary Fig. 3b,c). We also identified several peaks correlated with *CEBPB* expression in cancer cells, which may be enhancers that promote the expression of *CEBPB* (Fig. 8f,g). Finally, scMEGA analysis allowed us to construct separate gene regulatory networks for DCCD and non-DCCD cancer cells (Fig. 8h and Supplementary Figs. 4 and 5). Taken together, these data provided insights into the transcription factor regulatory networks involved in the progression of ccRCC.

## Discussion

The integration of multi-omics data catenates multilevel ITH of ccRCC, revealing the interplay between genomic, transcriptomic and metabolic regulation. In this work, we identified a distinct subgroup, DCCD-ccRCC. We also performed spatial metabolomics profiling, revealing that partial metabolic reprogramming contributes to ITH in ccRCC.

Through a TME-based molecular stratification system, we identified a distinct subtype, DCCD-ccRCC, distinguished by absent LDs in cancer cells. LD accumulation has been documented as a protective factor against lipotoxicity and endoplasmic reticulum (ER) stress in malignant cells[15,43–46]. However, results from these cell line-based studies contradict real-world observations from large sequencing cohorts. Both *HIF2A* and *PLIN2*, the key proteins involved in the accumulation of LDs in RCC, are associated with good prognosis in clinical cohorts[47,48]. In this study, our analysis revealed that *HIF2A*-dependent LD accumulation occurs mainly in the IM2 subtype or IM2-like region of ccRCC, reconciling this contradiction. Based on multilevel profiles, we propose that *HIF2A*–*PLIN2* axis-dependent LDs are essential for restricting ER stress and providing energy via the FAO (fatty acid oxidation)/AMPK (AMP-activated protein kinase) pathway in early-stage cancer cells. This could explain why these tumors commonly have a higher response rate to TKI treatment. In contrast, DCCD tumors, characterized by enhanced nutrient uptake, rely less on LDs for energy. Because DCCD signature is highly continuous at both transcriptomic and spatial levels and is not SCNA-dependent, these findings raise the speculation that DCCD is induced by the processes involved in local TME remodeling, such as hypoxia, but additional evidence is needed.

DCCD in ccRCC is related to poor outcomes in all clinical cohorts involved in this study. Even for patients with stage I DCCD tumors, nephrectomy seems insufficient to cure the disease. This discovery challenges the present therapeutic schedule for ccRCC. A potential explanation is that DCCD tumors have increased metastatic potential, seeding to adjacent tissues, draining lymph nodes or distant solid organs to form micrometastasis niches at an early stage. These niches may aid in the initiation of recurrence. If we could confirm the tissue tropism of these DCCD cancer cells and perform targeted extended resection, these diseases may be curable at an early stage. Moreover, postnephrectomy drug treatment may be necessary for DCCD patients, even at stage I.

Although several independent studies suggested that first-line combined treatment could improve the PFS of patients with mRCC[49,50], TKI treatment remains the primary first-line treatment for patients with mRCC. In this study, we determined that combined therapy with anti-VEGF+ immune checkpoint blockade or TKI+ immune checkpoint blockade is not superior to TKI monotherapy with sunitinib for IM2 patients who have not undergone DCCD. In contrast, combined treatment with immune checkpoint blockade significantly improved the prognosis of partial or complete DCCD patients. We also evaluated the utility of molecular subtyping for guiding second-line treatment. In meta-CheckMate[51], regardless of the molecular subtype, the therapeutic efficacy of nivolumab (PD-1 inhibitor) was not inferior to that of everolimus (mTOR inhibitor); thus, this treatment should be recommended as a priority.

In summary, this study identified a special subtype of ccRCC with distinctive metabolic features. It suggests new treatment opportunities for patients with treatment-resistant RCC.

## Online content

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

[1]Department of Urology, Tongji Hospital, Tongji Medical College, Huazhong University of Science and Technology, Wuhan, China. [2]Department of Urology, Union Hospital, Tongji Medical College, Huazhong University of Science and Technology, Wuhan, China. [3]Shanghai Luming Biotech, Shanghai, China. [4]Shanghai OE Biotech, Shanghai, China. [5]Department of Urology, The First Affiliated Hospital of Zhengzhou University, Zhengzhou, China. [6]Department of Pathology and Laboratory Medicine, Indiana University School of Medicine, Indianapolis, IN, USA. [7]Department of Pathology, Baylor Scott & White Medical Center, Temple, TX, USA. [8]Shanghai Institute of Hematology, State Key Laboratory of Medical Genomics, National Research Center for Translational Medicine at Shanghai, Ruijin Hospital Affiliated to Shanghai Jiao Tong University School of Medicine, Shanghai, China. [9]Department of Pathogen Biology, School of Basic Medicine, Tongji Medical College, Huazhong University of Science and Technology, Wuhan, China. [10]Pennsylvania Cancer and Regenerative Medicine Research Center, Baruch S. Blumberg Institute, Philadelphia, PA, USA. [11]Department of Radiation Oncology, Tianjin Medical University Cancer Institute and Hospital, National Clinical Research Center for Cancer; Tianjin's Clinical Research Center for Cancer; Key Laboratory of Cancer Prevention and Therapy, Tianjin, China. [12]Department of Oncology, Tongji Hospital, Tongji Medical College, Huazhong University of Science and Technology, Wuhan, China. [13]Department of Urology, Second Affiliated Hospital of Naval Medical University, Shanghai, China. [14]These authors contributed equally: Junyi Hu, Shao-Gang Wang, Yaxin Hou, Zhaohui Chen. ✉e-mail: deniskwan@gmail.com; ztao@tmu.edu.cn; renshancheng@gmail.com; shenke@hust.edu.cn

## Methods

### Experimental model and participants' details

**Human participants.** A total of 100 participants, with an age range of 27–84, were included in this study. This cohort contained males (*n* = 63) and females (*n* = 37), corresponding to the sex distribution of ccRCC[14]. Histopathological diagnosis was confirmed by at least two different pathologists per sample and only ccRCC cases were included in this sequencing cohort. Institutional review board approval (Tongji Hospital) and informed consent were obtained before tissue acquisition and analysis. All individuals involved granted their consent through written affirmation before they participated in the study. Clinical characteristics are summarized in Supplementary Table 1.

**Publicly available cohorts.** IMmotion 151 is a phase 3 trial comparing atezolizumab (anti-PD-L1) plus bevacizumab (anti-VEGF) with sunitinib in patients with treatment-naive mRCC[32,50]. In total, 826 cases with complete sequencing data were involved in this analysis. Normalized expression matrix and paired clinical data were obtained from the supplementary material of the original paper[32].

JAVELIN is a phase 3 trial involving patients with treatment-naive mRCC that compared avelumab (anti-PD-L1) plus axitinib (TKI) with the standard-of-care sunitinib[33,49]. Normalized expression matrix and paired clinical data were obtained from the supplementary material of the original paper.

Checkmate 025 is a randomized, phase 3 study comparing nivolumab with everolimus in patients with mRCC who had previously been treated with one or two anti-angiogenic regimens[51]. In a study discussed in ref. 34, CheckMate 025 was merged with phase I (CheckMate 009)[52] and phase II (CheckMate 010) cohorts to generate a meta-checkmate dataset. The mixed checkmate cohort with 311 mRCC cases was involved here. Normalized expression matrix and paired clinical data were obtained from the supplementary material of the original paper

### WES sample processing, data collection and analysis

**WES data quantification and mutation calling.** Fastp (v0.20.1)[53], an ultra-fast FASTQ preprocessor, was used to process raw fastq data. Clean reads were then mapped to the reference human genome UCSC (University of California, Santa Cruz) hg38 by Sentieon (v202010.04) BWA[54]. After duplicates removal and base quality score recalibration (BQSR), variant calling was conducted following GATK best practice workflow. Somatic single-nucleotide variations (SNAs) and small indels were detected using Sentieon TNhaplotyper2 (same as MuTect2 in GATK 3.8) and were annotated using ANNOVAR[55] based on known genes in UCSC refGene. SNAs or indels with variant allele frequency (VAF) < 0.05 were filtered out. Five NATs failed to pass the quality control, and no more tissue was available to sequence once again. Hence, paired-tumor samples of these five NATs were not involved in SNA and SCNA calling. Instead, we applied the GATK germline mutation calling workflow and only selected the 50 most frequent SNAs in the other 95 tumors to state the total mutation rate. Somatic mutation rate was stated by the maftools[56] package. SNAs in each sample are deposited in Supplementary Table 5.

**SCNAs calling.** SCNAs calling were performed with CNVkit (v0.9.10)[57] with all the parameters set as default. Only 95 tumor samples with matched NATs were involved in this analysis. Then Genomic Identification of Significant Targets in Cancer (GISTIC2.0, v2.0.23)[58] was applied to identify arm-level events, with *q* < 0.05 considered significant.

**Mutation signature analysis.** Single-nucleotide mutation has six substitution patterns (C > A, C > G, C > T, T > A, T > C and T > G). Together with 5′- and 3′-flanking nucleotides, it could be stratified into 96 base substitutions in trinucleotide sequence contexts. NMF algorithm in sigminer (v2.1.3)[59] package was used to decipher the mutation signatures in somatic mutations. Then the cosine similarity was calculated to map these signatures to the COSMIC[60] database.

### WTS sample processing, data collection and analysis

**RNA quantification.** Fastp[53] was used to process raw fastq data. Clean data were obtained after quality control, adapter trimming, quality filtering and per-read quality cutting. The paired-end clean reads were then aligned to the hg38 human reference genome (UCSC hg38) using HISAT2 (v2.1.0)[61]. All the parameters were set as default for HISAT2. FeatureCounts (v2.0.1)[62] was then applied to count the read numbers mapped to each gene. Batch effect was estimated and removed with the combat-seq algorithm in the sva (v3.42.0) package[63]. Then raw count was transformed into transcripts per kilobase of exon model per million mapped reads (TPM). The count matrix was used in the DEG analysis, and the other analysis of bulk transcriptome was based on the TPM matrix.

### Single-nucleic transcriptome and ATAC-seq

**Processing of snRNA-seq and 10× multiome data.** Raw fastq data of snRNA-seq was processed using Cell Ranger (v6.1.2), with all the parameters set as default. GRCh38 built following the manufacturer's instructions of Cell Ranger was used as the reference genome. Raw data of multiome were processed using Cell Ranger ARC (v2.0.0).

UMI (unique molecular identifier) count matrix generated by Cell Ranger and Cell Ranger ARC was transferred into Seurat (v4.0.5) for further processing. First, single nuclei with nUMI <1,000 or have >10% mitochondrial-derived transcripts were considered as low-quality data and filtered out. Then the UMI count matrix was log-transformed with the NormalizeData function. FindVariableFeatures was performed to call variable genes (nFeatures = 3,000), and then PCA analyses were performed to reduce the dimension (nPCs = 50). Sample number was used to regress out potential batch effect using Harmony (v0.1.0)[64] algorithm. Then, the top 50 Harmony-corrected dimensions were used in the RunUMAP function of Seurat. FindNeighbors and FindClusters (resolution = 1.0) were then performed sequentially to find cell subgroups in ccRCC. Identity of major cell types was determined according to known markers[23]. Cell subpopulations were named based on top markers found by the COSG (v0.9.0)[65] package. Due to the absence of mitochondria and ribosomes in cell nuclei, cell clusters exhibiting a high proportion of transcripts derived from mitochondria or ribosomes, along with the absence of other specific marker genes, were categorized as low-quality data and subsequently excluded. Impact of the cell cycle was estimated using the CellCycleScoring function of Seurat. Two normal kidney-derived samples were processed following the same process. Cell type identification was performed based on markers from the single-cell atlas of human kidney[66,67]. According to the high heterogeneity of tumor cells between different individuals, reclustering of malignant cells was based on 'pca' instead of 'Harmony' to maintain sample-specific features.

**Odds ratio (OR) analysis.** To characterize the subtype preference distribution of cell subclusters, OR was calculated following the code generated in ref. 68 (https://zenodo.org/record/5461803). *P* values were calculated by Fisher's exact test and then adjusted using the BH method. OR > 1.5 and adjusted *P* value < 0.05 indicated that this subpopulation was more preferred to distribute in this immune subtype, whereas OR < 0.5 and adjusted *P* value < 0.05 indicated that it was preferred not to distribute in this immune subtype. Endothelial cells were analyzed separately to investigate the IM subtype preference of endothelial cells.

**Pathway analysis with gene sets variation analysis (GSVA).** To estimate the metabolic difference among cell subpopulations, we implemented GSVA (v1.42.0)[69] analysis in 'ssGSEA' mode. First, to reduce

the computational burden, mean expression per gene was calculated in each cell subpopulation using the AverageExpression function of Seurat. Then, ssGSEA analysis was performed on this normalized matrix using MSigDB hallmark, C2 and C5 gene sets. It was also performed on the gene list defined in this paper (Supplementary Table 6). To estimate the correlation of fibroblast and TAM subcluster, ssGSEA was also conducted on the TPM matrix of TJ-RCC to obtain cell subtype signature per sample, followed by Pearson's correlation analysis in *R*.

**Copy number analysis.** To verify the SCNA difference between IM2-like and IM4-like malignant cells, infercnv[70] was used to estimate CNA in each cell. A baseline reference was generated with the normal renal cortex sample. Because ccRCC is mainly derived from PT of the human kidney, the medulla sample was not involved. The output matrix was visualized with ComplexHeatmap (v2.16.0) package[71].

**Trajectory analysis based on transcription factor profiling.** TFs are known as drivers of the differentiation process. To obtain a DCCD-related TF list, we downloaded a TF list containing all the already known human TFs from GitHub (https://github.com/aertslab/SCENIC)[72]. Then, we intersected DEGs between IM4/IM4-like and IM2/IM2-like groups at both bulk and single-cell levels and intersected them. The result was then intersected with the TF list. Finally, a TF list of 83 TFs was used for further analysis.

Trajectory analysis was performed to describe the transformation of cancer cells from IM2 to IM4-like patterns. Because UMI count has a strong effect on the dimensional reduction process, only half of the malignant cells with higher UMI count were involved in this analysis. First, a diffusion map of tumor cells was constructed using destiny (v3.8.1) package[73], with Harmony-corrected dimensions used as the input data. Then pseudotime was predicted with Monocle 3 (v1.0.0)[74] and scanpy (v1.9.1)[75], which show high similarity, and only the result of Monocle 3 was retained.

**Processing of snATAC-seq data.** SnATAC-seq data from the 10× multiome platform was processed separately. Fragments from ten different tumors were combined by the cellranger-arc aggr algorithm. The output of Cell Ranger ARC was imported into R environment using ArchR (v1.0.2)[76] strategy. Because all snATAC data profiles by 10× multiome have paired snRNA-seq data, only single nuclei that passed the quality control of snRNA-seq were extracted for further analysis.

Enrichment of ATAC-seq accessibility at TSSs was performed to quantify data quality. Single nuclei with TSS <2 or fragments number <1,000 were removed due to low quality. ArchR function, addIterativeLSI and addUMAP were performed sequentially to get a uniform manifold approximation and projection (UMAP) plot of snATAC data. Pseudobulk replicates were generated using the addReproduciblePeakSet function before peak calling with MACS2 (ref. [77]). To identify differential transcription factors mediating differentiation from IM2-like to IM4-like status, motif enrichment analysis was performed using the getVarDeviations function.

Paired snRNA and snATAC data were combined based on cell barcodes by ArchR. Then the correlation between gene expression and chromatin accessibility peaks was evaluated with the getPeak2GeneLinks function with a correlation cutoff of 0.45 and resolution of 1. Footprinting analysis was performed to trace the imprint of TFs on binding sites. It was finished by getFootprints and plotFootprints functions of ArchR, sequentially. Cells were grouped based on identity to create pseudobulk ATAC-seq profiles. Normalization was performed to subtract the Tn5 bias from the footprinting signal. scMEGA (v0.2.0)[78] is a pipeline designed to infer gene regulatory network by using single-cell multi-omics data. It was implemented to create TF modules that drive the difference between IM2-like and IM4-like single nuclei with both snRNA and snATAC data. This process follows the manufacturer's instructions with all the parameters set as default.

## MS sample processing and data collection

**Global proteomics. Global proteomics data interpretation.** The raw spectra from each fraction were searched against the UniProt database (https://www.uniprot.org/) separately by Proteome Discoverer 2.2 (PD 2.2, Thermo Fisher Scientific). The parameters were set as follows: mass tolerance for precursor ion was 10 ppm and mass tolerance for product ion was 0.02 Da. In Proteome Discoverer 2.2, we specified carbamidomethyl as a fixed modification, oxidation of methionine as a dynamic modification and acetylation as an N-terminal modification. Missed cleavage sites were allowed up to two times.

To improve the quality of the analysis results, the retrieval results were further filtered in Proteome Discoverer 2.2 as follows: peptides with credibility higher than 99% were identified as peptide spectrum matches (PSMs). A protein containing at least one unique peptide is defined as an identified protein. Only the identified PSMs and proteins were retained. False discovery rate (FDR) was calculated, and PSMs and proteins with FDR ≥1.0% were filtered out.

The result of peptide and protein identification in Proteome Discoverer 2.2 software was imported into Spectronaut (version 14.0, Biognosys) to generate a library. To generate a target list, peptides and ion-pair selection rules were set to select the qualified peptides and product ions from the spectrum[79]. After that, the DIA data were imported into the software, and the ion-pair chromatographic peaks were extracted according to the target list. Then we matched the ion and calculated the peak area to qualify and quantify the peptides. The iRT added into the sample was used to correct the retention time, and the *q* value cutoff of precursor ion was set to 0.01. The output matrix was normalized by the VSN algorithm in NormalyzerDE (v1.12.0) package[80]. The final matrix was saved for downstream analysis.

## Untargeted metabolomics

**LC–MS data processing and metabolites annotation.** The original LC–MS data were processed using Progenesis QI V2.3 (Nonlinear Dynamics) for baseline filtering, peak identification, integral, retention time correction, peak alignment and normalization. Main parameters were set as follows: precursor tolerance, 5 ppm; product tolerance, 10 ppm and product ion threshold, 5%. First, round compound identification was performed based on the precise mass-to-charge ratio (*m/z*), secondary fragments and isotopic distribution using in-house metabolites databases. Second, we also performed further metabolite identification based on the Human Metabolome Database[81] and the METLIN[82] database. Output data were used for further processing. Peaks with a missing value (ion intensity = 0) in more than 50% in either group were removed. Then zero values were replaced by half of the minimum value. Resulting scores were defined as the sum of matching scores of accurate molecular weights of MS1 (20), isotope distribution of MS1 (20) and fragments of MS2 (20). Compounds with resulting scores below 36 (of 60) points were also considered to be inaccurate and removed. The final data matrix was a combination of both the positive and negative ion data.

**GC–MS data processing and metabolites annotation.** The raw GC–MS data in .D format were transferred to .abf format using Analysis Base File Converter. Then, all the data were imported into MS-DIAL[83] to perform peak detection, peak identification, MS2Dec deconvolution, characterization, peak alignment, wave filtering and missing value interpolation. Metabolite characterization is based on in-house LUG database (untarget database of GC–MS from Luming Bio) and a data matrix was derived, including sample information, the name of the peak of each substance, retention time, retention index, *m/z* ratio and signal intensity. In each sample, all peak signal intensities were segmented and normalized according to the internal standards with relative standard deviation (RSD) greater than 0.3 after screening.

After normalization, redundancy removal and peak merging were performed to obtain the final data matrix.

## ST

**Processing of ST data.** Raw sequencing data in fastq format were processed with Space Ranger (version 2.0.0). The output was then transferred into R environment using SPATA2 (v0.1.0)[84]. Quality control and dimension reduction were finished automatically in this process. Before further analysis, runAutoenconderDenoising was performed to denoise the expression matrix, which is based on a neural network model. Spatial CNV analysis was performed using runCnvAnalysis of SPATA2, which is based on infercnv. ST data from a normal renal cortex sample were used to generate a reference genome. Pathway score of pyrimidine metabolism was visualized using the plotRidgeplot function with 'ssGSEA' mode. IM2 and IM4 scores were conducted using the plotSurfaceAverage function. Spatial trajectory was constructed by the createTrajectories function, with IM2-like region set at the beginning.

## Spatial metabolomics

**Data acquisition and MSI analysis.** AFADESI-MSI is a spatially resolved metabolomics approach discussed in ref. 85. The analysis was carried out with an AFADESI-MSI platform (Beijing Victor Technology) along with a Q-Orbitrap mass spectrometer (Q Exactive, Thermo Fisher Scientific). ACN/H2O (8:2) was used as the solvent formula at negative mode and ACN/H2O (8:2, 0.1% FA) at positive mode. The parameters were set as follows: the solvent flow rate, 5 µl min$^{-1}$; the transporting gas flow rate, 45 l min$^{-1}$; the spray voltage, ±0 kV; the distance between the sample surface and the sprayer, 3 mm; MS resolution, 70,000; scan range, 70–1000 Da; scan mode, full MS; the automated gain control target, $2 \times 10^6$; the maximum injection time, 200 ms; the S-lens voltage, 55 V and the capillary temperature, 350 °C. The MSI experiment was carried out with a constant rate of 0.2 mm s$^{-1}$ continuously scanning the surface of the tumor or NAT section in the $x$ direction and a 0.1 mm s$^{-1}$ vertical step in the $y$ direction. The scanning area was 10 mm × 10 mm.

**MSI data processing.** Raw MSI data in .raw format was converted into .imML format and imported into MSiReader (v1.02)[86] to perform ion image reconstruction and background subtraction. SmetDB database and pySM annotation framework were used to perform FDR-controlled metabolite annotation for high-resolution imaging MS. Region-specific MS profiles were precisely extracted by matching both histological features and ST. IM2-like and IM4-like regions were selected, respectively. Wilcoxon test was used to identify differentially distributed metabolites between IM2-like and IM4-like regions within the same sample. $P$ values were adjusted by the FDR strategy. Differential metabolites with |log$_2$(FC)| > 1 and adjusted $P$ value < 0.05 were considered statistically significant.

## Statistical analysis

**PCA.** We performed PCA analysis on transcriptomics (TPM), proteomics and metabolomics data matrix of all the samples to get a general view of the difference between tumor and NATs. PCA analysis was conducted with R base packages, and the top two dimensions were used to visualize the results.

**Tumor versus NATs differential genes, proteins and metabolites analysis.** WTS count matrix experienced batch correction was used to identify DEGs between tumor tissues and NATs. DEGs analysis was conducted using edgeR (v3.36.0)[87] package. Genes with |log$_2$(FC)| > 1 and adjusted $P$ value < 0.01 were considered upregulated genes, while |log$_2$(FC)| < −1 and adjusted $P$ value < 0.01 were considered downregulated genes. The same approach was also used to obtain DEGs between immune subclusters (one IM subtype versus the other three subtypes).

For DIA-based proteomics data, the Wilcoxon test was conducted to evaluate the differential abundance (DA) of proteins between tumors and NATs. Cutoff value was set the same as transcriptomics.

Wilcoxon test was performed to determine the DA of metabolites. The $P$ value was adjusted using the Benjamini–Hochberg (BH) procedure. |log$_2$(FC)| > 1 and adjusted $P$ value < 0.01 were considered as statistically significant.

**DA score of metabolomics.** The DA score represents the tendency of a pathway whether have increased or decreased levels of metabolites, relative to the control group[6]. It was defined as the ratio of the difference between upregulated and downregulated metabolites to all the measured metabolites in this pathway, as defined in ref. 6. The Kyoto Encyclopedia of Genes and Genomes (KEGG) metabolites database was used to calculate the DA scores.

**Enrichment of altered metabolites.** Altered metabolites in IM4 compared to other tumors and IM1–IM3 to NATs were calculated as described above. To identify the pathways with the greatest changes between subtypes, these metabolites were then imported into Metaboanalyst 5.0 (ref. 88) to perform pathway enrichment analysis, respectively. Pathway impact was used to estimate the metabolic difference among ccRCC subtypes.

**Gene set enrichment analysis.** To evaluate the pathway-level difference between tumors and NATs, we performed ssGSEA analysis in GSVA[69] package to calculate the pathway score of known gene sets collected in hallmark, C2, C5 of MSigDB per sample in both transcriptomics and proteomics data. Then a linear model from the Limma (v3.50.0)[89] package was applied to calculate the difference between tumor and NATs. When comparing differences between multiple groups, differential analysis was performed by comparing the selected group to all other samples. ssGSEA was also implied to determine cell abundance using TPM matrix based on manually defined cell signatures (Supplementary Tables 6 and 7). $Z$ score transformation was performed to allow easier visualization in heatmaps.

To evaluate the correlation between T cells and pyrimidine metabolism, Pearson's correlation analysis was conducted between the ssGSEA score of effector T (T$_{eff}$) cell and pyrimidine metabolism.

For metabolic analysis of four immune subgroups, GO terms with 'metabolic', 'catabolic' or 'biosynthesis' were selected and manually checked. Finally, ssGSEA scores of 1,001 metabolic-associated GO terms were used for pathway-level analysis.

**xCell analysis.** xCell[16] predicted cell composition matrix of TCGA KIRC was downloaded from the website (https://xcell.ucsf.edu/). xCell was also performed on the TPM matrix of TJ-RCC using the online tool.

**Pathway enrichment analysis.** GO enrichment analysis was performed using clusterProfiler (v4.2.0)[90]. Four gene modules in Figs. 2a and 4a were analyzed separately. To prevent discrimination caused by stromal cells, all extracellular matrix-related genes were excluded in this analysis. The $q$ value cutoff was set as 0.05 and the $P$ value cutoff was 0.01. The $P$ values were adjusted by the BH procedure. Adjusted $P$ value < 0.01 was considered statistically significant.

**Correlation between SCNA, transcriptome and proteome.** To evaluate the effect of SCNAs, Spearman's correlations between gene-level SCNA values returned by GISTIC2 and mRNA or protein abundances were calculated and visualized by multiOmicsViz (v1.18.0)[91]. FDR threshold was set at 0.01 for both mRNA and protein analysis.

**Extraction of TME signature from scRNA-seq data.** Because considerable cell types have been confirmed to have tissue-specific or cancer-specific features, a signature matrix extracted from single-cell

sequencing data obtained from ccRCC may help us better understand the ITH of ccRCC. Therefore, single-cell sequencing data of ccRCC, profiled by a study discussed in ref. 20, was used to generate a signature matrix that stratifies features of all the cell types in the TME of ccRCC. Data of NATs were not involved. All the analysis was performed using Seurat (v4.0.5)[92]. At first, single cells with UMI counts less than 1,000 or over 30% mitochondrial-derived transcripts were considered as low quality and filtered out. The UMI count matrix was then log-transformed with the NormalizeData function. The top 3,000 variable genes were called by FindVariableFeatures, and then PCA analyses were performed to find the top 50 PCAs. The patient-derived batch effect was removed with the Harmony[64] algorithm, with all the parameters set as default. Then, the top 50 Harmony-corrected dimensions were used in the RunUMAP function of Seurat. FindNeighbors and FindClusters (resolution = 1.0) were then performed sequentially to find cell subgroups in ccRCC. Major cell groups were identified with previously known markers[23], and cell subgroups were annotated by the top marker of each cell cluster. Finally, 25 different cell types were annotated.

FindAllMarkers function was then performed to identify markers of each cell subgroup. DEGs with |log$_2$(FC)| > 1 and adjusted $P$ value < 0.001 were considered as candidates. To balance the weight of each cell subgroup in the clustering of bulk-seq data, only the top 60 markers of each subpopulation with the highest log$_2$(FC) were restrained. Finally, we obtained a 911-gene signature matrix of ccRCC and subjected it to molecular subtyping analysis.

**Identification of TME-based molecular subtypes of ccRCC.** TPM matrix of 911 signature genes selected in scRNA-seq was used to identify molecular subtypes. Unsupervised clustering of RNA-seq data was performed with ConsensusClusterPlus 2 (v1.58.0)[93]. The most suitable $k$ value was selected according to the relative change in area under the CDF curve. Finally, $k$ value was set at 4, and four immune subgroups were identified, with distinct features in multi-omics data.

**Extending IM subgroups to other ccRCC cohorts.** First, we selected the signature genes with higher expression in one group relative to others for each of the four subtypes using |log$_2$(FC)| > 1 and adjusted $P$ value < 0.01 as the criteria. $P$ values were calculated by Wilcoxon's test and adjusted by the BH procedure. Signature genes used here are listed in Supplementary Table 8. We identified 226 signature genes for IM1, 189 for IM2, 146 for IM3 and 295 for IM4. Compared to 911 immune-based signature described above, this 856-gene signature is influenced by the gene expression of both malignant cells and TME cells.

To match the ccRCC samples to each subtype, we used the nearest template prediction (NTP) algorithm[94] packaged in CMScaller[95] to assign each patient to a molecular subtype. The log$_2$-transformed expression matrix was normalized by the ematAdjust function with the RLE method and then annotated by ntp function (nPerm = 1,000, seed = 42, nCores = 30). Four independent cohorts, including TCGA KIRC, IMmotion 151, JAVLIN and checkmate renal, were analyzed separately.

**Survival analysis.** KM analysis was performed using the survminer package to evaluate prognosis value of molecular subtype system in four cohorts, respectively. log-rank $P$ value < 0.05 was considered statistically significant. Pairwise comparison was conducted using the pairwise_survdiff algorithm in survminer. $P$ values were corrected by the BH algorithm. HR (harzard ratio) was calculated using COX regression model and visualized by forestploter.

**IHC staining.** OCT (optimal cutting temperature compound)-embedded frozen samples were used to perform IHC staining. All the samples used to perform IHC staining were involved in multi-omics sequencing. IHC staining was performed as previously described.

All the slices were screened by Pannoramic DESK (3DHISTECH) and analyzed with CaseViewer 2.2. The following antibodies were used: anti-human CD8 (Abcam, ab178089; 1:100), anti-human CD31 (Proteintech, 11265-1-AP; 1:2,000), anti-Human DCN (Abcam, ab277636, 1:2,000), anti-human CD3 (Proteintech, 17617-1-AP; 1:1,000), anti-human CD163 (Proteintech, 16646-1-AP, 1:2,000), anti-human CX40 (Bioss, bs-1050R; 1:500) and anti-human a-SMA/ACTA2 (Proteintech, 14395-1-AP; 1:2500). CD31 marked endothelial cells, DCN marked fibroblasts, α-SMA marked SMCs, pericytes and fibroblasts, CD8 marks CD8$^+$ T cells, CD163 marked macrophages[13], CX40(GJA5) marked arterial vasculature and ACKR1 marked venous vasculature. Goat anti-rabbit IgG H&L (horseradish peroxidase (HRP); Abcam, ab205718;1:2000) was used as the secondary antibody.

**Multicolor immunofluorescence staining.** Multicolor immunofluorescence was conducted on a series of slices of those used to perform IHC. Antigen was retrieved by EDTA antigen repair buffer (pH = 8.0; Powerful Biology, B0035) for 15 min. Endogenous peroxidase was inactivated by incubation in 3% $H_2O_2$ at room temperature for 25 min. Nonspecific sites were blocked by incubating in 3% BSA for 30 min. Incubation in primary antibodies, HRP-labeled secondary antibody and TSA (tyramide signal amplification)-conjugated fluorescein were repeated sequentially for four cycles. The slices were heated by microwave in EDTA antigen repair buffer (pH = 8.0) for 25 min after each cycle. The primary antibodies used in the validation of colocalization of CAFs and IGF1+ TAMs were as follows: anti-human CD31 (Proteintech, 11265-1-AP; 1:2,000), anti-human ACKR1 (SAB, 56458; 1:200), anti-human PDGFRA (Abcam, ab203491; 1:500), anti-human F13A1 (Abcam, ab76105; 1:100) and anti-human CD8 (Abcam, ab178089; 1:100). Secondary antibodies were as follows: goat anti-rabbit IgG H&L (Cy3) preadsorbed (Abcam, ab6939; 1:200), goat anti-rabbit IgG H&L (Alexa Fluor 488; Abcam, ab150077; 1:500), goat anti-rabbit IgG H&L (Alexa Fluor 594; Abcam, ab150080;1:500), goat anti-rabbit IgG H&L (Cy5) preadsorbed (Abcam, ab6564;1:500). The antigenic binding sites were scanned using the PannoramicMIDI (3DHISTECH) according to the manufacturer's protocol and processed in CaseViewer 2.2.

**ORO staining.** OCT-embedded frozen samples used for IHC staining were also involved in ORO staining. The frozen sections were rewarmed, dried and fixed in 4% paraformaldehyde for 15 min. After washing and drying in the air, the slices were dipped in an oil red working solution for 8–10 min and then washed with water. After differentiating in 75% alcohol solution for 2–5 min, the slices were washed with water again. Then the slices were incubated in Gill's or Mayer's hematoxylin for 30 s to stain the cell nuclei and washed with water. The slices were then differentiated shortly in hydrochloric ethanol and washed thoroughly in running tap water for 3 min. Finally, the slices were dried and mounted with glycerin jelly.

**Estimating the DCCD score of TCGA KIRC and snRNA-seq.** To evaluate the IM2-like or IM4-like feature of IM1 and IM3 samples, we conducted a signature scoring strategy similar to a previous work[96]. First, we used marker genes of IM2 and IM4 subtypes as signature gene sets (Supplementary Table 8) and obtained signature scores by ssGSEA analysis. To balance the weight of the IM2 and IM4 scores, we scale-centered the IM2 and IM4 scores, respectively. The final score was defined as the IM4 score minus the IM2 score. Samples with score >0 were defined as IM4-like, whereas samples with score <0 were defined as IM2-like. The same procedure was performed on snRNA-seq data. When analyzing malignant cells of snRNA-seq data, all the single nuclei were mapped to IM2-like and IM4-like cells according to the DCCD score.

**Pseudotrajectory of TCGA KIRC samples.** To evaluate dynamic changes alongside the DCCD process in bulk-seq data, we constructed 3D diffusion map of TCGA KIRC samples with destiny package.

Similar distribution as snRNA-seq data, with IM2 samples localized at the beginning and IM4 samples at the end, could be observed. Then samples were ranked by DCCD score and the heatmap of TFs was smoothened by the genSmoothCurves function of Monocle (v2.22.0)[97].

## Inclusion and ethics statement

This research adhered to ethical standards consistent with the 1964 Helsinki Declaration and its subsequent amendments. The study protocols and consent procedures received approval from the Ethics Committee of Wuhan Tongji Hospital.

All participants were informed about the study's nature and their rights, and their written informed consent was obtained. All methods were performed following relevant guidelines.

## Reporting summary

Further information on research design is available in the Nature Portfolio Reporting Summary linked to this article.

## Data availability

Raw sequencing data have been uploaded to the GSA-Human database[98,99] under accession code PRJCA014547 (https://ngdc.cncb.ac.cn/bioproject/browse/PRJCA014547), but a DAC (discretionary access control) approval is necessary due to policy restrictions. Every researcher could submit an application on the website, and it would commonly take several weeks for the database administrator and DAC to review. All the processed sequencing data have been uploaded to Zenodo (https://zenodo.org/record/8063124) and figshare[100] (https://doi.org/10.6084/m9.figshare.24599295). Expression matrix of TCGA KIRC along with clinical features was obtained from UCSC Xena (https://xenabrowser.net/datapages/?cohort=GDC%20TCGA%20Kidney%20Clear%20Cell%20Carcinoma%20(KIRC)&removeHub=https%3 A%2 F%2Fxena.treehouse.gi.ucsc.edu%3A443). JAVLIN and checkmate datasets were obtained from the supplementary material of the original papers[33,34]. Data of IMmotion 151 (ref. 32) was obtained from the EGA (European Genome-Phenome Archive) database (https://ega-archive.org/studies/EGAS00001004353) with approval from the DAC. Single-cell sequencing data of ccRCC were downloaded from Mendeley (https://doi.org/10.17632/nc9bc8dn4m.1). FUSCC refers to the ccRCC cohort profiled by a team from Fudan University Shanghai Cancer Center (FUSCC). Processed WES data of FUSCC were obtained from NODE (https://www.biosino.org/node) under project ID: OEP000796. Peking University (PKU) refers to the ccRCC cohort profiled by a team from PKU. WES data of the PKU cohort collected under PRJNA596359 (https://www.ncbi.nlm.nih.gov/sra/?term=PRJNA596359) were downloaded from the SRA database. COSMIC database[60] (https://cancer.sanger.ac.uk/cosmic) was used to annotate the SBS signatures in WES data. Source data are provided with this paper.

## Code availability

Scripts for downstream analysis are available on GitHub (https://github.com/AndersonHu85/ccRCC_multiomics). No custom code was developed for this study.

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

## Acknowledgements

This work was supported by grants from the Research Foundation for Talented Scholars of Tongji Hospital (2021RCYJ005), the National Natural Sciences Foundation of China (82373330 and 82073356), Beijing-Tianjin-Hebei Basic Research Cooperation Special Project (22JCZXJC00180), the Science & Technology Development Fund of Tianjin Education Commission for Higher Education (2021ZD034) and Bethune Tumor Radiotherapy Translational Medicine Research Fund (FLZH202108). The authors would like to thank J. Lu, P. Ji and T. Yang from Shanghai Luming Biotech for providing AFADESI spatial-resolved metabolomics detection and help with bioinformatic analysis and metabolomic data analysis. The authors would also like to thank Z. Zhang, C. Wang and N. Li from Shanghai OE Biotech for their assistance in wet experiments of snRNA-seq, multiome and visium spatial transcriptome sequencing; HaploX Genomics Center for the library construction and sequencing of WES, WTS and proteomics profiles and the help of snRNA-seq; NovelBio Bio-Pharm Technology for the support of snRNA-seq experiment; Powerful Biology for the help of ORO staining, IHC staining and multicolor immunofluorescence staining; the Laboratory Animal Center, Huazhong University of Science and Technology for the help with sample processing; and X. Wang of Dynamic Biosystems for the assistance in project management. Ideagram in Figs. 1a and 4j and Extended Data Fig. 4g were drawn using BioRender (https://app.biorender.com/).

## Author contributions

K.C., S.R., S.W. and Z.T. conceived and designed the study. W.G., Z.C., S.W. and L.L. performed experiments or data collection. J.H., L.R. and Z.Y. performed computational, multi-omic and statistical analyses. Y.H., L.L., C.D., Y.L. and Z.D. finished data interpretation and biological analysis. J.H., S.W., Y.H. and Z.C. wrote the original drafts. K.C., S.R., Z.T., Y.H., H.L., T.W., Z.L., X.J., Y.Y. and L.W. reviewed and edited the original draft. N.L., F.L., L.Z., L.P.W., X.Y., C.D., Y.C.L., Z.D., Y.D. and Y.K. supervised all aspects of the study.

## Competing interests

The authors declare no competing interests.

## Additional information

**Extended data** is available for this paper at https://doi.org/10.1038/s41588-024-01662-5.

**Correspondence and requests for materials** should be addressed to Wei Guan, Zhen Tao, Shancheng Ren or Ke Chen.

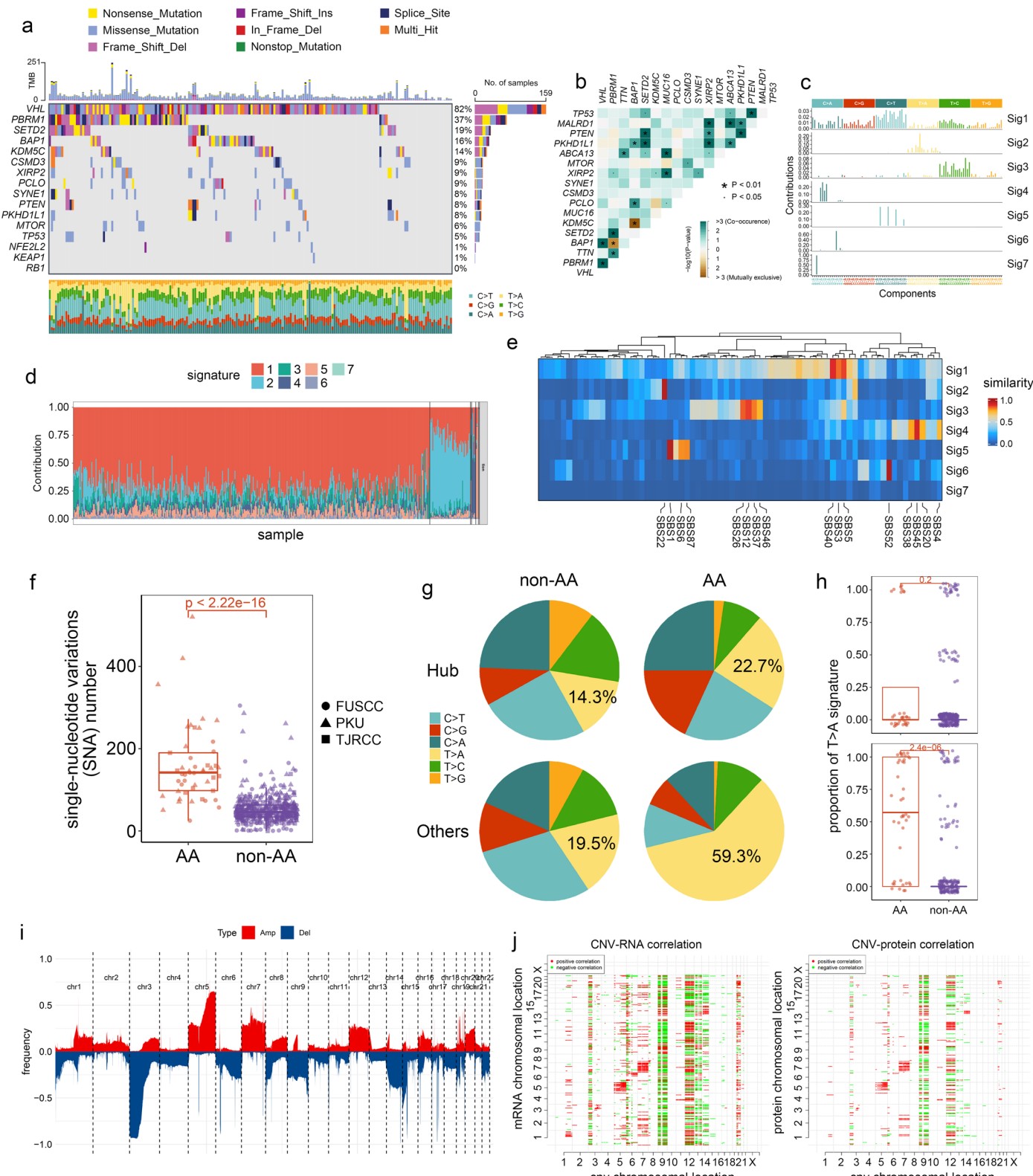

**Extended Data Fig. 1 | See next page for caption.**

**Extended Data Fig. 1 | Multi-omics characterization of TJ-RCC cohort.**
(**a**): Genomic profiles of PKU cohort. Ins: insertion; Del: deletion. (**b**): Co-occurrence of frequent nonsynonymous mutations in ccRCC. Combined TJ-RCC, PKU and FUSCC cohorts were used in the analysis. P-values were calculated by one-sided Fisher's exact test. (**c**): Seven mutational signatures deciphered from the combined WES data from 490 ccRCC genomes. (**d**): Contributions of each mutational signature in each sample. Combined TJ-RCC, PKU and FUSCC cohorts were used in the analysis. (**e**): Similarity of 7 mutational signatures versus COSMIC collected signatures. (**f**): The nonsynonymous mutational burdens of aristolochic acid-associated (AA) and not associated (non-AA) tumors in meta-cohort. Shape of each dot represents sample from different cohorts. P value was calculated by two-side student's t test. In boxplots, the central line represents the median value, box limits indicate the interquartile ranges, and the whiskers extend to 1.5 times the interquartile range. FUSCC: cohort from Fudan University Shanghai Cancer Center. PKU: cohort from Peking University. (**g**): Proportion of 6 single-nucleotide mutation patterns in given gene regions. Hub genes: *VHL*, *PBRM1*, *BAP1*, *SETD2* and *KDM5C*. (**h**): Proportion of T>A signature in each sample. P value was calculated by two-sided Student's t test. Top: n = 32 AA tumors versus n = 233 non-AA tumors. Bottom: n = 41 AA tumors versus n = 161 non-AA tumors. Tumor samples without mutation in targeted genes were ignored here. The central line of the box represents the median value, box limits indicate the interquartile ranges. (**i**): Frequency of SCNAs in TJ-RCC. Copy number gains and losses are colored in red and blue, respectively. Del: deletion; Amp: amplification. (**j**): Pearson's correlations of gene-level copy number variations (CNVs) (x-axis) with mRNA (left) and protein abundance (right). Only significantly positive (red) and negative (green) correlations are shown (q < 0.01).

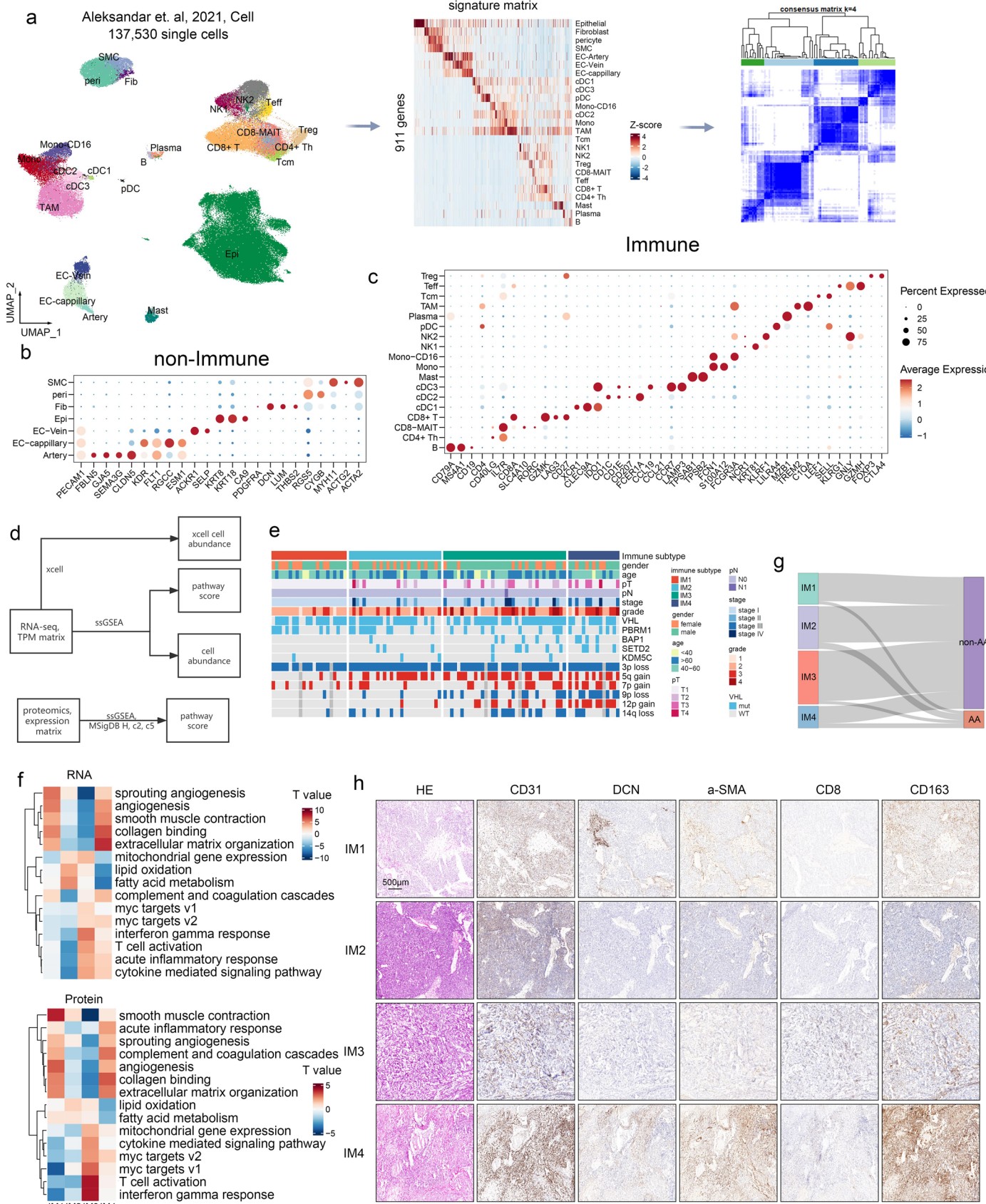

**Extended Data Fig. 2 | See next page for caption.**

**Extended Data Fig. 2 | TME-based molecular subtyping of ccRCC.**
(**a**): Working pipeline of TME-based molecular subtyping. Single cells were clustered into 22 cell types (left). The signature matrix showed the marker genes identified in the scRNA-seq data (middle). The consensus matrix showed result of ConsensusClusterPlus when k = 4 (right). Data from ref. 20. (**b**,**c**): Dot plot showing markers used to identify cell identities. The dot size represents percent of cells detected specific genes in this cell subgroup. (**d**): Working pipeline and visualization strategy of Fig. 2a. (**e**): Key clinical characteristics, somatic mutations and SCNAs in each sample of TJ-RCC, related to Fig. 2a.

(**f**): Heatmap showing T values from differential analysis using linear algorithm of Limma; Differential analysis was conducted via comparing ssGSEA scores of the samples of one IM subtype versus all other subtypes were compared. (**g**): Relationship between IM subgroups and AA subgroups. (**h**): IHC staining on tumors belonging to different immune subtypes. CD31 marked endothelial cells, DCN marked fibroblasts, a-SMA marked SMCs, pericytes and fibroblasts, CD8 marks CD8+ T cells and CD163 marked macrophages. Experiment has been repeated three times.

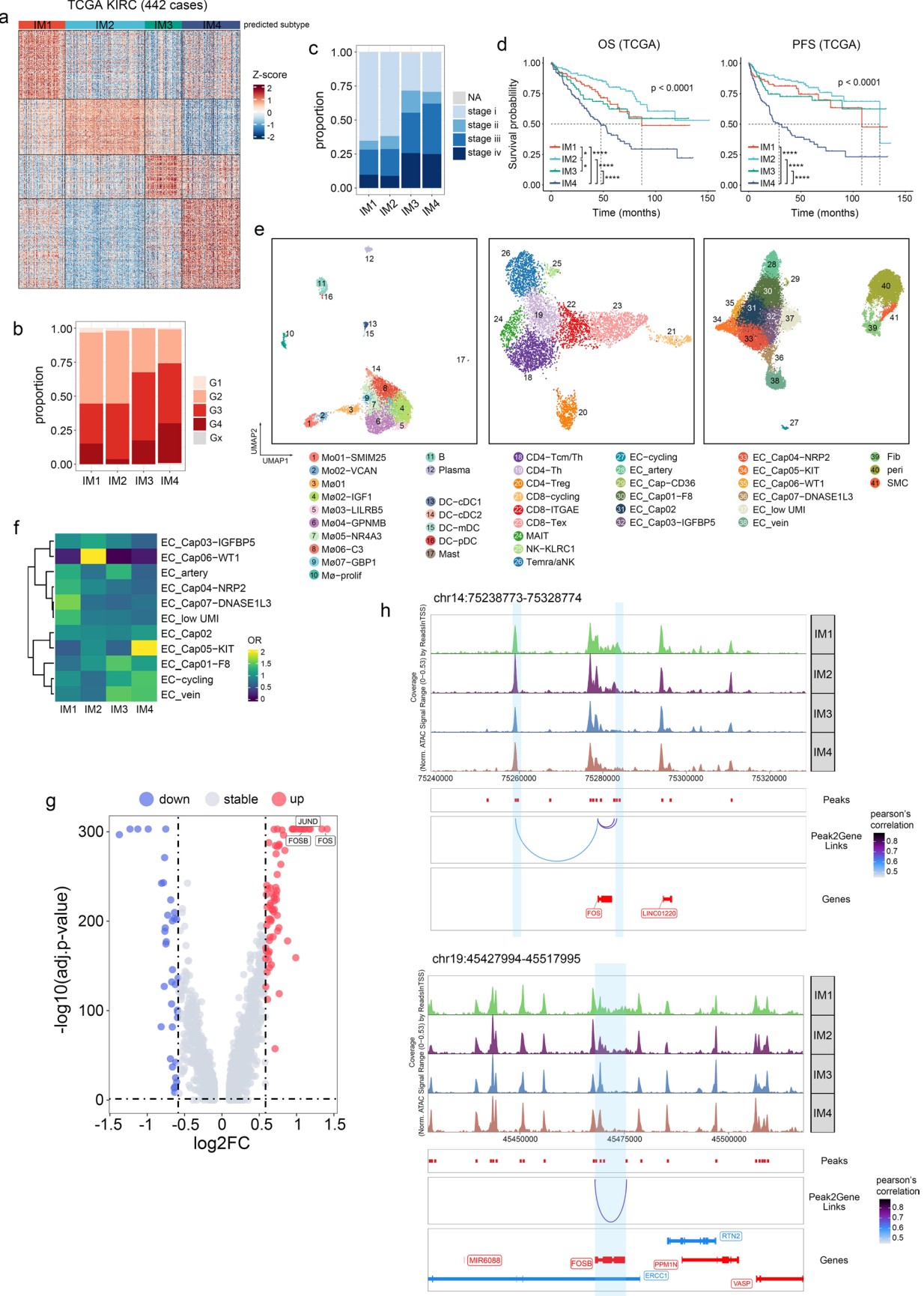

**Extended Data Fig. 3 | See next page for caption.**

**Extended Data Fig. 3 | Decoding intra-tumoral heterogeneity of each IM subgroup.** (**a**): Heatmap of signature genes in TCGA KIRC grouped by NTP predicted subtypes. (**b**,**c**): Contributions of each tumor grade and stage in 4 immune subtypes in TCGA KIRC. (**d**): KM plot of TCGA KIRC grouped by immune subtypes. The global p-value was calculated by log-rank algorithm. Square brackets show the result of pairwise comparison. ****: adj.p.value < 0.0001, *: adj.p.value < 0.05. (**e**): UMAP plot showing 17 myeloid sub-clusters (left), 9 lymphocyte sub-clusters (middle) and 15 endothelial sub-clusters (right). Cell subgroups were named by top marker identified using snRNA-seq data.

Mo: monocyte; Mø: macrophage; EC: endothelial; Cap: capillary; Fib: fibroblast; peri: pericyte; SMC: smooth muscle cell. (**f**): Heatmap of odds ratio (OR) showing different cell abundance of stromal cell types between immune subtypes. P-value was calculated by one-sided Fisher's exact test and corrected by BH algorithm. (**g**): Volcano plot showing differentially expressed genes between IM3-4 versus IM1-2 derived endothelial cells. P-values were calculated by Wilcoxon *t* test and adjusted by BH algorithm. (**h**): Genome accessibility track visualization of *FOS* (top) and *FOSB* (bottom) with peak-to-gene links. Genes translated from 5′ to 3′ are colored in red while others are colored in blue.

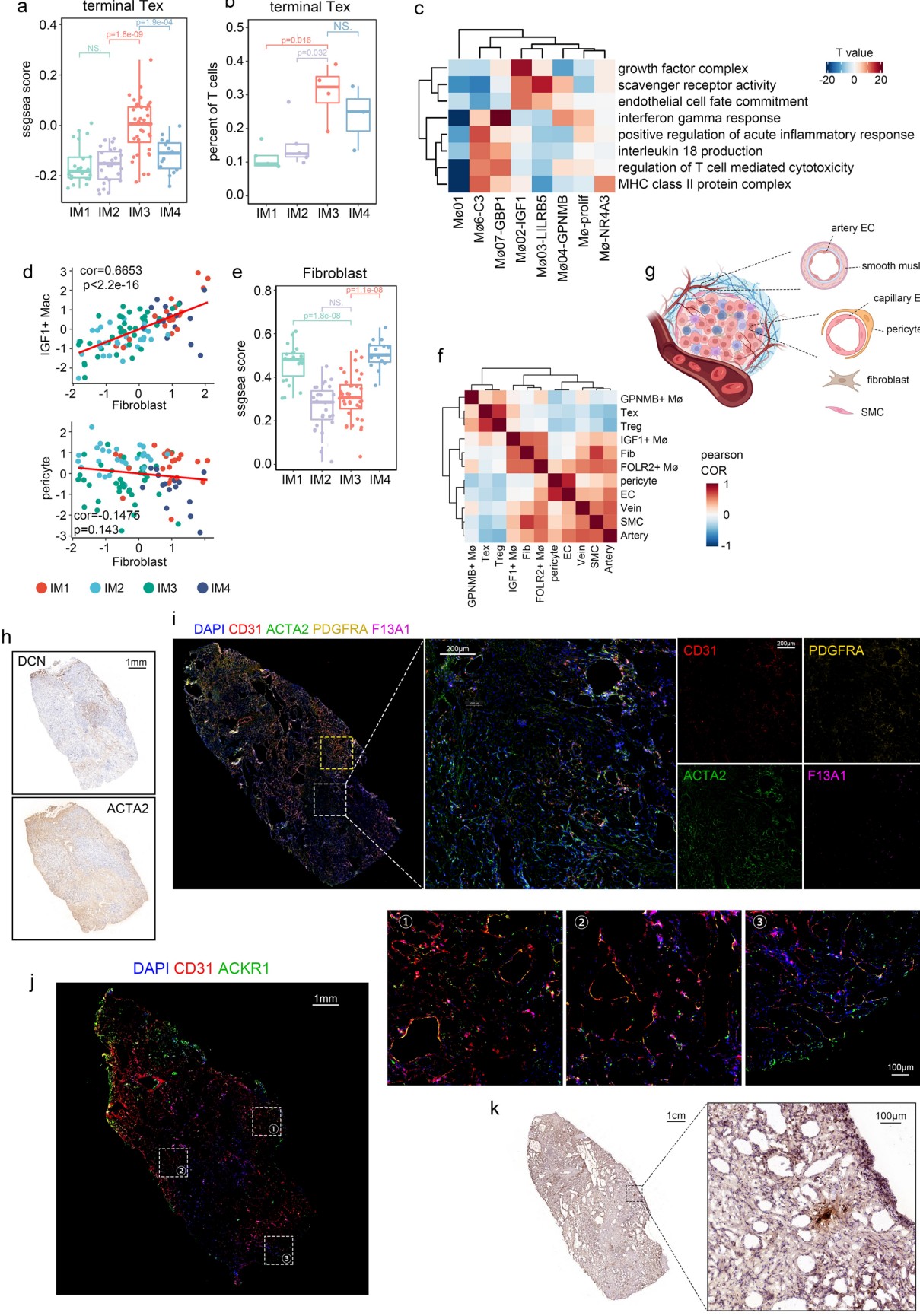

Extended Data Fig. 4 | See next page for caption.

**Extended Data Fig. 4 | Characteristics of stromal cells in ccRCC.**
(**a**,**b**): Boxplot of (**a**) $T_{ex}$ score calculated by ssGSEA algorithm in TJ-RCC tumors
and (**b**) proportion of $T_{ex}$ in total T cells in snRNA data. P values were calculated by
two-sided Turkey's test. IM1: n = 22, IM2: n = 27, IM3: n = 36, IM4: n = 15. In boxplots,
the central line represents the median value, box limits indicate the interquartile
ranges, and the whiskers extend to 1.5 times the interquartile range. (**c**): Heatmap
showing difference in pathway scores calculated by ssGSEA per cell between
different macrophage sub-clusters. Shown are t-values from a lineal model of
Limma via comparing selected subgroups to all the other samples. (**d**): Pearson's
correlation between specific subpopulations. P-values are from two-sided
Student's *t* test. Cell abundance was estimated by ssGSEA score in each sample
of TJ-RCC cohort. The red line shows the linear interpolation of the data. X-axes
and Y-axes represent ssGSEA score of specific cell types. (**e**): Boxplot of fibroblast

score calculated by ssGSEA algorithm in TJ-RCC tumors. P values were
calculated by two-sided Tukey's test. IM1: n = 22, IM2: n = 27, IM3: n = 36, IM4:
n = 15. The central line of the box represents the median value, box limits indicate
the interquartile ranges, and the whiskers extend to 1.5 times the interquartile
range. (**f**): Heatmap showing Pearson's correlation between cell abundance in
TJ-RCC. Cell abundance was represented by ssGSEA score calculated using marker
genes from snRNA-seq. (**g**): Ideogram of histological locations of given cell types.
(**h**): IHC staining of DCN and ACTA2 on W5_T. Experiment has been repeated three
times. (**i**): Multi-color fluorescence on W5_T, associated to Fig. 2h. White box
marks a region lack of fibroblast, while yellow box marks a fibroblast enriched
region. Figure 2h magnified the yellow box region. (**j**): Multi-color fluorescence on
W5_T. Experiment has been repeated three times. (**k**): IHC staining on W5_T. GJA5
marked arterial vascular. Experiment has been repeated three times.

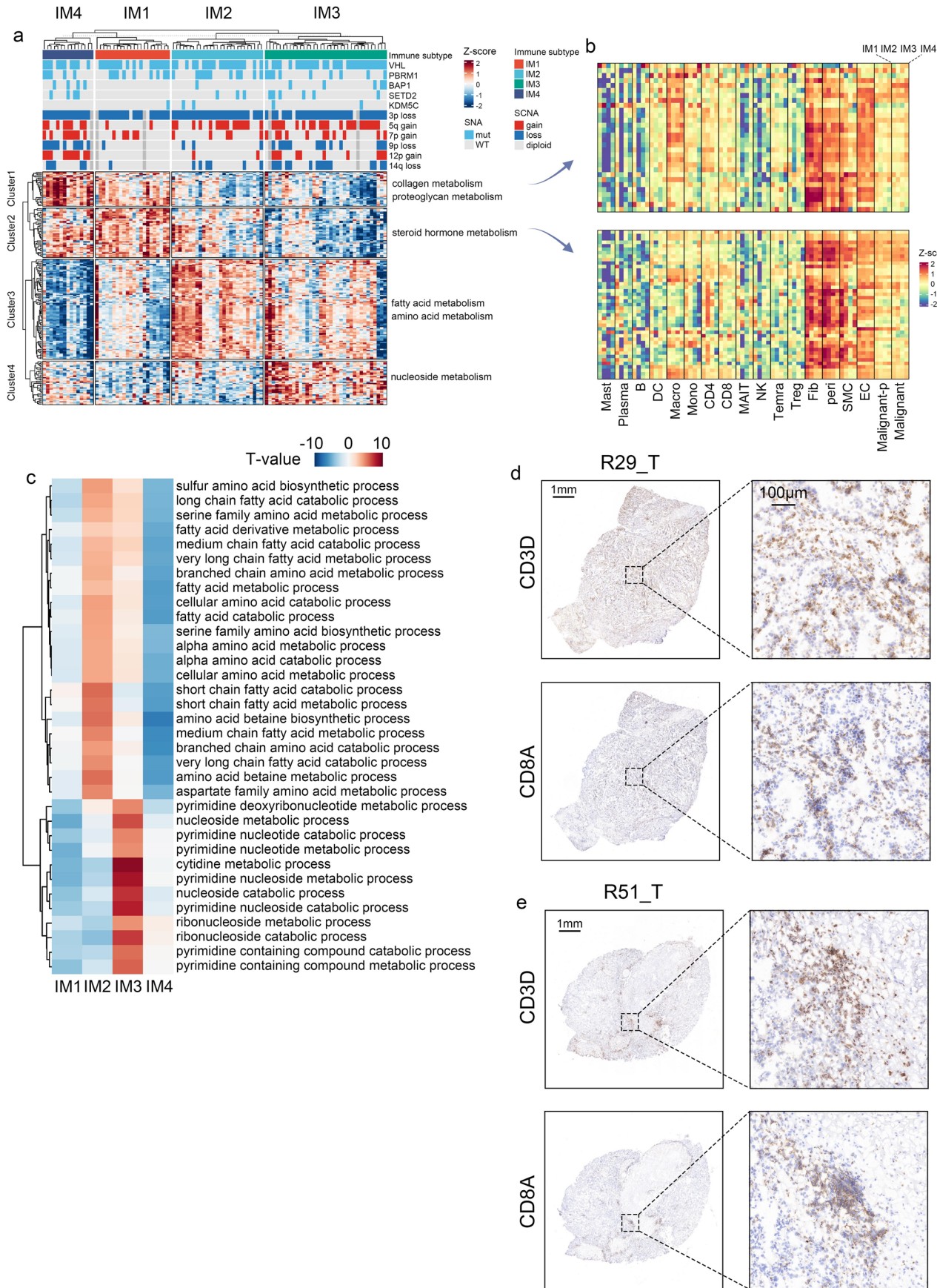

Extended Data Fig. 5 | See next page for caption.

**Extended Data Fig. 5 | Correlation between immune and metabolic heterogeneity.** (**a**): Heatmap showing ssGSEA scores of metabolic related GO terms per sample of TJ-RCC. Pathway scores (ssGSEA) were calculated using TPM matrix of bulk RNA-seq of 100 tumor samples. Differential analysis of ssGSEA scores was performed via comparing a subtype of interest to all other subtypes. 198 out of 1001 metabolic-related GO terms show significant difference between at least one subgroup, and other tumors were extracted for visualization. Z-scores were calculated based on mean value and standard deviation of the ssGSEA scores across all shown samples. 198 GO terms were clustered into 4 different clusters. Key terms frequently occurring within each module were labeled on the right. SNA: somatic single-nucleotide variations; mut: mutated; WT: wild type. (**b**): Heatmap showing ssGSEA scores of metabolic related GO terms in each cell sub-cluster. GO terms are ranked as in **a**. Macro: macrophage; Mono: monocyte; Malignant-p: malignant-proliferating. (**c**): Heatmap showing T values from differential analysis of ssGSEA scores using linear algorithm of Limma. (**d**,**e**): IHC staining of CD3D and CD8A on given tumor samples. Experiment has been repeated three times.

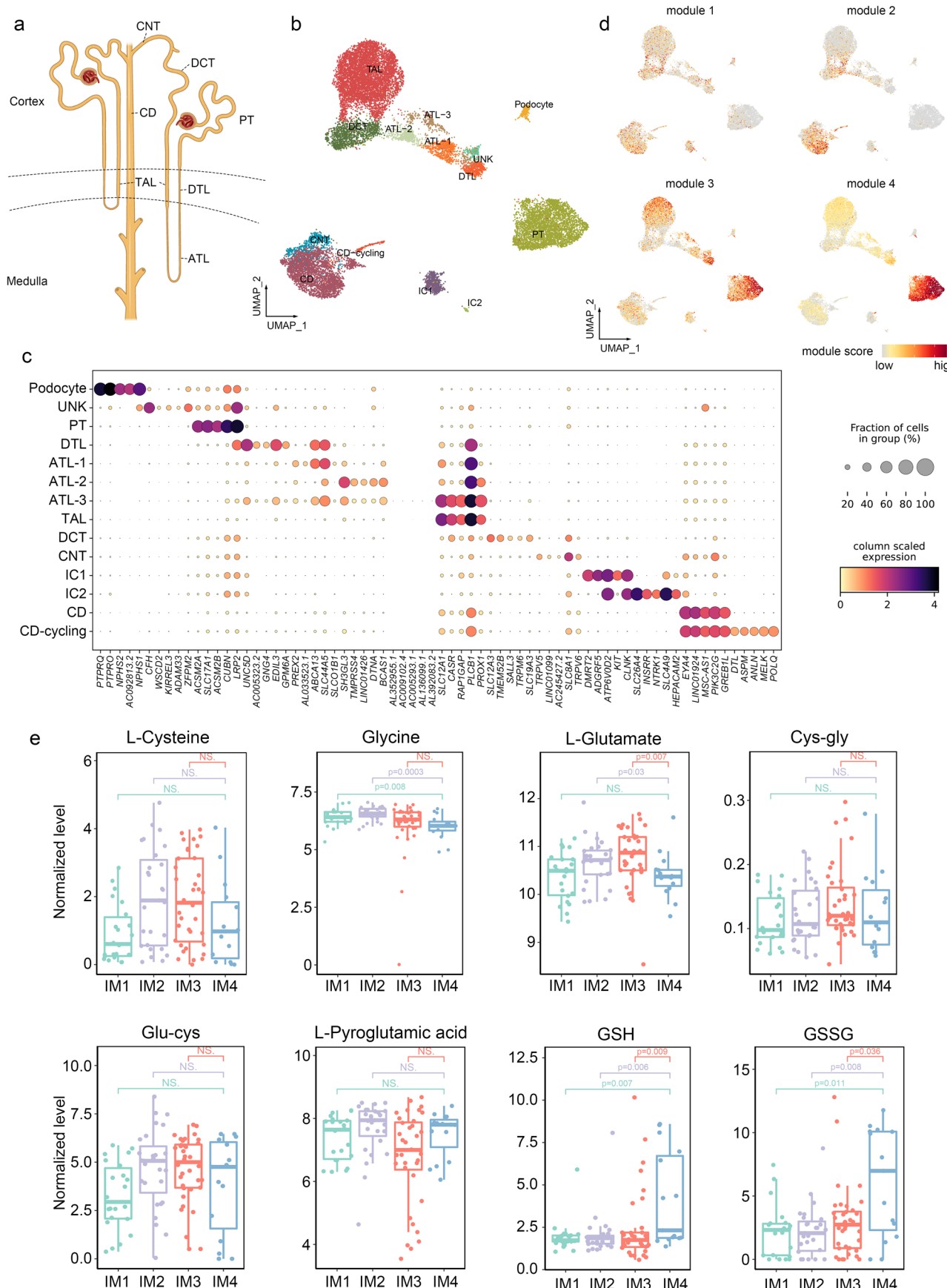

**Extended Data Fig. 6 | See next page for caption.**

**Extended Data Fig. 6 | Expression of IM signatures in normal kidney.**
(**a**): Structure of human normal renal corpuscle, PT: proximal tubular; DTL: descending thin limb; ATL: ascending limb; TAL: thick ascending limb; DCT: distal convoluted tubule; CNT: connecting tubule; CD: collecting duct. (**b**): UMAP plot of renal tubular cells colored by cell subclusters; IC: intercalated cells, Unk: unknow. (**c**): Dot plot showing marker genes of each cell sub-cluster.

(**d**): UMAP plot of tubular cells colored by gene module score calculated by AddModuleScore. (**e**): Boxplots showing the abundance of metabolites related to GSH biosynthesis. P values were calculated by two-sided Turkey's test. IM1: n = 22, IM2: n = 27, IM3: n = 36, IM4: n = 15. For boxplots, the central line represents the median value, box limits indicate the interquartile ranges, and the whiskers extend to 1.5 times the interquartile range.

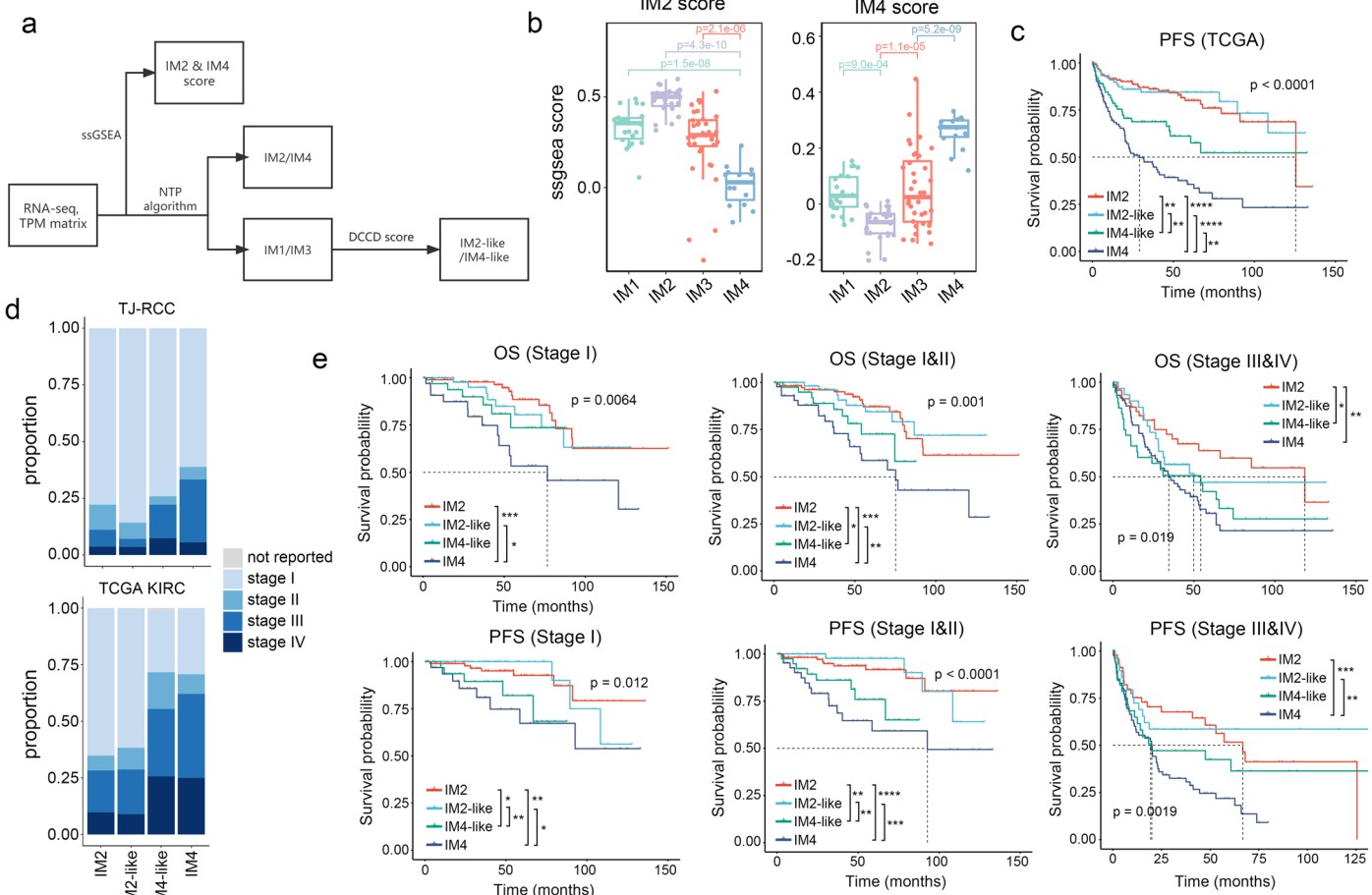

**Extended Data Fig. 7 | IM1 and IM3 ccRCC contain partial IM4 features.**
(**a**): Workflow of the identification of IM2-like and IM4-like ccRCC. (**b**): Boxplots
showing IM2 score and IM4 score in each sample of TJ-RCC. P values were
calculated by two-sided Turkey's test. IM1: n = 22, IM2: n = 27, IM3: n = 36, IM4:
n = 15. For boxplots, the central line represents the median value, box limits indicate
the interquartile ranges, and the whiskers extend to 1.5 times the interquartile
range. (**c**): KM plot of TCGA KIRC grouped by adjusted immune subtypes.
The global p-value was calculated by log-rank algorithm. Pairwise comparison

was performed when global p-value < 0.05. Square brackets show the result
of pairwise comparison. ****: adj.p.value < 0.0001, **: adj.p.value < 0.01.
(**d**): Contributions of each tumor stage in IM2, IM2-like, IM4-like and IM4 groups
in TJ-RCC (top) and TCGA KIRC (bottom). (**e**): KM plot of TCGA KIRC grouped
by adjusted immune subtypes. The global p-value was calculated by log-rank
algorithm. Pairwise comparison was performed when global p-value < 0.05.
Square brackets show the result of pairwise comparison. ****: adj.p.value < 0.0001,
***: adj.p.value < 0.001, **: adj.p.value < 0.01, *: adj.p.value < 0.05.

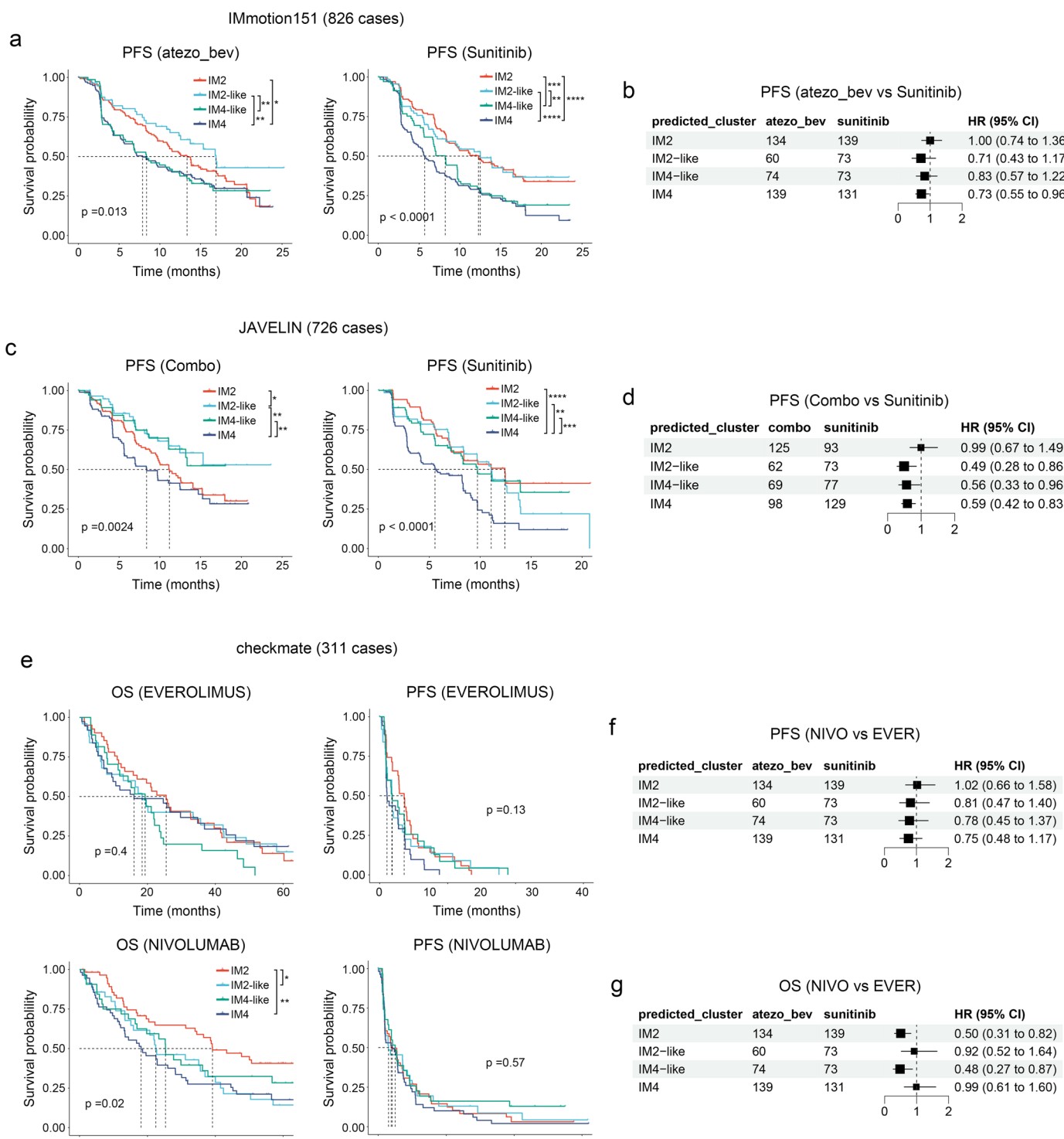

**Extended Data Fig. 8 | The role of adjusted IM classification in clinical cohorts.** (**a**,**c**,**e**): KM plot of immune subtypes in given cohorts (**a**) IMmoton 151, (**c**): JAVELIN, (**e**) CheckMate. The global p-values were calculated by log-rank algorithm. Pairwise comparison was performed when global p-value < 0.05. P values of pairwise comparison were corrected by BH algorithm. ****: adj.p.value < 0.0001, ***: adj.p.value < 0.001, **: adj.p.value < 0.01, *: adj.p.value < 0.05.

atezo_bev: atezolizumab plus bevacizumab; combo: avelumab plus axitinib. (**b**,**d**,**f**,**g**): Forest plot for hazard ratios in patients treated with combined therapy versus sunitinib; (**b**) IMmoton 151, (**d**): JAVELIN, (**f**,**g**): CheckMate. The numbers next to the forest plot indicated number of participants in each group. Centers for the error bars represent the hazard ratio and error bands represent 95% confidence intervals of hazard ratios.

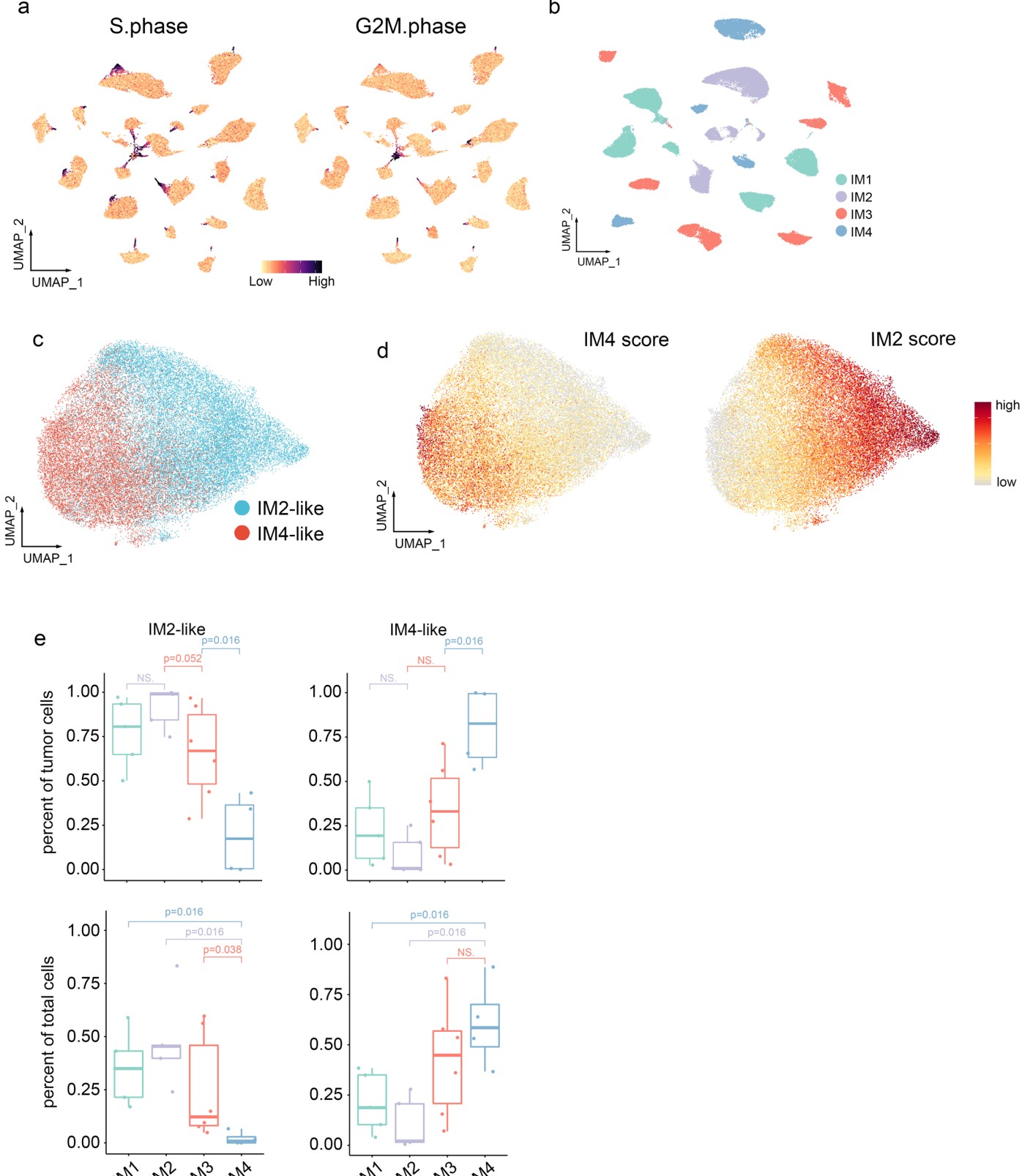

**Extended Data Fig. 9 | DCCD score mirrors proportion of IM4-like cells in malignant cells.** (**a**): UMAP of malignant cells in snRNA-seq colored by S phage and G2M phase scores. (**b**): UMAP of malignant cells in snRNA-seq colored by immune subtypes. (**c,d**): Harmony adjusted UMAP of malignant cells in snRNA-seq colored by (**c**) IM2-like and IM4-like identities; (**d**) IM4 (left) and IM2 scores (right). (**e**): Boxplots showing the comparison of the proportion of IM2-like and IM4-like cells in malignant cells (top) and total cells (bottom) in each immune subtype. P values were calculated by two-sided Tukey's test. IM1: n = 5, IM2: n = 5, IM3: n = 6, IM4: n = 4. For boxplots, the central line represents the median value, box limits indicate the interquartile ranges, and the whiskers extend to 1.5 times the interquartile range.

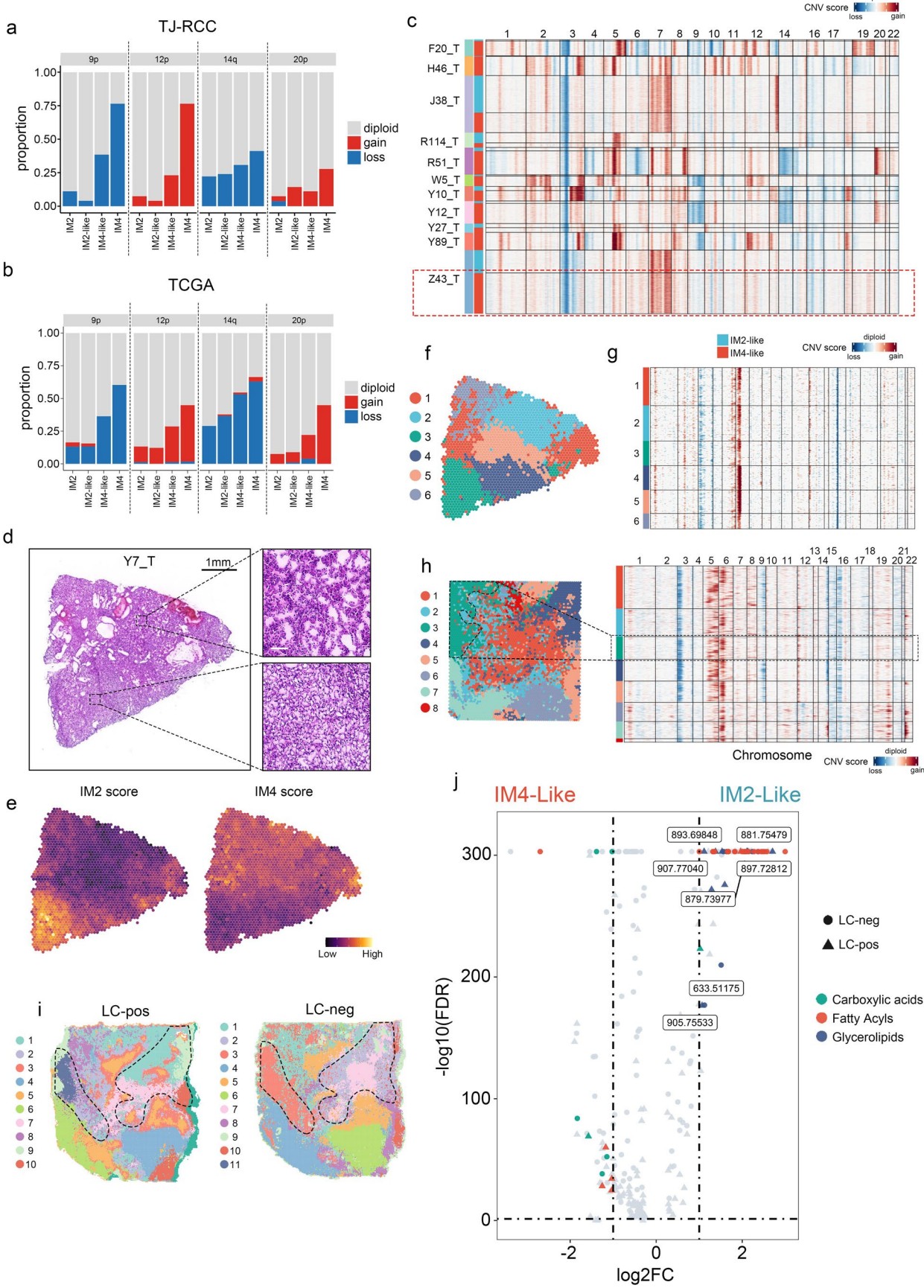

**Extended Data Fig. 10 | See next page for caption.**

**Extended Data Fig. 10 | Sub-cluster level shift from non-DCCD to DCCD.**
(**a**): Frequency of arm-level SCNAs in IM2, IM2-like, IM4-like and IM4 tumors in TCGA KIRC. (**b**): Frequency of arm-level SCNAs in IM2, IM2-like, IM4-like and IM4 tumors in TJ-RCC. (**c**): Frequency of arm-level SCNAs in snRNA-seq data of TJ-RCC. IM4-like cells of Z43_T had higher chr12 gain and chr19 gain signature than IM2-like cells. The dashed red box marked Z43_T with subclonal SCNA. (**d**): H&E staining of Y7_T. The slice is used for 10X visium profiling. (**e**): Spatial signature scores in visium data of Y7_T. (**f**): Visium captured dots of Y7_T, colored by Seurat clusters. (**g**): Visualization of spatial SCNA of Y7_T, grouped by Seurat clusters. (**h**): Seurat cluster of spatial transcriptome data of X98_T (left) and SCNA in each dot of the visium slice grouped by Seurat cluster (numbered 1–8) (right). IM2-like region contains diploid chr 9. (**i**): Spatial dot plot of metabolomics of X98_T, colored by clusters. LC-pos: positive mode LC-MS; LC-neg: negative mode LC-MS. (**j**): Differential metabolites between IM2-like and IM4-like regions in X98_T. Metabolites belonging to carboxylic acids, fatty acyls and glycerollipids were colored according to the metabolite class. The m/z values were labeled to differentially distributed fatty acyls.

# Reporting Summary

## Statistics

For all statistical analyses, confirm that the following items are present in the figure legend, table legend, main text, or Methods section.

| n/a | Confirmed | |
|---|---|---|
| ☐ | ☒ | The exact sample size (*n*) for each experimental group/condition, given as a discrete number and unit of measurement |
| ☐ | ☒ | A statement on whether measurements were taken from distinct samples or whether the same sample was measured repeatedly |
| ☐ | ☒ | The statistical test(s) used AND whether they are one- or two-sided<br>*Only common tests should be described solely by name; describe more complex techniques in the Methods section.* |
| ☐ | ☒ | A description of all covariates tested |
| ☐ | ☒ | A description of any assumptions or corrections, such as tests of normality and adjustment for multiple comparisons |
| ☐ | ☒ | A full description of the statistical parameters including central tendency (e.g. means) or other basic estimates (e.g. regression coefficient) AND variation (e.g. standard deviation) or associated estimates of uncertainty (e.g. confidence intervals) |
| ☐ | ☒ | For null hypothesis testing, the test statistic (e.g. *F*, *t*, *r*) with confidence intervals, effect sizes, degrees of freedom and *P* value noted<br>*Give P values as exact values whenever suitable.* |
| ☒ | ☐ | For Bayesian analysis, information on the choice of priors and Markov chain Monte Carlo settings |
| ☒ | ☐ | For hierarchical and complex designs, identification of the appropriate level for tests and full reporting of outcomes |
| ☐ | ☒ | Estimates of effect sizes (e.g. Cohen's *d*, Pearson's *r*), indicating how they were calculated |

*Our web collection on statistics for biologists contains articles on many of the points above.*

## Software and code

Policy information about availability of computer code

| Data collection | No software was used. |
|---|---|
| Data analysis | Software used include: fastp(v0.20.1),python (v3.7.12), CNVkit (v0.9.10), R (v4.1.2), GISTIC2.0 (v2.0.23), CellRanger sotware (v6.1.2), CellRanger-arc (V2.0.0), SpaceRanger (v2.0.0),Seurat (v4.0.1), sigminer (v2.1.3), edgeR (v3.36.0), ArchR (v1.0.2), ANNOVAR, GATK (v4.2.5.0), sentieon (v202010.04), HISAT2 (v2.1.0), FeatureCounts (v2.0.1), NormalyzerDE (v1.12.0), sva (v3.42.0), Seurat (V4.0.5), Harmony (v0.1.0), COSG (v0.9.0), ComplexHeatmap (v2.16.0), multiOmicsViz (v1.18.0), MACS2 (v2.2.7.1), chromVAR (v1.16.0), SPATA2 (v0.1.0), Monocle (v2.22.0), GSVA (v1.42.0), Monocle 3 (V1.0.0), scMEGA (v0.2.0), clusterProfiler (v4.2.0), ConsensusClusterPlus (v1.58.0), Limma (v3.50.0) and MSiReader (v1.02), scanpy (v1.9.1), destiny (v3.8.1), Proteome Discoverer 2.2 (PD 2.2, Thermo), MS-DIAL, Metaboanalyst 5.0. |

For manuscripts utilizing custom algorithms or software that are central to the research but not yet described in published literature, software must be made available to editors and reviewers. We strongly encourage code deposition in a community repository (e.g. GitHub). See the Nature Portfolio guidelines for submitting code & software for further information.

## Data

Policy information about <u>availability of data</u>

All manuscripts must include a <u>data availability statement</u>. This statement should provide the following information, where applicable:

- Accession codes, unique identifiers, or web links for publicly available datasets
- A description of any restrictions on data availability
- For clinical datasets or third party data, please ensure that the statement adheres to our <u>policy</u>

Raw sequencing data has been uploaded to the GSA-Human database under accession code PRJCA014547 (https://ngdc.cncb.ac.cn/bioproject/browse/ PRJCA014547) but a DAC approval is necessary due to policy restrictions. All the Processed sequencing data have been uploaded to Zenodo (https://zenodo.org/ record/8063124). Expression matrix of TCGA KIRC along with clinical features was obtained from UCSC Xena (https://xenabrowser.net/datapages/?cohort=GDC% 20TCGA%20Kidney%20Clear%20Cell%20Carcinoma%20(KIRC)&removeHub=https%3A%2F%2Fxena.treehouse.gi.ucsc.edu%3A443). JAVLIN and checkmate datasets were obtained from the supplementary material of the original paper. Data of IMmotion 151 was obtained from EGA database (https://ega-archive.org/studies/ EGAS00001004353) with approval from the DAC. Single cell sequencing data of ccRCC was downloaded from Mendeley (http://dx.doi.org/10.17632/ nc9bc8dn4m.1). FUSCC referred to the RCC cohort profiled by team from Fudan University Shanghai Cancer Center (FUSCC). PKU referred to the RCC cohort profiled by team from Peking University (PKU). WES data of PKU cohort collected under PRJNA596359 (https://www.ncbi.nlm.nih.gov/sra/?term=PRJNA596359) was downloaded from SRA database. Processed WES data of FUSCC was obtained from NODE (https://www.biosino.org/node) under Project ID: OEP000796. COSMIC database (https://cancer.sanger.ac.uk/cosmic) was used to annotate the SBS signatures in WES data.

## Research involving human participants, their data, or biological material

Policy information about studies with <u>human participants or human data</u>. See also policy information about <u>sex, gender (identity/presentation), and sexual orientation</u> and <u>race, ethnicity and racism</u>.

| | |
|---|---|
| Reporting on sex and gender | This cohort contained males (n = 63) and females (n = 37), corresponding to the sex distribution of ccRCC. |
| Reporting on race, ethnicity, or other socially relevant groupings | Race, ethnicity, or other socially relevant information was not involved in this study. |
| Population characteristics | A total of 100 participants, with an age range of 27-84, were included in this study. This cohort contained males (n = 63) and females (n = 37), corresponding to the gender distribution of ccRCC. Baseline population characteristics of patients with ccRCC are detailed in Supplementary Tables 1. |
| Recruitment | Histopathological diagnosis was confirmed by at least two different pathologists per sample and only ccRCC cases were included in this sequencing cohort. Informed consent was obtained prior to tissue acquisition. Our cohort included treatment-naive ccRCC patients underwent surgery at Wuhan Tongji Hospital in Jul 2020 and Apr 2021 without intentional selection. |
| Ethics oversight | Institutional Review Board approval (Tongji Hospital) and informed consent was obtained prior to tissue acquisition and analysis. |

Note that full information on the approval of the study protocol must also be provided in the manuscript.

# Field-specific reporting

Please select the one below that is the best fit for your research. If you are not sure, read the appropriate sections before making your selection.

☒ Life sciences          ☐ Behavioural & social sciences          ☐ Ecological, evolutionary & environmental sciences

For a reference copy of the document with all sections, see nature.com/documents/nr-reporting-summary-flat.pdf

# Life sciences study design

All studies must disclose on these points even when the disclosure is negative.

| | |
|---|---|
| Sample size | Clinical characteristics are summarized in Table S1. No statistical methods were used to predetermine sample size. 100 tumors with paired adjacent normal tissues (NATs) were used in whole exon sequencing. Sample size of whole transcriptome sequencing, global proteomics, non-target metabolomics were 100 tumors and 50 NATs. The quantification of sample sizes employed in these multi-omics analyses was based on existing norms within the discipline, as established by parallel investigations in the realm of solid tumor multi-omics research (PMID: 33577785, 34534465, 33212010).<br>20 out of 100 tumor samples were selected for single nucleic transcriptome (n=10) or 10X multiome (n=10) sequencing based on molecular subtypes. Furthermore, 10 out of these 20 tumors were randomly selected for spatial transcriptome (n=10) and spatial metabolome (n=10) profiles. A paired normal renal cortex and medulla were sequenced by single nucleic transcriptome, spatial transcriptome and spatial metabolome and was used as a normal control. |
| Data exclusions | WES data of 5 NATs failed to pass the quality control and no more tissue was available to sequence once again. Hence, paired-tumor samples of these 5 NATs were not involved in SNA and SCNA calling. Instead, we applied GATK germline mutation calling workflow and only select 50 |

most frequent SNAs occured in the other 95 tumors to stat the total mutation rate.

| | |
|---|---|
| Replication | The reported findings were replicated across multiple biological samples. Oil red O staining, IHC and immunofluorescent imaging were performed on 20 different tumor samples and replicated 3 times on each sample. No other experiment was involved. |
| Randomization | Because all treatment-naive ccRCC patients underwent surgery at Wuhan Tongji Hospital between Jul 2020 and Apr 2021 were involved, acquisition of primary patient tumor samples was not randomized. Samples were randomized by case and control status during RNA isolation or library preparation. Tumor samples involved in single nuclei sequencing and spatial sequencing were randomly selected from 100 cases. |
| Blinding | Blinding of the tissue was not possible. All analyses were performed in an automated manner across conditions. |

# Reporting for specific materials, systems and methods

We require information from authors about some types of materials, experimental systems and methods used in many studies. Here, indicate whether each material, system or method listed is relevant to your study. If you are not sure if a list item applies to your research, read the appropriate section before selecting a response.

## Materials & experimental systems

| n/a | Involved in the study |
|---|---|
| ☐ | ☒ Antibodies |
| ☒ | ☐ Eukaryotic cell lines |
| ☒ | ☐ Palaeontology and archaeology |
| ☒ | ☐ Animals and other organisms |
| ☒ | ☐ Clinical data |
| ☒ | ☐ Dual use research of concern |
| ☒ | ☐ Plants |

## Methods

| n/a | Involved in the study |
|---|---|
| ☒ | ☐ ChIP-seq |
| ☒ | ☐ Flow cytometry |
| ☒ | ☐ MRI-based neuroimaging |

## Antibodies

| | |
|---|---|
| Antibodies used | Anti-Human CD8 (1:100, Abcam, Cat# ab178089)<br>Anti-Human CD31 (1:2000, Proteintech, Cat#:11265-1-AP)<br>Anti-Human DCN (1:2000, Abcam, Cat# ab277636)<br>Anti-Human CD3 (1:1000, Proteintech, Cat# 17617-1-AP)<br>Anti-Human CD163 (1:2000, Proteintech, Cat# 16646-1-AP)<br>Anti-Human a-SMA/ACTA2 (1:2500, Proteintech, Cat# 14395-1-AP)<br>Anti-Human PDGFRA (1:500, Abcam, Cat# ab203491)<br>Anti-Human F13A1 (1:100, Abcam, Cat# ab76105)<br>Anti-Human ACKR1 (1:200, SAB, Cat# 56458)<br>Anti-Human CX40 (1:500, BIOSS, Cat# bs-1050R)<br>Goat Anti-Rabbit IgG H&L (HRP) (1:2000, Abcam, Cat# ab205718)<br>Goat Anti-Rabbit IgG H&L (Cy3 ®) preadsorbed (1:200, Abcam, Cat# ab6939)<br>Goat Anti-Rabbit IgG H&L (Alexa Fluor® 488) (1:500, Abcam, Cat# ab150077)<br>Goat Anti-Rabbit IgG H&L (Alexa Fluor® 594) (1:500, Abcam, Cat# ab150080)<br>Goat Anti-Rabbit IgG H&L (Cy5 ®) preadsorbed (1:500, Abcam, Cat# ab6564) |
| Validation | All antibodies used in this study are commercially available. They are validated by the vendors for the specific assay and species used. The validation is available on the vendors website.<br>Anti-Human CD8: https://www.abcam.cn/products/primary-antibodies/cd8-alpha-antibody-sp239-ab178089.html<br>Anti-Human CD31: https://www.ptgcn.com/Products/PECAM1-Antibody-11265-1-AP.htm#product-information<br>Anti-Human DCN: https://www.abcam.cn/products/primary-antibodies/decorin-antibody-epr24097-105-ab277636.html<br>Anti-Human CD3: https://www.ptgcn.com/products/CD3E-Antibody-17617-1-AP.htm<br>Anti-Human CD163: https://www.ptgcn.com/products/CD163-Antibody-16646-1-AP.htm<br>Anti-Human a-SMA/ACTA2: https://www.ptgcn.com/products/ACTA2-Antibody-14395-1-AP.htm<br>Anti-Human PDGFRA: https://www.abcam.cn/products/primary-antibodies/pdgfr-alpha-antibody-epr22059-270-ab203491.html<br>Anti-Human F13A1: https://www.abcam.cn/products/primary-antibodies/factor-xiiia-antibody-ep3372-ab76105.html<br>Anti-Human ACKR1: https://www.sabbiotech.com.cn/g-320124-DARC-Rabbit-mAb-56458.html<br>Anti-Human CX40: http://www.bioss.com.cn/SpeNew01.asp?id=258&pro37=1&pro33=101&guige01=50ul<br>Goat Anti-Rabbit IgG H&L (HRP): https://www.abcam.cn/products/secondary-antibodies/goat-rabbit-igg-hl-hrp-ab205718.html<br>Goat Anti-Rabbit IgG H&L (Cy3 ®) preadsorbed: https://www.abcam.cn/products/secondary-antibodies/goat-rabbit-igg-hl-cy3--preadsorbed-ab6939.html<br>Goat Anti-Rabbit IgG H&L (Alexa Fluor® 488): https://www.abcam.cn/products/secondary-antibodies/goat-rabbit-igg-hl-alexa-fluor-488-ab150077.html<br>Goat Anti-Rabbit IgG H&L (Alexa Fluor® 594): https://www.abcam.cn/products/secondary-antibodies/goat-rabbit-igg-hl-alexa-fluor-594-ab150080.html<br>Goat Anti-Rabbit IgG H&L (Cy5 ®) preadsorbed: https://www.abcam.cn/products/secondary-antibodies/goat-rabbit-igg-hl-cy5--preadsorbed-ab6564.html<br>Goat Anti-Rabbit IgG H&L (Cy3 ®) preadsorbed: https://www.abcam.cn/products/secondary-antibodies/goat-rabbit-igg-hl-cy3--preadsorbed-ab6939.html |

