## [Peer Review File · Nature Genetics]

Peer Review Information

Manuscript Title: Multi-omic profiling of clear cell renal cell carcinoma identifies metabolic reprogramming associated with disease progression

Corresponding author name(s): Professor Ke Chen, Professor Shancheng Ren, Professor Wei Guan, Professor Zhen Tao

Reviewer Comments & Decisions:

Decision Letter, initial version:

14th Apr 2023

Dear Professor Chen,

Your Article, "Intergrated Multi-omics Sequencing Reveals Metabolic Reprograms in the Progression of ccRCC" has now been seen by 3 referees. You will see from their comments copied below that while they find your work of considerable potential interest, they have raised quite substantial concerns that must be addressed. In light of these comments, we cannot accept the manuscript for publication, but would be very interested in considering a revised version that addresses these serious concerns.

We hope you will find the referees' comments useful as you decide how to proceed. If you wish to submit a substantially revised manuscript, please bear in mind that we will be reluctant to approach the referees again in the absence of major revisions.

If you choose to revise your manuscript taking into account all reviewer and editor comments, please highlight all changes in the manuscript text file. At this stage we will need you to upload a copy of the manuscript in MS Word .docx or similar editable format.

*2) If you have not done so already please begin to revise your manuscript so that it conforms to our Article format instructions, available [here](http://www.nature.com/ng/authors/article_types/index.html). Refer also to any guidelines provided in this letter.

[redacted]

If you wish to submit a suitably revised manuscript we would hope to receive it within 6 months. If you cannot send it within this time, please let us know. We will be happy to consider your revision so long as nothing similar has been accepted for publication at Nature Genetics or published elsewhere. Should your manuscript be substantially delayed without notifying us in advance and your article is eventually published, the received date would be that of the revised, not the original, version.

Thank you for the opportunity to review your work.

Sincerely,

Safia Danovi
Editor
Nature Genetics

Referee expertise:

Referee #1: RCC multi-omics

Referee #2: RCC genomics, clinical

Referee #3: RCC proteomics, metabolomics

Reviewers' Comments:

Reviewer #1:

Remarks to the Author:

Integrated Multi-omics Sequencing Reveals Metabolic Reprograms in the Progression of ccRCC

The authors introduce a molecular classification for clear cell Renal Carcinoma (ccRCC) that succeeds in separating patients with high and low survival rates in four independent sources, i.e. TCGA Kidney Renal Clear Cell Carcinoma (KIRC) as well as three different clinical studies. Using an initial classification signature obtained from single cells in the TME to cluster their own 100 bulk transcriptomic ccRCC datasets, the authors generated a final signature of 856 genes. Their final signature could describe four different IM subtypes. The authors could document continuous progression from the low-risk IM2 to the high-risk IM4 subtype across different samples and within the same sample as analyzed by spatial omic technologies. Consequently, they reassigned sample classification of IM1 and IM3 subtypes based on their similarity to the IM2 and IM4 subtypes. Pathway enrichment analysis documents a shift from fat catabolism to transmembrane nutrient uptake in the high-risk IM4 subtype. Morphological analysis documents loss of lipid droplets characteristic for ccRCC, in support of the predicted shift in energy source. These observations made the authors rename their IM4 subtype in the De-clear Cell Differentiated (DCCD) Subtype as the high-risk end stage of the ccRCC. Transcriptional profiling could document expression changes of known transcription factors associated with subtype progression.

This reviewer thinks that the study is of high interest for the readers of Nature Genetics and can have a high impact on the field of kidney cancer biology. It identifies a classification of clinical relevance and gives new mechanistic insight underlying disease progression. The authors generate and integrate multiple datasets, i.e. genomic, bulk transcriptomic, metabolomics and proteomic data, spatial transcriptomics and metabolomics, single nucleus and snATAC RNAseq as well as morphological analysis. To validate the relevance of their new classification, the authors use four independent published datasets.

Major comments

Utilized datasets

The authors should clearly state which datasets they generated and which were used for which analysis to facilitate an easier understanding of their manuscript. It is also very important to add the 100 + 50 bulk transcriptomic, proteomic and metabolomics datasets to the flow chart in figure 1A. This way the reader knows that most of the following analyzes are based on bulk omic datasets, before the results obtained by spatial and single nucleus omic datasets are discussed in support of the

conclusions drawn from the bulk omics data. It should also be mentioned in the results section that the 50 NAT samples were derived from the same kidneys as 50 of the 100 tumor samples.

Extension of initial IM signature

The authors should explicitly state in the results section that the 911 protein coding genes that were used to identify the four IM subtypes in the bulk transcriptomic data from their own 100 tumor samples were replaced by a new set of 856 signature genes. This information should not be hidden in the methods section. It should also be explicitly stated that the new signature is influenced by gene expression of the malignant cells and the TME that add up to bulk transcriptomic profiles, in contrast to the original signature. This is an important information to understand why the proportion of IM2- and IM4-like malignant cells vaguely follows the overall tumor subtype. Here, it would be very interesting how much of the new signature is still dominated by the TME or by the malignant cells themselves (see below).

Algorithm for new signature

The new signature is applied to independent datasets. This reviewer assumes that the enrichment for each IM subtype specific signatures was calculated using the ssGSEA algorithm that was also used for pathway predictions (as described for the IM2-like and IM4-like signatures). Please explain in the methods section and briefly add to the results section.

Immunological differences between Immune Subtypes 1 and 2

Since the IM1-4 classification is introduced as an immune subtype classifier, it seems to be of particular interest, how subtypes IM1 and IM2 differ from each other based on immunological criteria. The authors document using bulk and single nucleus RNAseq of their own samples that these two subtypes both lack immune cell infiltration and had similar abundancies of resident immune cells.

IM subtype classification of AA ccRCC subgroup

The authors reason the search for a new classification of ccRCC with the observation that progression and development of AA-induced ccRCCs that constitute 43% of their tumor samples is not linked to the known mutations involved in ccRCC progression and development, since mutations in the related genes are underrepresented in the AA subgroup. Nevertheless, they never describe how the new tumor classification relates to the AA group. This could be presented in the supplemental information. Contradictory conclusions about the association of CD8 infiltration and survival

Since metabolic programming seems to have a large influence on the TME, the authors grouped their samples into four clusters based on the expression of selected metabolic pathways curated from Gene Ontology (Suppl. Figure 4A). The authors identify that cluster 4 is mainly related to nucleoside metabolism and allows separation of IM3 from all other subtypes. They linked this observation to CD8+ Tcell infiltration using morphological analysis. In the TCGA KIRC dataset, increased pyrimidine/cytidine metabolism correlates with increased CD8+ T-cell infiltration and poor prognosis, a finding that this is consistent with their findings in the TJ-RCC cohort (Figure 3C/D). Here, it seems to be important to point out that pyrimidine metabolism correlates with a poor prognosis and independently of that with effector T-cell infiltration (or CD8+ T-cells as this reviewer assumes, please use the same term in the figure). CD8+ T-cell infiltration is a characteristic of the IM3 subtype that has a similar survival rate than the IM1 subtype and a better survival rate than the IM4 subtype. In a previous section, the authors comment on this observation that their results suggest that "T cell infiltration may limit tumor progression in ccRCC" (line 198), which seems to contradict the conclusions implied by this paragraph. As far as this reviewer understands it, a low survival probability correlates with pyrimidine metabolism that correlates with T-cell infiltration that correlates with the

IM3 subtype. Stringing together these three pairwise correlations does not allow concluding that a low survival probability correlates with the IM3 subtype, as documented by the authors' data. The authors should clearly point out that such a conclusion is not valid and disproved by the direct comparison of the survival rates in CD8+ infiltrated IM3 subtypes, as discussed in a previous section. They should also discuss, which potential mechanisms could explain these observations. Maybe, it is the failure of sufficient CD8+ recruitment by increased pyrimidine metabolism that determines the poor outcome? Could it be linked to exhausted T-cells? Are exhausted T-cells part of the CD8+ T-cell population, as line 213 implies? How do pyrimidine metabolism and CD8+ T-cell infiltration correlate with the IM2-like/IM4-like sub classification?

Algorithm for the assignment of IM1-4 or IM2-like/IM4-like subtypes

The authors should briefly mention in the results section that the IM2 and IM4 are ssgsea enrichment scores for IM2 and IM4 marker genes. This documents that the same methodology that was used to predict pathways and cell type compositions was also used for the tumor subclassification. This reviewer also assumes that the same algorithm was used to group samples into IM1-4 subtypes, i.e. bulk transcriptomic profiles were subjected to ssgsea enrichment analysis using the four gene signatures, followed by scale centering of each score across all tumor samples. Searching for the highest scale centered score allowed sample subtype classification. This approach would ensure that IM4 and IM2 samples would be classified as IM4-like and IM2-like, respectively. Please describe the algorithm in the method section and briefly in the results section.

Survival analysis using IM2- and IM4-like subtype classification

The authors replace the initial categorization of samples into IM1 and IM3 subtypes by a new classification, i.e. IM2- and IM4-like. They developed an IM2 and an IM4 score and used the difference of both scale centered scores for reclassifying tumors initially classified as IM1 and IM3 tumors into IM2- or IM4-like tumors.

The new classification assigned different samples into the same groups and does not correlate with the initial IM1 and IM3 classification. The authors document a gradual decrease in the survival rate from IM2/IM2-like over IM4-like to IM4 tumors, using the TCGA KIRC data. To document, if the re-categorization of the IM1 and IM3 samples increased the predictive strength, the authors should briefly compare the results to the Kaplan Meier curve for the initial IM1-4 classification that they reference in an earlier section of their manuscript (Suppl. Figure 2F).

Before introduction of the new classification, the authors document how the initial IM1-IM4 classification succeeds in predicting survival rates in three chemotherapeutic studies and can help with the decision of drug treatment schemes for each patient. The authors finally prefer the IM2, IM4, IM2-like and IM4-like classification and show that it improves the survival prediction, using the TCGA KIRC patients. It should be of high interest for the reader, if a similar improvement can be observed for the three clinical studies as well. The analysis of the clinical studies based on the new classification should technically become part of the main manuscript, while the results from the initial classification should be part of the supplemental information.

Kaplan Mayer comparisons based on tumor stages

To see if the new initial and/or final classification outperforms the traditional one, all Kaplan Mayer analyses should be generated after grouping the patients based on the currently used clinical classification (probably pTNM, or only without consideration of N and M criteria for a fair comparison).

Correlation between IM2/IM4 scores in malignant cells and bulk transcriptomic IM subtype

The IM subtype gene expression signature was initially developed as a TME signature using marker genes obtained by single cell RNAseq of only non-malignant cell types. Applying this signature to 100 bulk transcriptomic samples lead to the identification of a second signature set, that now is also

influenced by the malignant cells that contribute to the bulk expression values. This overall signature is then used to classify malignant cells obtained from the same samples into IM2-like and IM4-like malignant cells. The authors document that the relative malignant cell subtype proportions vaguely follow the subtype classifications annotated based on the bulk transcriptomic data. Would it not be more reasonable to compare the cell counts of the IM2- and IM4-like malignant cells relative to all cells in each single nucleus RNAseq dataset, since these should give a better estimation of how much gene expression in malignant cells contributes to bulk expression?

Statistical analysis of spatial transcriptomics and metabolomics including all samples

The authors present results of spatial transcriptomic and metabolomics analyses in figures 3G, 3H, 3J, 3L, 3K, 3M, 6J, 6M and 6N. In the mentioned figures, they document in two different samples that spatial hypoxanthine and guanine abundancies correlate with spatial cytotoxic T-cell infiltration within the same sample. Using one of those samples, they document that the activity for the GO pathway 'pyrimidine metabolic process' correlates with the areas expressing a T-cell marker gene and hypoxanthine expression. Although the authors subjected 14 samples to spatial transcriptomic and metabolomics analysis (Figure 1A), they only show the results for two samples. How does the correlation look for the other 12 samples? All three compared entities, i.e. pyrimidine metabolism activity, CD8A, hypoxanthine and guanine expression are quantitatively described for each hexagon. It should be possible to calculate the correlation for each sample and then show a beeswarm or similar plot for all samples. This should be applicable at least for CD8A expression and pathway activity, since they were both determined from the same dataset. If non-overlapping (hexagon) fields between spatial transcriptomic and metabolomics do not allow such a comparison, the authors should clearly state this. If samples have to be excluded, because they do not show transition from IM2-like to IM4-like areas, the authors should state this as well.

Gene, protein, metabolite and pathway expression metrics

It is generally difficult to follow the different value types that are submitted to the different analyses and are used for the visualization. It is also not clear, if the data is calculated and presented using the same algorithms and value types across the different figures.

Z-score transformed gene expression

Figures 2A, 4A, 4H, 5A, 5D, 5E, 6B, 7C, 7E, 7G all show gene expression values after z-score transformation. Although it looks like the documented values were generated by the same algorithm for all different datasets, there are some things that seem to contradict this conclusion. This reviewer assumes, that the Z-scores were calculated by scaling the expression of each gene over all documented tumor samples, as figure legends 5A/D/G suggest. Scaling means, subtracting the mean from each sample and dividing by the standard deviation.

Nevertheless, Figure 2A and the legend of figure 2A use the term differentially expressed genes, which normally labels gene sets that are significantly different between two conditions, e.g. each immune subtype and NAT.

Figure 4A does not only show z-scores for the four immune subtypes, but also for the NATs. All other figures show only z-scores across the tumor samples. This raises the question if the Z-scores in figure 4A are different from the ones shown in figure 2A, since they now also consider the gene expression in the NAT samples. Were the NAT samples used for all Z-score calculations, but excluded from the visualization in the other figures? This reviewer thinks that the TCGA KIRC samples and the clinical studies do not contain NAT samples that could be included.

The legend to figure 4A states that this figure shows the clustering results of up- and down-regulated genes in DCCD, i.e. IM4 tumors. To the understanding of this reviewer, the figure shows the clustering results of the row z-score distribution across all IM subtype and NAT samples, using only those genes

that were up- or downregulated in IM4 samples vs all other IM subtypes. Please clarify.

Figure 4H shows row z-scores for selected transporter genes, but this time the gene expression of all samples has been somehow collapsed on each immune subtype and NAT. This reviewer could not find a description of the algorithm that generated these values. For consistency, the z-score values in this figure should be shown in all samples of each immune subtype after ordering them by immune subtype (as in figure 2A). Since this figure also contains NAT samples, it is again not clear to this reviewer, if the Z-scores were calculated on a different basis (i.e., under consideration of NAT samples) than in figures 2A, 5A, 5D and 5E.

Differentially expressed genes, proteins and metabolites in tumor versus NAT

For a more general overview of tumor-associated gene expression, the authors document differentially expressed genes, proteins and metabolites in all tumor samples versus NAT (Figures 1I). They use their results to document generally consistent results obtained by the three omic technologies for selected pathway activities. Similarly, they document how differentially expressed genes, proteins and metabolites between IM4 and the other three IM subtypes agree and affect selected metabolic pathway activities. This is regular data visualization strategy and is well suited to support the authors conclusions.

Z-score transformed metabolite expression

Similar questions arise when investigating the documented metabolite expression values in figure 3E and 4F. Metabolite expression values are presented as Z-scores over all immune subtypes and NAT after calculation of some mean values, and not for each individual sample. Again, the question arises, why did the authors not stay with their standard visualization and how were the subtype selective values calculated. Were the NAT samples included in the Z-score calculation, in contrast to what the gene expression Z-scores shown in figures 2A, 5A/D/E imply?

Pathway enrichment analysis

Figures 1G, 1H, 2A, 3A, 3F, 3M, 4B, 4D, 4E, 4I, 7C, 7P show results of pathway enrichment analysis of transcriptomic, proteomic and metabolomics datasets. To the understanding of this reviewer, the authors seem to use multiple strategies for pathway enrichment analysis.

Pathway analysis of differentially expressed genes

Figure 2A shows pathway enrichment results of the genes part of the different clusters. This reviewer assumes that all genes that are grouped together by the clustering algorithm were subjected to pathway enrichment analysis, using clusterProfiler, if understood correctly by this reviewer.

Z-score transformed ssGSEA scores

Single sample GSEA (ssGSEA), as implemented in GSVA package, was used to analyze gene expression values in each sample, followed by row Z-score transformation of the ssGSEA scores across all IM subtype samples, ignoring NAT samples. This reviewer thinks that this strategy was used for the blocks labeled 'ssgsea (RNA)' and 'pathways' in figure 2A and for figure 3A. This enrichment strategy generally matches the strategy to show gene expression values. This reviewer assumes that the authors used Z-score transformed ssGSEA scores to allow easier visualization of the distribution of pathway activities over the different samples. ssGSEA values probably range over different scales for different pathways, making the clustering and visualization of differential pathway activities in the same heatmap difficult. The authors should state in the methods section, why they subjected the ssGSEA scores to Z-score transformation (and do the same for genes).

Nevertheless, the authors should always state the used methodology in the figure captions and label the dendrogram in each figure with the same label that documents the input scores for the Z-score

calculation and the dataset used for enrichment. Labeling one block with 'ssgsea (RNA)' and another with 'pathways' is misleading, since it suggests different analysis strategies. Use of the term single sample GSEA facilitates and easier understanding, since not every reader might be familiar with ssgsea.

Linear model to calculate differential pathway activities using ssGSEA scores

The authors subjected ssGSEA scores to differential pathway expression analysis using a linear model implemented in limma. This was applied for figure 3B, that compares pathway predictions between transcriptomic and proteomic datasets. This strategy is briefly outlined in the method section (line 1125) and should be explained in more detail. It is not clear to this reviewer, which samples were compared with each other? Each tumor sample versus all NAT samples? Each tumor sample versus its matching NAT sample, if available (and possible with n=1)? What do the gray dots represent? Why are they so many? Do they visualize other pathways? Why are there only a few dots with the same color? Why do the numbers of dots with the same color differ between the colors?

In addition, the method section implies that it was not used for GO, as in figure 3B, since it explicitly names the gene lists that were used for analysis. This is misleading.

Pathway activity

The label 'Pathway activity' implies a 3rd strategy for pathway enrichment analysis, though this reviewer assumes that it is the ssGSEA score (figures 3D, 3F, 3M). The method section implies this conclusion in line 1127. If so, please replace all pathway activity labels by ssGSEA.

Differential abundance score for metabolites

In figure 1H the authors calculate the DA score for metabolic pathways that considers the difference between up- and down-regulated metabolites of the same pathway. The authors correctly argue, that this penalizes identification of pathways with contradicting expression changes

MetaboAnalyst pathway enrichment

In figures 4D and E, the authors use MetaboAnalyst to identify pathway with over or underrepresented activities in the IM4 subtype.

Different methods to describe pathway activities

The application of different methods for pathway enrichment analysis of transcriptomic, proteomic and metabolomics datasets could give the impression to the reader that presented results depend on the selected method.

In figure 2A two different approaches of pathway enrichment analysis were used: subjection of cluster genes to pathway enrichment analysis using clusterProfiler and sample specific GSEA. Two of the functions described by the pathways identified by each analysis show functional overlap ('ECM organization' + 'collagen-binding' vs 'collagen containing extracellular matrix' and 'sprouting angiogenesis' vs 'angiogenesis'). Gene and pathway z-scores in this module show similar trends. The other shown pathways do not overlap. One cannot conclude from the pathway enrichment analysis results of the cluster specific genes that an IM subtype of interest has a higher expression of a particular pathway. It can only be concluded that genes involved in a particular pathway cluster together across the different tumor samples. Genes mapping to the first module are up-regulated in IM1 and IM4, but the analysis does not allow a conclusion if those genes are the ones mapping to the pathways on the left side. Were the identified pathways also identified as up- or downregulated in a particular IM subtype based on ssGSEA analysis, followed by limma analysis? This analysis would allow the conclusions that the results section reports. If so, the enrichment results shown next to the DEGs cluster can be removed. Please update figure 2A accordingly or correct your conclusions (line 167- line182).

In contrast, the enrichment analysis of the module genes in figure 4B is legitimate, since these genes

were identified as differentially expressed in IM4 vs IM1-3, as long as these pathways are not linked to activities in NAT or non-IM4 subtypes.

Similarly, the enrichment of cluster genes in suppl. Figure 4A is legitimate, since the results section describes relationships between clusters and pathways.

As another example, the authors document correlation between differential pathway ssGSEA between tumor and NAT samples, as obtained for proteomic and transcriptomic datasets using limma in figure 3B. It would be advantageous to use the ssGSEA scores here as well. If use of the original ssGSEA scores does not allow documentation of a correlation, the authors should justify their approach in the methods section and briefly the results section.

Correlating pathway ssGSEA scores and T-effector cell type abundancies (figure 3D) seems to be a reasonable approach. T effector cell abundancies describe values that are obtained independently for each sample, i.e. these values are not normalized with regard to the abundancies in other samples, as Z-scores and differential pathway activities are. Consequently, independent pathway enrichment scores need to be utilized as well, therefore use of the ssGSEA scores. If so, the authors should describe this line of reasoning for the used pathway enrichment score in the method section and maybe briefly mention it in the results section.

The same accounts for the cross-comparison of pathway activities across all IM subtypes in figure 3F. If the documented pathway activities are ssGSEA scores, it is more reasonable to use the ssGSEA scores instead of the derived Z-score transformed values for statistical tests.

Regarding metabolite enrichment analysis, the reader might also wonder, why were differentially expressed metabolites between tumor and NAT subjected to DA analysis and differentially expressed metabolites between IM4 and all other subtypes or NAT subjected to MetaboAnalyst pathway enrichment analysis.

Generally, it seems important to this reviewer to introduce the different metrics to describe pathway activities and restrict them to a minimum of different approaches. If the 'pathway activity' score is the ssGSEA, it would make sense to describe that the basic measure of all pathway enrichment results are the ssGSEA scores. Similar to gene expression values, these were Z-score transformed to allow clustering and visualization of pathways with similar trends over all samples, independently of the scale of change. Otherwise, the reader might get the impression that the different pathway enrichment scores are selected based on favorable results. Axis labels in the figures should also describe the underlying scores using the same terms (e.g. replace pathway activities by ssGSEA or add ssGSEA in brackets, if this is the used algorithm).

Test for differential pathway activities

Any time the authors conclude from their pathway enrichment analysis that a particular pathway is more or less active in an IM subtype, they should state a P/T-value for their conclusion, using the linear model from limma (e.g. lines 167-182 or line 262, lines 288-290).

Differential expression analysis of the transporter genes (figure 4H) should be added as a documentation of significance as well.

Methods section

It is difficult to align the methods section to the results and the figures. It should give a general overview of which datasets were analyzed by which method and with which gene library to obtain which type of values. The method section would profit from flow charts that summarize the analysis steps starting with each raw dataset (e.g., raw read counts) and how one value (e.g. Z-scores) derives from another (e.g. ssGSEA scores). Please add such flow charts to the supplemental information.

Figure captions for main and supplemental figures

The figure captions of all figures are insufficient. They should contain enough information, so that the reader can get a sufficient understanding of the figures without reading any other part of the manuscript. The supplemental figures are missing figure captions. Please add, following the same principal.

Minor comments

Single nucleus RNAseq

The authors show the overall results of single nucleus RNAseq in figures 2E, 2F, 2G, 2H and focus on malignant cells in figures 6A, 6B, 6C, 6D, 6E and 6F. While figure 2E shows one island that contains the malignant cell types, figure 6D shows multiple islands of malignant cell types, one for each analyzed samples. What is the difference between the analyses used to generate both figures, besides the focus on only one cell type in figure 6D? Normally, resubmission of a cluster to the clustering pipeline to identify subclusters does not separated cells that were initially close neighbors from each other.

Please also clearly state that the malignant single cells were only mapped to IM2-like and IM4-like cells, but not to IM2 and IM4 subtypes. In case of the own and TCGA bulk transcriptomic datasets only IM1 and IM3 samples were mapped to IM2-like and IM4-like samples, while at the same time the initial IM2 and IM4 subtype classification was kept. Assuming the same approach for the single malignant cells, the reader might think that the IM2 and IM4 cells were excluded from the analysis shown in figures 7A/B/C, which is inconsistent with their consideration for the TCGA data in figures 7F/G.

Arrangement of malignant cells along pseudotime

The authors document that malignant cells can be arranged along one trajectory that correlates with the IM2-like and IM4-like scores (Figure 7A). It should be clearly stated that the IM2 and IM4 scores shown in figure 7B were calculated based on the gene expression in the malignant cells (and not based on the samples they map to). Consequently, the arrangement of the malignant cells along the pseudotime is dominated by those genes that contribute to the IM2/4 scores.

Pseudotimeline and transcription factor profiling

To identify potential transcription factors that determine the progression from IM2 to IM4 subtypes the authors calculate differentially expressed genes between IM4-like and IM2-like malignant cells and samples. The authors should add that only the malignant cells were used from the single cell RNAseq dataset, as this reviewer assumes from the previous analyses. Please explain in the methods section and explain in the results section why only malignant cells (or all cells) were used for the analysis (e.g. add 'to identify the TF motifs in malignant cells').

Discussion

Since the IM4 subtypes depend on increased nutrient import, could the changed TME support the tumors? Generally, figure 2A suggests upregulation of angiogenesis in IM1 and IM4 subtypes, although endothelial cells were primarily associated with IM1 and IM2.

Origin of the initial IM gene expression signature

The authors re-analyzed published single cell RNAseq data to identify an initial signature of 911 protein-coding genes for initial subtype classification. Here, the authors are missing any description why the initial approach of using these datasets should succeed in the identification of a suitable signature. How many tumor samples were analyzed in this published dataset? Are the analyzed published samples representative of the existing ccRCC spectrum, i.e. do they span a great variety of know tumor stages, molecular and clinical characteristics (i.e. survival rates)? The methods section offers a sufficient description of the underlying methodology that lead to the identification of the

signature and suppl. figure 2A visualizes it graphically. Please state in the results section that these 911 genes are the merged top 60 significant cell type specific marker genes with the highest fold changes. Please also add which cell types were used for this analysis in the results section (i.e. using all cells of the TME, excluding the malignant cells), when referencing suppl. figure 2A.

Clear labeling of analyzed datasets

The authors separate their own samples into the four IM subtypes and characterize cell-type composition, pathway activities and potential cell type interactions of the different subtypes (Figure 2A). Here, the authors should state, that predicted cell type abundancies arise from enrichment analysis of bulk gene expression profiles, to prevent misunderstanding that the described results are based on single cell/nucleus datasets.

Sample numbers

Please always mention sample numbers subjected to each omics assay in the method, results and figure captions.

xCell

How was xCell applied to the data? This reviewer could not find any description in the methods section or a reference.

Abbreviations

Please double check that all abbreviations are introduced by their full terms in the main text (e.g. TCGA KIRC). A supplemental table with general abbreviations, on top of the one with metabolite abbreviations, would be very helpful too.

Line 191: For an easier understanding, it would be helpful, if the survival analysis of the four subtypes in the TCGA data would form a new paragraph.

Line 225: Please replace ontology by pathway or gene list.

Line 251: Using the full term 'Tumor Microenvironment' in the headline might be better, if in agreement with the journal's style

Line 259: 'Cluster 2 was derived from interstitial cells and endothelial cells, but not from malignant cells.' Please explain how the single nucleus data allowed this conclusion.

Line 274: Please specify how many samples were subjected to spatial transcriptomic and metabolomics analysis (figure 1A) and which subtypes these samples belonged to. In particular, how many IM3 samples were analyzed to support the infiltration of colocalization of CD8+ T-cells and gyanine and hypoxanthine metabolism.

Line 352: Here a brief sentence could be added that the downregulation of metabolic energy generation pathways implies alternative energy sources for IM4 tumors. Consequently, the expression of membrane transporters was investigated.

Line 367: Please state that IM4 is renamed into DCCD.

Line 368: Please introduce abbreviation and state that IM4 is DCCD.

Line 380: If a drugs mechanism of action can be linked to identified pathway of a particular IM subtype, it would be helpful to briefly mention it (e.g. axitinib inhibits angiogenesis in line 395).

Line 382: Please state what kind of sequencing data was used (i.e. bulk transcripomic, zscore)

Line 412: Adding the expression low-risk IM2 and high-risk IM4 might better illustrate why IM1 and IM3 subtypes are reclassified.

Line 512: Should it not be AR negative tumor to indicate that AR is downregulated in H46_T relative to the other tumors?

Line 482: Since the clusters are not annotated, it might be better to use the expressions top and bottom clusters instead of cluster 1 and 2.

Line 550: The HIF2A-dependend LD accumulation occurs only in one molecular subtype of ccRCC. How is this conclusion made?

Line 583: Why does the observation that "loss of AR is a continuous, important process of DCCD" indicate that "lymph nodes are the potential colonization site for these micro metastatic niches"?

Line 600: Mentioning of drug targets can facilitate an easier understanding of the underlying biology.

Line 711: Please add that the 50 NAT samples are from the same specimen as 50 of the tumor samples, as this reviewer assumes.

Line 720: Does it mean "enrich target mRNA"?

Line 741: Please write "centrifuged".

Line 802: Were p-values and adjusted p-values considered in the manuscript?

Line 807: Please also describe that bulk transcriptomic data was subjected to ssgsea based pathway enrichment analysis.

Line 812: Please state that ssgsea was also used with other gene libraries.

Line 814: Does the description of the correlation refer to figure 2H looks? That looks more like Pearson correlation.

Line 824: Please add transcription factor profiling to the headline.

Line 827: Please indicate that the IM2-like and IM4-like samples contain all four initial IM subtype classifications, as this reviewer assumes. IM2-like and IM4-like subgroupings were only generated based on the IM1 and IM3 subtypes of the bulk transcriptomic datasets. This might cause confusion, if the IM2 and IM4 subtypes contributed to the identification of transcription factors as well.

Line 916: Please use lower case letters for 'parent' or replace comma by dot.

Line 1089: Maybe, add 'between these regions within the same sample' for more clarity.

Line 1140: Were three cutoffs applied, one p-value, one q-value and one adjusted p-value cutoff?

Figure 1

Figure 1A: Is this figure missing the generation of 100 + 50 bulk transcriptomic and 100 + 50 bulk proteomic datasets?

Figure 1B: The arrangement of the figure legends that explain the colors is initially misleading, since the top figure is separated from its legend by the bottom figure.

Figure 1B: What is the difference of this figure and suppl. figure 1A? Why do the values differ?

Figure 1C: What do the abbreviations FUSCC, PHU and TJRCC stand for?

Figure 1D: Please reference in the figure caption that the colors are the same as in figure 1B.

Figure 1F: The flow chart in figure 1A documents generation of two transcriptomic datasets, i.e. single nucleus RNAseq and Spatial transcriptomics. Figure 1F compares gene expression profiles of each sample. Please add that each dot represents the bulk transcriptomic, proteomic and metabolomics profiles of 100 tumor and 50 NAT samples?

Figure 1G: How were the T-values calculated? The method section describes in the chapter 'Gene set enrichment analysis' that a linear model implemented in limma was used to calculate the difference between ssGSEA pathway scores between tumor and NAT samples. Nevertheless, it does not mention that is used Gene Ontology for this. Were GO biological processes used in this figure? Were redundant pathways removed as a functionality offered by clusterProfiler that is referenced in the next method section 'Pathway enrichment analysis'? How were the shown biological processes selected?

Figure 1H: What do the dashed lines show? What do the numbers show (i.e. total number of up- and down-regulated metabolites in pathway)? Please specify in the figure caption.

Figure 1H: Which metabolite library was used for the analysis?

Figure 1H: Please write differential abundance score or explain the abbreviation DA in the figure caption.

Figure 2

Figure 2A: Adding the four IM subtypes as a headline over the four blocks would enhance faster understanding, in particular since the text in the legend is very small.

Figure 2A: The top block of this figure also contains too much information. Please remove all information from this figure that is not discussed in the text. Detailed information can be shown separately as a supplemental figure.

Figure 2A: How were the DEGs calculated? What is the denominator (all reference samples)? Are the DEGs also visualized as Z-scores, as the figure legend implies?

Figure 2A: How were the Z-scores from the single sample GSEA values calculated?

Figure 2A: Which algorithm was used to calculate pathway Z-scores? Are these Z-scores also based on single sample GSEA?

Figure 2A: A description of how xCell was used is missing in the method section.

Figure 2B: Please reference CNA as Copy number alteration.

Figure 2C: Please reference TMB as Tumor mutational burden.

Figure 2G: What do the Odds ratios compare? Cell type counts in one immune subtype versus all others? Please frame all ORs discussed in the results section.

Figure 2H: The number of dots suggests that this is bulk transcriptomic data. Which values were correlated? Are these Z-scores? If so, why does the figure not correlate ssgsea scores, to be consistent with previous figures. Please specify used value types in the figure caption and/or add to axis labels. Which correlation metrics was used, 'pearson' as implied by the red interpolation line or 'spearman' as mentioned in the methods section (line 814). Please briefly describe what the red line shows.

Figure 3:

Figure 3A: Please remove all information from the figure top that is not discussed in the results section.

Figure 3A: Please label the row axis of the heatmap using the same label as used in figure 2A, if, as this reviewer assumes, the same algorithm was used.

Figure 3A: Please add that this analysis is based on the bulk transcriptomic profiles, as this reviewer assumes.

Figure 3A: Though it seems to be obvious, but the explanation that Z-scores were calculated based on mean and standard deviation of the ssGSEA scores across all shown samples should be added to the figure caption for proper documentation.

Figure 3B: "T-values were calculated by a linear model of Limma." Please explain in detail. Which values were used as input, ssGSEA values that were probably also used for Figure 3A? Please also add a section to the methods.

Figure 3B: How do the pathways 'cytidine', 'nucleoside' and 'pyrimidine' relate to the GO pathways shown in A? Please at least refer to the full GO names in the figure caption.

Figure 3C: Please specify which data was used to distinguish between high and low pyrimidine metabolism (e.g., bulk transcriptomic data).

Figure 3C: Please specify which values were used to separate samples into high and low pyrimidine metabolic activity. ssGSEA scores/Z-scores from figure 3A or T-values from figure 3B?

Figure 3C: Please use the full expression 'Kaplan-Meier Plot' in the figure legend.

Figure 3D: Please specify which data (e.g., bulk transcriptomic) and which value types were used for this analysis? Please add the score (e.g. ssGSEA) to the axis.

Figure 3D: Please specify in which unit the T effector abundance is shown. Are these relative cell type counts?

Figure 3D: Please specify which correlation was used ('pearson?').

Figure 3D: Please briefly describe that the red line shows the linear interpolation of the data in the figure caption.

Figure 3E: Please specify that this figure shows the results from bulk metabolomics analysis, as this reviewer assumes.

Figure 3E: Why does this figure not show the metabolite expression in the individual samples, as figure 3A does? How were the Z-scores calculated?

Figure 3F: Which score is shown for the pathway activity? Does the figure show ssGSEA scores?

Figure 3G: Please add the dataset shown in this figure. Is it spatial transcriptomics, as this reviewer assumes?

Figure 3G: Please add the subtype of the sample to this figure.

Figure 3G: To which figure does the scale bar below the second image belong to? It does not seem to represent the colors in this figure.

Figure 3G: It would be helpful to add "Cytotoxic T-cell" to the label CD8A as well.

Figure 3H: Which sample is shown for this slide (probably R29_T)? Please add sample information to the figure.

Figure 3H: What do the two figures in the lower row show? Is this the background noise?

Figure 3I: Here, it would also be helpful to add the cell types that are labeled by both marker proteins as well.

Figure 3I: Please write Immunohistochemistry staining in the figure caption.

Figure 3I: Please add the subtype of the sample to this figure.

Figure 3J-L: Please write spatial transcriptomics.

Figure 3M: How are the pathway activities calculated? Which data is used? Do the densities show the ssGSEA score calculated for the gene expression in each hexagon on the spatial transcriptomics slide?

Figure 4

Figure 4A: Please remove all information that is not mentioned in the results section from the top figure block. Increasing the height of the bars that show the tumor subtypes can help as well.

Figure 4B: A more common visualization of the enrichment results is to show the $-\log_{10}(\text{FDR})$ values as bars and color the bars based on the number of genes (or simply write the number of genes into each bar).

Figure 4C: It would be helpful for the reader, if the figure contains the information that Oil Red O stains lipid droplets or lipids.

Figure 4D: It would be advantageous to either show $-\log_{10}(\text{p-values})$ or $-\log_{10}(\text{FDR})$ in both figures 4B and 4D.

Figure 4F: Why does the figure only contain one value for each metabolite in each IM subgroup? Same comment as in figure 3F.

Figure 4H: Why does the figure not show gene expression values in all samples as all previous figures? Can the overexpression of transporters in IM4 be supported by a statistical test?

Figure 4I: Which data was subjected to pathway enrichment analysis? What do the circle sizes visualize? Does a negative T-value imply downregulation? Were all samples mapping to one tumor subtype or NAT compared with all others?

Figure 4K: Does this figure show TCGA KIRC data? Please add to figure legend and results section. Which comparisons do the p-values refer to? This could be visualized using curly brackets connecting the compared subtypes.

Figure 5

Figures 5A/D/G: The figure caption says "Expression were normalized into row z-score", as this reviewer assumed for figures 2A and 4A as well (although the legend to figure 2A says that DEGs are shown).

Figure 5B/E/H: Which comparisons do the p-values refer to?

Figure 6

Figure 6A: What is the IM4 score? How were the genes for the score selected?

Figure 6B: It should be stated that the DCCD score and IM4 score are two different labels for the same values, as this reviewer assumes.

Figure 6B: Adding the IM2-like and IM4-like labels as headlines to the figure would be helpful.

Figure 6C: It should be explicitly stated, that IM2- and IM4-like are defined by a positive or negative IM4-IM2 difference, as figure 6B implies.

Figure 6C: Which comparison does the p-value refer to? Are all other comparisons not significant, as this reviewer assumes?

Figure 6D: Which data was used for this UMAP, the same single nucleus RNAseq data shown in figure 2E? Why do the cells cluster by sample and not by cell type? Does the figure show only one cell type, i.e. only the malignant cells shown in figure 2E, as figure 6F implies? Was the data reanalyzed under different parameters, using only the malignant cells?

Figure 6M: What do the numbers how?

Figure 6N: It would be good to show the same spectra in figure 6M and 6N as well as to round the m/z spectra to the same number of decimals as in figure 6M.

Figure 7

Figure 7B: Please explain the colors in the figure legend.

Figure 7C: Please summarize in the figure legend how the TFs were identified (e.g. differentially expressed TFs between IM4-like and IM2-like malignant cells and bulk transcriptomic samples).

Figure 7C: Which TF-target gene database was used for the analysis?

Figure 7D: Which sample is shown? Please also state that this is based on ssGSEA scores calculated from spatial transcriptomics data.

Figure 7E: Are these values also Z-scores? Why does the legend distinguish between high and low instead of showing Z-scores?

Figure 7F: What are DC1, 2 and 3?

Figure 7F: What do the arrows show? Why are they colored in blue (IM2) and red (IM1)?

Figure 7C/E/G: Since these figures all show Z-score normalized expression values of selected transcription factors (assumed by this reviewer), they should all be colored using the same color scheme, i.e. the one used in figure 7G that also agrees with the color scheme's used for the gene expression values in previous figures.

Figure 7I: Please increase the text size. There is still space in the figure.

Figure 7J: Please add the sample names instead of the numbers to the figure.

Reviewer #2:

Remarks to the Author:

The manuscript Intergrated Multi-omics Sequencing Reveals Metabolic Reprograms in the Progression of ccRCC by Hu et al provides a detailed multiomic analysis clear cell renal cell carcinoma. Although the entire study population is large (100 patients), only a subset of these went on to have a true multiomic examination. Having said that, this is the first paper that has combined all of these approaches. As a consequence, the dataset is unique and in itself valuable. The authors should be commended on this.

The analysis also appears sound in general, with the conclusions plausible and important. As with all large scale projects, it is impossible for reviewers to fully check the veracity of the conclusions. However, there are no obvious flaws in the approach.

The paper is in general well written and presented. There are numerous typographical errors which I

shall not comprehensively discuss.

I have some specific questions which need addressing prior to publication, but otherwise feel that this paper is a welcome and significant addition to the field.

- 1) Line 37 - define TJ-RCC
- 2) Line 42 - 'extremely' how is this defined?
- 3) Line 77 - 'mutation hot spots' - these are no hot spots per se, as they are all in tumour suppressor genes. Perhaps it would be more accurate to say enriched in driver genes.
- 4) Line 79 - 'finding indicates that data obtained from clinical cohorts from different countries are necessary for understanding the heterogeneity of ccRCC'. Heterogeneity can mean many things, but for this I presume you are meaning geographically related intertumoural heterogeneity. Most RCC researchers use heterogeneity to mean intra-tumoural het.
- 5) Line 92 - 'hot spot' again, another use of hot spot which is confusing.
- 6) Line 108 - Table S1. It would be valuable to include the precise mutation in this table, eg genomic location, base change, etc.
- 7) Line 117 - COSMIC needs referencing
- 8) Line 121 - 'In contrast, TTN, MUC16, MTOR, XIRP2, etc., were more frequently mutated in the AA group'. This doesn't mean anything in the context of an enhanced mutation rate due to AA exposure. Can you run dN/dS to see whether there might be additional driver events in this cohort?
- 9) Line 124 - can you define which 'mutation hub genes'. I presume this is established drivers, but you need to specify which you have included, rather than list a handful.
- 10) Line 125 - Please apply some statistics to test differences in these cohorts and consider using something other than a pie chart. For one, the values have been normalised which means that we lose data on the absolute values.
- 11) Line 127 - This is interesting and perplexing:
 - 1) Many of the drivers ?"hub genes" are indels, can the authors reassure that these have not inadvertently been ascribed a mutational context.
 - 2) One interpretation of the data is that AA mutagenesis occurs once the tumours have formed with their respective genetic drivers. However, once tumours have formed, they would no longer act as a tubule or filtration unit of the kidney, and they would therefore presumably no longer be exposed to high levels of AA. How do the authors explain this dichotomy?
- 12) Line 148 LD needs defining
- 13) Line 161 - There is only a single dataset referenced here. Which are the others that were used - how did you select them?
- 14) Paragraph beginning line 167: This entire section needs careful rewording as often cells are described as being rare or enriched. As there is no spatial RNA/ protein or morphology described here, the language needs toning down. One could say that RNA signals are present that might infer these cell types.
- 15) Line 203: How did you chose these 20 samples?
- 16) Line 209: 'Subcluster analysis revealed that the distribution of immune cell clusters strongly discriminated across immune subtypes'. Evidence? stats?
- 17) Line 227: Where is the evidence and stats that fibroblasts are enriched in this cluster?
- 18) Line 229: Did you look at other associations, for instance ACKR1 expressing endothelial cells and macrophage subsets for instance?
- 19) Line 260 'In addition, that difference is dependent on stromal cell abundance and is weakly affected by molecular subtypes'. I don't see the evidence for this in the figures.
- 20) Line 434 I agree that the plot supports this, but Is there not a nicer way to plot this information?

In this case, the location in the UMAP means nothing as most variance can be attributed to batch variance.

21) Line 444: 75% seems high to me rather than 'only'

22) Line 447: where is the subclonal SCNA - it is not clear on this figure? If it subtle, how would you know that the expression differences are not a consequence of the IM2 to IM4 gradient rather than SCNA?

Reviewer #3:

Remarks to the Author:

In this paper, the authors generated a multi-omic dataset from samples of a cohort of 100 ccRCC patients. They profiled the multi-omic samples, focusing on potential exposure to aristolochic acid and comparing tumor vs normal adjacent tissue. Then they defined a cell composition gene signature of ccRCC from a public single cell ccRCC dataset and used it to cluster the patients of their cohort into 4 groups. They further characterized the defining features of these 4 groups, at a pathway and clinical level. They determined that 2 of these subgroups, IM2 and IM4, were particularly associated with a cellular phenotype they coined De-clear cell differentiated cells (DCCD). They evaluated the relevance of this cellular phenotype with respect to various treatment strategies, and characterized some of the patient samples in more depth with single nuc RNA and visium slices. Finally, they studied the pseudotime progression of this cellular phenotype across patient samples.

Overall, this study provides an analysis of a large amount of multi-omic data on ccRCC samples, especially regarding Asian ccRCC patients. These can be important for the understanding of ccRC and ultimately lead to treatment decisions that would rely more on molecular profiles of patients. It is however not our expertise, so we can not make authoritative assessments of the relevance of the findings.

Further, we believe that there are several major points that are important to address.

Major comments

1) While extracting a gene signature representative of cell composition is a very interesting idea to cluster ccRCC patient samples, it is held back by the potential quality of this signature. Since almost the entire study relies on the definition of this signature, we would expect to see more evidence regarding the quality of this signature (rely heavily on Harmony performance and the subsequent cell type definition).

2) It would be interesting to see how these findings would compare to results obtained with a gene signature obtained not by a union of the top marker genes of each cell type, but instead the top DEG (healthy vs tumor) for each cell type.

3) The findings of this study are also heavily dependent on the division of the patient cohort into 4 subgroups. While the authors show that the choice of this number isn't entirely arbitrary, it still isn't clear how this number was the only obvious choice. If it was not, then the authors should provide insights into the robustness of their conclusion if this number would change.

4) Same this as 4), but regarding the choice of 911 genes for the signature, based on the top 60 for each cell type. What happens if a number other than top 60 is chosen ?

5) Instead of defining the IM2-like and IM4 like signatures (which are supposed to represent low and high DCCD), it would improve the clarity of the study a lot to instead define a single signature directly representative of the DCCD phenotype that can be used to score samples.

- 6) The integrated metabolic analysis feels very arbitrary with respect to the enzymatic pathways that were highlighted. For example, why isn't branched chain amino acid metabolism considered, even though it is among the most deregulated metabolic pathways highlighted in this study and many others on ccRCC?.
- 7) Make the code available for reviewers to check for clarity and reproducibility. The provided repository only contains a small subset of scripts.
- 8) Make the data easier accessible for reviewers and eventually readers. the Mendeley link does not work, even after registering into Mendely with an account. We suggest to use rather a repository like Zenodo.
- 9) There are other large multi-omic characterizations of ccRCC cohorts, and this study should be place in the context of those, finding compared, etc.

Minor comments

- 1) An enrichment score does not represent activity per se. The authors often refer to a metabolic pathway enrichment score as a metabolic pathway activity. This is a semantic shortcut that changes the meaning and can be misleading.
- 2) Figure S2B: why were those markers chosen, provide more insights into which markers validate which previous claim.
- 3) It isn't clear why the authors claim that the urea cycle disorder is not dependent on the dysregulation of the enzymes.
- 4) The comparison between the different survivability curves for different treatment combinations should be made clearer, regarding the figures. It's hard to compare them by looking at them side by side. A statistical test should be made to formally assess the differences.
- 5) Figure 6H: the piece with a high IM2 score is very small on the slice.
- 6) Line 469: Please clarify in which section it is detailed.
- 7) Line 475: I don't understand what the authors mean by motifs here.
- 8) "Cell type identification was performed based on markers from single cell atlas of human kidney" Shouldn't authors instead use human cancer kidney markers? from cohort authors poicked 20 for snRNA and 10 for snATAC - why those number,s and how were those chosen?
- 10) SCENIC regulons are normally built in a bespoke manner for each context, so using the ones in the server are not the best ("downloaded a TF list containing all the already known human TFs from github 807 (<https://github.com/aertslab/SCENIC>).") Furthermore, now there is the chance to use ATAC + RNA to build regulons with SCENIC+ and other tools of other groups.

Author Rebuttal to Initial comments

Reviewer #1:

Major comments

Utilized datasets

The authors should clearly state which datasets they generated and which were used for which analysis to facilitate an easier understanding of their manuscript. It is also very important to add the 100 + 50 bulk transcriptomic, proteomic and metabolomics datasets to the flow chart in figure 1A. This way the reader knows that most of the following analyzes are based on bulk omic datasets, before the results obtained by spatial and single nucleus omic datasets are discussed in support of the conclusions drawn from the bulk omics data. It should also be mentioned in the results section that the 50 NAT samples were derived from the same kidneys as 50 of the 100 tumor samples.

Response: We agree with the reviewer that a modified Figure 1A could make it easier for the readers to understand the major findings of this paper. The new version has been displayed in the revised manuscript.

The reviewer suggested that “the 50 NAT samples were derived from the same kidneys as 50 of the 100 tumor samples” should be mentioned in the results section. We agree with the reviewer’s opinion. It has been detailed at the beginning of the results section (Line 105-106).

Extension of initial IM signature

The authors should explicitly state in the results section that the 911 protein coding genes that were used to identify the four IM subtypes in the bulk transcriptomic data from their own 100 tumor samples were replaced by a new set of 856 signature genes. This information should not be hidden in the methods section. It should also be explicitly stated that the new signature is influenced by gene expression of the malignant cells and the TME that add up to bulk transcriptomic

profiles, in contrast to the original signature. This is an important information to understand why the proportion of IM2- and IM4-like malignant cells vaguely follows the overall tumor subtype. Here, it would be very interesting how much of the new signature is still dominated by the TME or by the malignant cells themselves (see below).

Response: Thanks for the reviewer's suggestion. We have detailed the extraction process of this 856-gene signature in the methods (Line 1261-1267) and explicitly pointed out that extension of the initial IM signature was based on a new gene set instead of the initial 911 protein-coding genes (Line 1256-1267).

Algorithm for new signature

The new signature is applied to independent datasets. This reviewer assumes that the enrichment for each IM subtype specific signatures was calculated using the ssGSEA algorithm that was also used for pathway predictions (as described for the IM2-like and IM4-like signatures). Please explain in the methods section and briefly add to the results section.

Response: Thanks for the reviewer's suggestion. The IM2-like and IM4-like signatures were calculated by the ssGSEA algorithm respectively. We have detailed this process in the methods (Line 1324-1327) and briefly mentioned it in the results section (Line 453).

Immunological differences between Immune Subtypes 1 and 2

Since the IM1-4 classification is introduced as an immune subtype classifier, it seems to be of particular interest, how subtypes IM1 and IM2 differ from each other based on immunological criteria. The authors document using bulk and single nucleus RNAseq of their own samples that these two subtypes both lack immune cell infiltration and had similar abundancies of resident immune cells.

Response: The reviewer asked ‘how subtypes IM1 and IM2 differ from each other based on immunological criteria’. In this paper, IM clusters were classified using the whole TME-based signature. It not only contains immune cells but also contains stromal cells, including fibroblasts, smooth muscle cells, pericytes and endothelial cells. Compared to the IM2 group, the IM1 group have higher fibroblast and smooth muscle cell scores (**Figure 2A**), and it was verified by IHC staining (**Extended Data Fig. 2H**). Since then, although both IM1 and IM2 lack immune infiltration, they were still classified into two distinct groups.

IM subtype classification of AA ccRCC subgroup

The authors reason the search for a new classification of ccRCC with the observation that progression and development of AA-induced ccRCCs that constitute 43% of their tumor samples is not linked to the known mutations involved in ccRCC progression and development, since mutations in the related genes are underrepresented in the AA subgroup. Nevertheless, they never describe how the new tumor classification relates to the AA group. This could be presented in the supplemental information.

Response: We appreciate the reviewer for pointing out this important analysis. We are sorry for the loss of supplementary figure legends in the original manuscript. 43 AA samples investigated

in this work are derived from 3 different Chinese cohorts (FUSCC, PKU and TJ-RCC), and only 11 of these tumors are from the cohort profiled by our team (TJ-RCC cohort). No matter in which cohort, only about 10% ccRCC samples contain AA signature.

The reviewer suggested that the relationship between the new classification and AA groups should be presented in the supplementary information. We agree with the reviewer's suggestion and have presented a Sankey plot showing the relationship between AA subgroup and IM clusters in the revised supplementary figure (**Extended Data Fig. 2G**). AA feature is not related to IM subgroups.

Contradictory conclusions about the association of CD8 infiltration and survival

Since metabolic programming seems to have a large influence on the TME, the authors grouped their samples into four clusters based on the expression of selected metabolic pathways curated from Gene Ontology (Suppl. Figure 4A). The authors identify that cluster 4 is mainly related to nucleoside metabolism and allows separation of IM3 from all other subtypes. They linked this observation to CD8+ Tcell infiltration using morphological analysis. In the TCGA KIRC dataset, increased pyrimidine/cytidine metabolism correlates with increased CD8+ T-cell infiltration and poor prognosis, a finding that this is consistent with their findings in the TJ-RCC cohort (Figure 3C/D). Here, it seems to be important to point out that pyrimidine metabolism correlates with a poor prognosis and independently of that with effector T-cell infiltration (or CD8+ T-cells as this reviewer assumes, please use the same term in the figure). CD8+ T-cell infiltration is a characteristic of the IM3 subtype that has a similar survival rate than the IM1 subtype and a better survival rate than the IM4 subtype. In a previous section, the authors comment on this observation that their results suggest that "T cell infiltration may limit tumor progression in ccRCC

” (line 198), which seems to contradict the conclusions implied by this paragraph. As far as this reviewer understands it, a low survival probability correlates with pyrimidine metabolism that correlates with T-cell infiltration that correlates with the IM3 subtype. Stringing together these three pairwise correlations does not allow concluding that a low survival probability correlates with the IM3 subtype, as documented by the authors’ data. The authors should clearly point out that such a conclusion is not valid and disproved by the direct comparison of the survival rates in CD8+ infiltrated IM3 subtypes, as discussed in a previous section. They should also discuss, which potential mechanisms could explain these observations. Maybe, it is the failure of sufficient CD8+ recruitment by increased pyrimidine metabolism that determines the poor outcome? Could it be linked to exhausted T-cells? Are exhausted T-cells part of the CD8+ T-cell population, as line 213 implies? How do pyrimidine metabolism and CD8+ T-cell infiltration correlate with the IM2-like/IM4-like sub classification?

Response: We thank the reviewer for the comments. The reviewers pointed out a contradiction in our conclusion regarding the infiltration of CD8+ T cells, pyrimidine metabolism and prognosis correlation. We believe this was due to a misunderstanding caused by unclear descriptions, which we have now corrected. To investigate the correlation between metabolic reprogramming and immune reprogramming, we selected metabolic-related GO terms that showed differences between IM subtypes. The cluster mentioned in lines 278-301 referred to modules of GO terms (row clustering, **Extended Data Fig. 5A**), not sample clustering. Cluster 4 of GO terms is highly enriched in the IM3 group, that’s why the scores of GO terms related to pyrimidine metabolism are positively correlated to CD8+ T cell abundance in both TJ-RCC and TCGA KIRC. Previously, several independent works have reported that higher CD8+ infiltration is related to poor prognosis in ccRCC^{1,2}. Therefore, we found that pyrimidine metabolism is

associated with poor prognosis, which is consistent with previous findings. This is because ccRCCs infiltrated by CD8⁺ T cells usually have poor prognostic DNA events such as CNV or SNV events including 14q loss, 9q loss and/or BAP1 mutations, rather than tumor progression caused by T cells themselves. In this study, we found that IM3 and IM4 contained similar levels of advanced tumors (stage III & IV, **Figure 2D & Extended Data Fig. 3C**), most of which included the aforementioned poor prognostic DNA events. Under this premise, the prognosis of IM3 was significantly better than that of IM4, while IM3 had more tumor-infiltrating T cells compared to IM4. Therefore, in the results section, we documented that this better prognosis may be due to the restriction on tumor progression by CD8⁺ T cell infiltration (Line 214-216). This is a comparison between IM3 and IM4 rather than the entire ccRCC population. In fact, among all the IM subtypes, IM4 was ranked second in terms of T cell infiltration (**Response Figure 1A, Figure 2A & Extended Data Fig. 3H-3I**); thus at an overall level CD8⁺ T cell infiltration still strongly correlates with poor prognosis.

The reviewer asked “Could it be linked to exhausted T-cells? Are exhausted T-cells part of the CD8⁺ T-cell population, as line 213 implies?”. Through snRNA-seq, we observed a more widespread T cell exhaustion in IM3 (**Extended Data Fig. 3H-3I**). At the same time, in spatial transcriptomics, we also found that the regions with higher pyrimidine metabolism signals contained higher T cell exhaustion signals (**Response Figure 1B, Figure 3J**). This suggests a possible correlation between them. However, previous studies have shown that T cell exhaustion usually occurs in tumors enriched with T cells, while in cold tumors, T cells may play a bystander role without activation and exhaustion³, similar to what we observed in this work (**Figure 2G**). The high co-linearity between T cell infiltration and exhaustion made it difficult to draw a conclusion about an independent correlation between pyrimidine metabolism and T cell exhaustion that is not related to T cell infiltration.

The reviewer also asked “how do pyrimidine metabolism and CD8⁺ T-cell infiltration correlate with the IM2-like/IM4-like sub classification?”. The original IM subgroups and the IM2-like/IM4-like classification describe the sub-group characteristics of ccRCC from two different

dimensions. To provide an intuitive perspective, we performed UMAP dimensionality reduction on TJ-RCC samples (**Response Figure 1C-1D**). The original IM subgroups describe the characteristics of the entire tumor microenvironment, while IM2-like/IM4-like focuses on the characteristics of tumor cells. Therefore, pyrimidine metabolism only shows correlation with the original IM subgroups (enriched in IM3), but is not related to the classification of IM2-like/IM4-like.

Response Figure 1: A: ssgsea scores of CD8+ T cell signature in TJ-RCC. P-values were calculated turkey's test. ***: $p < 0.001$, **: $p < 0.01$; B: Spatial transcriptome of Y7_T colored by ssgsea score of Tex signature; C-D: UMAP of TCGA KIRC samples colored by (C) ntp predicted IM subgroups, (D) ssgsea score of particular gene signatures.

Reference

1. Clark, D.J., *et al.* Integrated Proteogenomic Characterization of Clear Cell Renal Cell Carcinoma. *Cell* **179**, 964-983.e931 (2019).
2. Hu, J., *et al.* Single-Cell Transcriptome Analysis Reveals Intratumoral Heterogeneity in ccRCC, which Results in Different Clinical Outcomes. *Molecular therapy : the journal of the American Society of Gene Therapy* **28**, 1658-1672 (2020).
3. Zhang, J., Huang, D., Saw, P.E. & Song, E. Turning cold tumors hot: from molecular mechanisms to clinical applications. *Trends in immunology* **43**, 523-545 (2022).

Algorithm for the assignment of IM1-4 or IM2-like/IM4-like subtypes

The authors should briefly mention in the results section that the IM2 and IM4 are ssgsea enrichment scores for IM2 and IM4 marker genes. This documents that the same methodology that was used to predict pathways and cell type compositions was also used for the tumor subclassification. This reviewer also assumes that the same algorithm was used to group samples into IM1-4 subtypes, i.e. bulk transcriptomic profiles were subjected to ssgsea enrichment analysis using the four gene signatures, followed by scale centering of each score across all tumor samples. Searching for the highest scale centered score allowed sample subtype classification. This approach would ensure that IM4 and IM2 samples would be classified as IM4-like and IM2-like, respectively. Please describe the algorithm in the method section and briefly in the results section.

Response: We thank the reviewer for the suggestion. We have pointed out that IM2 and IM4 scores are calculated using ssGSEA algorithm in the results section (Line 453) as suggested by the reviewer.

The reviewer also expressed that the algorithm for grouping samples into IM1-4 subtypes should be detailed in the methods and briefly mentioned in the results. Extension of the IM classifier of ccRCC was based on NTP algorithm using 856-gene signature (Line 202-207, Line 1268-1273). This algorithm could classify bulk-seq samples directly into preset subgroups. Since IM1 and IM3 show intermediate IM2 and IM4 scores, we applied a ssGSEA-based algorithm to group them into IM2-like or IM4-like group. Firstly, IM2 score and IM4 score were calculated per sample. Then, these ssgsea score were scale centered to balance the weight of IM2 and IM4 scores. Finally, IM1 and IM3 samples with IM2 score > IM4 score were defined as IM2-like, while others were defined as IM4-like (Line 1323-1333). Both two approaches have been detailed in the methods and mentioned in the results section (Line 169-175, Line 458-460) now.

Survival analysis using IM2- and IM4-like subtype classification

The authors replace the initial categorization of samples into IM1 and IM3 subtypes by a new classification, i.e. IM2- and IM4-like. They developed an IM2 and an IM4 score and used the difference of both scale centered scores for reclassifying tumors initially classified as IM1 and IM3 tumors into IM2- or IM4-like tumors.

The new classification assigned different samples into the same groups and does not correlate with the initial IM1 and IM3 classification. The authors document a gradual decrease in the survival rate from IM2/IM2-like over IM4-like to IM4 tumors, using the TCGA KIRC data. To document, if the re-categorization of the IM1 and IM3 samples increased the predictive strength, the authors should briefly compare the results to the Kaplan Meier curve for the initial IM1-4 classification that they reference in an earlier section of their manuscript (Suppl. Figure 2F).

Before introduction of the new classification, the authors document how the initial IM1-IM4 classification succeeds in predicting survival rates in three

chemotherapeutic studies and can help with the decision of drug treatment schemes for each patient. The authors finally prefer the IM2, IM4, IM2-like and IM4-like classification and show that it improves the survival prediction, using the TCGA KIRC patients. It should be of high interest for the reader, if a similar improvement can be observed for the three clinical studies as well. The analysis of the clinical studies based on the new classification should technically become part of the main manuscript, while the results from the initial classification should be part of the supplemental information.

Response: We thank the reviewer for raising this important analysis. As suggested by the reviewer, we applied the updated classification to three immune therapy cohorts. For patients receiving only TKI treatment, this classification showed similar but relatively weaker predictive ability compared to TCGA (IMmotion, JAVELIN), possibly because these cohorts included only mRCC patients, and a similar phenomenon was observed when analyzing only advanced cases in TCGA (**Extended Data Fig. 7E & 8A-8G**). When we applied the corrected classification to the immune therapy group, there were significant changes. In IMmotion151, the corrected classification had a weaker predictive ability for treatment outcomes than before correction. Our previous analysis indicated that atezo_bev combination therapy could improve the prognosis of IM3-type ccRCC but not provide additional benefits for IM1-type. If IM1 and IM3 types are corrected as IM2-like and IM4-like respectively, neither group can benefit from combination therapy. We believe that this can be explained by the features of the two classification methods themselves. The initial IM classification was based on signals from the entire microenvironment. IM3 responded better to immune therapy while no benefit was observed for IM1 which suggests that tumors with T cell infiltration may have higher response rates to ICB therapies, consistent with conclusions drawn from checkmate cohort studies¹. While the corrected classification system classifying ccRCCs into an "IM2-like" or "IM4-like" category attempt to exclude factors

originating from microenvironments and focus on expression profile characteristics of cancer cells themselves. It indicates a possible dose-dependent relationship between DCCD transformation degree and ccRCC prognosis in a cohort with non-ICB nature. However, regarding predicting immune therapeutic effects, TME patterns may play a dominant role so applying these classifications did not show advantages consistent with those seen in TKI groups alone.

At the same time, there is a similar problem presented in the checkmate cohort where nivolumab has obvious relative advantages in the IM3 group whereas after correction this advantage disappeared, further increasing the credibility of this explanation.

Based on this, we have adopted the reviewer's suggestions and added the performance of the corrected classification method in the immunotherapy cohorts to the supplementary figure (**Extended Data Fig. 8**), and discussed it in the text (Line 473-483).

Reference

1. Braun, D.A., *et al.* Interplay of somatic alterations and immune infiltration modulates response to PD-1 blockade in advanced clear cell renal cell carcinoma. *Nature medicine* **26**, 909-918 (2020).

Kaplan Mayer comparisons based on tumor stages

To see if the new initial and/or final classification outperforms the traditional one, all Kaplan Mayer analyses should be generated after grouping the patients based on the currently used clinical classification (probably pTNM, or only without consideration of N and M criteria for a fair comparison).

Response: We appreciate and have adopted the suggestions of the reviewer to perform KM analysis on TCGA KIRC cohort based on tumor stage. Due to uneven distribution of samples across different stages and too less stage ii & iv cases are involved in TCGA KIRC, analyzing tumors of stage ii and iv separately failed to yield statistically significant results. However, when grouping according to localized (stage i, ii) versus advanced (stage iii, iv) tumors, we found that in the advanced tumor group, IM2 subtype had the best prognosis including OS and PFS while no significant differences were observed among other groups (**Figure 4K**). In early-stage samples, IM4 had the worst prognosis with no significant differences observed among other three subtypes (**Figure 4K**). The corrected classification showed that in early-stage cases $IM2 \approx IM2\text{-like} > IM4\text{-like} > IM4$. In advanced tumors, PFS still exhibited similar features after correction but there was no significant difference in OS between IM1, IM3 and IM4 tumors (**Extended Data Fig. 7E**). Overall these results suggested that our new classification system can provide a more precise categorization of ccRCC within existing frameworks. Similar analyses could not be performed on clinical trial cohorts as TNM stage was unavailable for all three clinical treatment cohorts which consisted solely of mRCC.

Correlation between IM2/IM4 scores in malignant cells and bulk transcriptomic IM subtype

The IM subtype gene expression signature was initially developed as a TME signature using marker genes obtained by single cell RNAseq of only non-malignant cell types. Applying this signature to 100 bulk transcriptomic samples lead to the identification of a second signature set, that now is also influenced by the malignant cells that contribute to the bulk expression values. This overall signature is then used to classify malignant cells obtained from the same samples into IM2-like and IM4-like malignant cells. The authors document that the relative malignant cell subtype proportions vaguely follow the subtype classifications annotated based on the bulk

transcriptomic data. Would it not be more reasonable to compare the cell counts of the IM2- and IM4-like malignant cells relative to all cells in each single nucleus RNAseq dataset, since these should give a better estimation of how much gene expression in malignant cells contributes to bulk expression?

Response: We are grateful for the reviewer's helpful suggestions. When designing the IM2-like and IM4-like signatures, we filtered all genes related to extracellular matrix in order to obtain signals only from cancer cells. Following the reviewer's suggestion, we calculated the proportion of malignant cells with IM4-like and IM2-like signatures among all single-cell nuclei. We found that this method yielded inter-group differences consistent with those obtained using the original approach. By calculating scores for all cells, we discovered that the IM4 signature only came from tumor cells and CAFs, while the IM2 signature only came from tumor cells (**Response Figure 2**). This may explain why both of them could represent bulk-level features well. According to the reviewer's suggestion, we added the result from this new model to better describe how much malignant cells contribute to bulk expression (**Extended Data Fig. 9E**).

Response Figure 2: UMAP showing IM2 score and IM4 score per cell. Each dot was colored by ssGSEA score.

Statistical analysis of spatial transcriptomics and metabolomics including all samples

The authors present results of spatial transcriptomic and metabolomics analyses in figures 3G, 3H, 3J, 3L, 3K, 3M, 6J, 6M and 6N. In the mentioned figures, they document in two different samples that spatial hypoxanthine and guanine abundancies correlate with spatial cytotoxic T-cell infiltration within the same sample. Using one of those samples, they document that the activity for the GO pathway ‘pyrimidine metabolic process’ correlates with the areas expressing a T-cell marker gene and hypoxanthine expression. Although the authors subjected 14 samples to spatial transcriptomic and metabolomics analysis (Figure 1A), they only show the results for two samples. How does the correlation look for the other 12 samples? All three compared entities, i.e. pyrimidine metabolism activity, CD8A, hypoxanthine and guanine expression are quantitatively described for each hexagon. It should be possible to calculate the correlation for each sample and then show a beeswarm or similar plot for all samples. This should be applicable at least for CD8A expression and pathway activity, since they were both determined from the same dataset. If non-overlapping (hexagon) fields between spatial transcriptomic and metabolomics do not allow such a comparison, the authors should clearly state this. If samples have to be excluded, because they do not show transition from IM2-like to IM4-like areas, the authors should state this as well.

Response: The reviewer asked “how does the correlation look for the other 12 samples”. 14 samples subjected to spatial omics profiles include 12 tumor samples and 2 adjacent normal samples. The adjacent normal samples were designed as negative control but not involved in tumor subcluster analysis. The spatial transcriptome data of renal cortex was used as the reference in spatial CNV analysis (Line 1114-1116). In the 12 tumor samples, only 2 samples (R29_T & Y7_T) show significant T cell infiltration and only 2 samples show transition from IM2-like to IM4-like areas. We have added this statement to the results section now (Line 315-316, Line 502-503).

Gene, protein, metabolite and pathway expression metrics

It is generally difficult to follow the different value types that are submitted to the different analyses and are used for the visualization. It is also not clear, if the data is calculated and presented using the same algorithms and value types across the different figures.

Response: The reviewer expressed the concern that “It is generally difficult to follow the different value types that are submitted to the different analyses and are used for the visualization.”. We thank the reviewer for the comments and have added clear statement to the methods. For the bulk RNA-seq data, the count matrix was only used in DEGs analysis and the other analysis of bulk transcriptome was based on the TPM matrix (Line 783-785). The results of global proteomics were quantified as described in the methods and then normalized using VSN algorithm (Line 1015-1017). Similarly, LC-MS and GC-MS were quantified respectively and only one output matrix was involved in this work (Line 1073-1074, Line 1085-1087). Except results of GO enrichment in Figure 2A and Figure 4B, all other pathway enrichment analysis was

based on the same output matrix of ssGSEA methods (Line 1194-1202). To make the figures friendly to the readers, we have also modified these figure legends.

Z-score transformed gene expression

Figures 2A, 4A, 4H, 5A, 5D, 5E, 6B, 7C, 7E, 7G all show gene expression values after z-score transformation. Although it looks like the documented values were generated by the same algorithm for all different datasets, there are some things that seem to contradict this conclusion. This reviewer assumes, that the Z-scores were calculated by scaling the expression of each gene over all documented tumor samples, as figure legends 5A/D/G suggest. Scaling means, subtracting the mean from each sample and dividing by the standard deviation.

Nevertheless, Figure 2A and the legend of figure 2A use the term differentially expressed genes, which normally labels gene sets that are significantly different between two conditions, e.g. each immune subtype and NAT.

Response: We thank the reviewer for pointing this out. Our z-score transformation strategy was the same as described by the reviewer and has been detailed in each figure legend that involved this approach now.

In **Figure 2A**, DEGs referred to the differentially expressed by each IM subtype comparing to all the other groups. They were calculated by edgeR strategy as described in the methods (Line 1166-1171). Expression matrix of DEGs of each IM subtype was then transformed into row z-score and used to draw the heatmap shown in **Figure 2A**. This information has also been added to the figure legend now.

Figure 4A does not only show z-scores for the four immune subtypes, but also for the NATs. All other figures show only z-scores across the tumor samples. This raises the question if the Z-scores in figure 4A are different from the ones shown in figure 2A, since they now also consider the gene expression in the NAT samples. Were the NAT samples used for all Z-score calculations, but excluded from the visualization in the other figures? This reviewer thinks that the TCGA KIRC samples and the clinical studies do not contain NAT samples that could be included.

Response: The reviewer pointed out that only Figure 4A visualized both tumors and NATs. We thank the reviewer for this comment. Due to the unique changes of classic cancer genes and ccRCC-related metabolic features in IM4, we introduced NAT samples in Figure 4A to assist in illustrating our findings. These samples were only included in the analysis of Figure 4. As mentioned by the reviewer, the analysis of TCGA KIRC and Figure 2A did not involve NATs. This information has been stated in the modified figure legend to make it clear now.

The legend to figure 4A states that this figure shows the clustering results of up- and down-regulated genes in DCCD, i.e. IM4 tumors. To the understanding of this reviewer, the figure shows the clustering results of the row z-score distribution across all IM subtype and NAT samples, using only those genes that were up- or downregulated in IM4 samples vs all other IM subtypes. Please clarify.

Response: We thank the reviewer for pointing this out. The genes used in Figure 4A are differentially expressed genes between IM4 and other IM subtypes, including upregulated and downregulated genes. Due to the widespread downregulation of classic cancer genes (*PLIN2*, *HIF2A*, etc.) in IM4, an advanced ccRCC subtype, it contradicts the classical theory of tumor

progression. In order to better understand this change, we introduce NATs here to describe the continuous changes of tumor-related genes in carcinogenesis and tumor progression. Therefore, the z-score in Figure 4A is transformed based on the combined expression matrix of 100 tumors and 50 NATs. It has been clarified in the figure legend of Figure 4A now.

Figure 4H shows row z-scores for selected transporter genes, but this time the gene expression of all samples has been somehow collapsed on each immune subtype and NAT. This reviewer could not find a description of the algorithm that generated these values. For consistency, the z-score values in this figure should be shown in all samples of each immune subtype after ordering them by immune subtype (as in figure 2A). Since this figure also contains NAT samples, it is again not clear to this reviewer, if the Z-scores were calculated on a different basis (i.e., under consideration of NAT samples) than in figures 2A, 5A, 5D and 5E.

Response: The reviewer pointed out that the approach for visualization of multiple heatmaps was not clear. We thank the reviewer for helping us improve the clarity of our manuscript. In **Figure 4H**, we first calculated the mean expression of selected genes in each IM subtype and NATs and then transformed these mean values into row z-score. NATs are involved due to the same reason as we used them in Figure 4A. We have detailed this method in the figure legend now. The approach for z-score transformation of other heatmaps has also been added, respectively.

Differentially expressed genes, proteins and metabolites in tumor versus NAT

For a more general overview of tumor-associated gene expression, the authors document differentially expressed genes, proteins and metabolites in all tumor samples versus NAT (Figures 1I). They use their results to document generally consistent results obtained by the three omic technologies for selected pathway activities. Similarly, they document how differentially expressed genes, proteins and metabolites between IM4 and the other three IM subtypes agree and affect selected metabolic pathway activities. This is regular data visualization strategy and is well suited to support the authors conclusions.

Response: We are grateful for the reviewer's acknowledgement of our thorough analysis.

Z-score transformed metabolite expression

Similar questions arise when investigating the documented metabolite expression values in figure 3E and 4F. Metabolite expression values are presented as Z-scores over all immune subtypes and NAT after calculation of some mean values, and not for each individual sample. Again, the question arises, why did the authors not stay with their standard visualization and how were the subtype selective values calculated. Were the NAT samples included in the Z-score calculation, in contrast to what the gene expression Z-scores shown in figures 2A, 5A/D/E imply?

Response: The reviewer asked “why did the authors not stay with their standard visualization and how were the subtype selective values calculated”. In Figure 3E and 4F, expression matrix of selected metabolites was transformed into mean value per subgroup (row axis) and then transformed into row Z-score. This visualization scheme is designed to adapt the size of the

heatmap to fit the empty space on the layout while preserving the same information as heatmaps drawn by sample. All heatmaps involving Z-score transformation were based on data presented in their respective figures. Only Figure 4A/F/H involved NAT samples because we only considered differences between tumor samples when exploring ccRCC molecular subtypes. As we found that IM4 subtype features enhanced ccRCC tumor-related metabolic characteristics, including fatty acid metabolism and amino acid metabolism (Line 340-341), subsequent analysis also showed that typical ccRCC-driven pathway activity was suppressed in IM4, even lower than levels observed in NATs. Therefore, to investigate this stepwise change from NAT/early tumor to IM4-like tumors, we included NAT tissues in multiple visualization schemes in Figure 4, which were not included in other heatmaps. This has been explained further in the figure legend now.

Pathway enrichment analysis

Figures 1G, 1H, 2A, 3A, 3F, 3M, 4B, 4D, 4E, 4I, 7C, 7P show results of pathway enrichment analysis of transcriptomic, proteomic and metabolomics datasets. To the understanding of this reviewer, the authors seem to use multiple strategies for pathway enrichment analysis.

Response: We thank the reviewer for this comment. Figure 1G (1H in modified Figure 1) show T-value of differential analysis between tumors and NATs (Line 1197-1198). Figure 1H (1I in modified Figure 1) show DA score at pathway level to show metabolic difference between tumors and NATs (Line 1180-1184). Figure 2A,3A show z-score transformed ssgsea score in each tumor sample. Figure 3F show ssgsea score of selected pathways. Figure 3M show ssgsea score of spatial transcriptome data (Line 1116-1118). Figure 4B show results of GO enrichment of module 3 and module 4 genes (Line 1166-1167). Figure 4I show T value of differential analysis of ssgsea score (Line 1198-1200). Figure 7C show continuous changes in the expression

of transcription factors alongside pseudotime. Figure 7P show the results of scMEGA including TF expression, TF activity and targets expression. We have detailed these strategies and briefly mentioned in the figure legends.

Pathway analysis of differentially expressed genes

Figure 2A shows pathway enrichment results of the genes part of the different clusters. This reviewer assumes that all genes that are grouped together by the clustering algorithm were subjected to pathway enrichment analysis, using clusterProfiler, if understood correctly by this reviewer.

Response: We thank the reviewer for pointing out it. Pathway annotated in Figure 2A is the result of clusterProfiler. According to the reviewer's suggestion below, this annotation has been removed from the modified Figure 2A now.

Z-score transformed ssGSEA scores

Single sample GSEA (ssGSEA), as implemented in GSVA package, was used to analyze gene expression values in each sample, followed by row Z-score transformation of the ssGSEA scores across all IM subtype samples, ignoring NAT samples. This reviewer thinks that this strategy was used for the blocks labeled 'ssgsea (RNA)' and 'pathways' in figure 2A and for figure 3A. This enrichment strategy generally matches the strategy to show gene expression values. This reviewer assumes that the authors used Z-score transformed ssGSEA scores to allow easier visualization of the distribution of pathway activities over the different samples. ssGSEA values probably range over different scales for different pathways, making the clustering and visualization of differential pathway activities in the same heatmap difficult. The

authors should state in the methods section, why they subjected the ssGSEA scores to Z-score transformation (and do the same for genes).

Nevertheless, the authors should always state the used methodology in the figure captions and label the dendrogram in each figure with the same label that documents the input scores for the Z-score calculation and the dataset used for enrichment. Labeling one block with ‘ssgsea (RNA)’ and another with ‘pathways’ is misleading, since it suggests different analysis strategies. Use of the term single sample GSEA facilitates and easier understanding, since not every reader might be familiar with ssgsea.

Response: The reviewer’s suggestions have been well taken. As the reviewer assumed, pathway score in **Figure 2A** and **3A** was calculated by ssGSEA and transformed into Z-score to allow easier visualization. We have added this statement to the methods as the reviewer suggested (Line 1202). We have also modified the labels of figure 2A to avoid potential misleading.

The reviewer also suggested that “the authors should always state the used methodology in the figure captions and label the dendrogram in each figure with the same label that documents the input scores for the Z-score calculation and the dataset used for enrichment”. We have modified the captions and labels of figures and supplementary figures following this suggestion.

Linear model to calculate differential pathway activities using ssGSEA scores

The authors subjected ssGSEA scores to differential pathway expression analysis using a linear model implemented in limma. This was applied for figure 3B, that compares pathway predictions between transcriptomic and proteomic datasets. This strategy is briefly outlined in the method section (line 1125) and should be explained in more detail. It is not clear to this reviewer, which samples were compared with each

other? Each tumor sample versus all NAT samples? Each tumor sample versus its matching NAT sample, if available (and possible with n=1)? What do the gray dots represent? Why are they so many? Do they visualize other pathways? Why are there only a few dots with the same color? Why do the numbers of dots with the same color differ between the colors?

In addition, the method section implies that it was not used for GO, as in figure 3B, since it explicitly names the gene lists that were used for analysis. This is misleading.

Response: We thank the reviewer for these questions. Figure 3B was implied to show the consistence between transcriptome and proteome. The x-axis was the T-value from differential analysis of ssGSEA score between IM3 and the other tumor samples. Only GO terms were included in this analysis. The y-axis was the T-value from the same analysis performed on the proteomics data. GO terms associated with pyrimidine, cytidine and other nucleoside were labeled by the specified color while other GO terms were colored in grey. We aimed to show that increased pathway score of pyrimidine metabolism in IM3 group is consistent at both proteome and transcriptome levels. We have added this detail to the figure legend now.

Pathway activity

The label 'Pathway activity' implies a 3rd strategy for pathway enrichment analysis, though this reviewer assumes that it is the ssGSEA score (figures 3D, 3F, 3M). The method section implies this conclusion in line 1127. If so, please replace all pathway activity labels by ssGSEA.

Response: The reviewer's suggestion has been well taken. All the pathway activity came from the results of ssGSEA. We have replaced the labels by "ssGSEA score" as suggested by the reviewer.

Differential abundance score for metabolites

In figure 1H the authors calculate the DA score for metabolic pathways that considers the difference between up- and down-regulated metabolites of the same pathway. The authors correctly argue, that this penalizes identification of pathways with contradicting expression changes

Response: We appreciate the reviewer's comment and acknowledgment of the worth of our analysis.

MetaboAnalyst pathway enrichment

In figures 4D and E, the authors use MetaboAnalyst to identify pathway with over or underrepresented activities in the IM4 subtype.

Different methods to describe pathway activities

The application of different methods for pathway enrichment analysis of transcriptomic, proteomic and metabolomics datasets could give the impression to the reader that presented results depend on the selected method.

In figure 2A two different approaches of pathway enrichment analysis were used: subjection of cluster genes to pathway enrichment analysis using clusterProfiler and

sample specific GSEA. Two of the functions described by the pathways identified by each analysis show functional overlap (‘ECM organization’ + ‘collagen-binding’ vs ‘collagen containing extracellular matrix’ and ‘sprouting angiogenesis’ vs ‘angiogenesis’). Gene and pathway z-scores in this module show similar trends. The other shown pathways do not overlap. One cannot conclude from the pathway enrichment analysis results of the cluster specific genes that an IM subtype of interest has a higher expression of a particular pathway. It can only be concluded that genes involved in a particular pathway cluster together across the different tumor samples. Genes mapping to the first module are up-regulated in IM1 and IM4, but the analysis does not allow a conclusion if those genes are the ones mapping to the pathways on the left side. Were the identified pathways also identified as up- or downregulated in a particular IM subtype based on ssGSEA analysis, followed by limma analysis? This analysis would allow the conclusions that the results section reports. If so, the enrichment results shown next to the DEGs cluster can be removed. Please update figure 2A accordingly or correct your conclusions (line 167–line182).

Response: We thank the reviewer for this suggestion. The pathways labeled on the right side of the top heatmap in Figure 2A represented the modular segmentation of different IM subtype marker genes and GO enrichment analysis for different modules. Following the reviewer's suggestion, we conducted ssGSEA analysis on these pathways and found that they exhibited similar trends in ssGSEA results, reflecting more specific IM subtype specificity (**Figure 2A, Extended Data Fig. 2F**).

We agree with the reviewer’s opinion that “The application of different methods for pathway enrichment analysis of transcriptomic, proteomic and metabolomics datasets could give the impression to the reader that presented results depend on the selected method” and have removed the annotation on the right passage of the heatmap. The corresponding results have been added to

the ssGSEA heatmap, and statistical significance results from linear algorithm have been added to **Extended Data Fig. 2F**.

In contrast, the enrichment analysis of the module genes in figure 4B is legitimate, since these genes were identified as differentially expressed in IM4 vs IM1-3, as long as these pathways are not linked to activities in NAT or non-IM4 subtypes.

Response: We thank the reviewer for this comment and the confirmation of our analysis.

Similarly, the enrichment of cluster genes in suppl. Figure 4A is legitimate, since the results section describes relationships between clusters and pathways.

As another example, the authors document correlation between differential pathway ssGSEA between tumor and NAT samples, as obtained for proteomic and transcriptomic datasets using limma in figure 3B. It would be advantageous to use the ssGSEA scores here as well. If use of the original ssGSEA scores does not allow documentation of a correlation, the authors should justify their approach in the methods section and briefly the results section.

Response: We thank the reviewer for this suggestion. Figure 3B shows the correlation of T-value derived from differential analysis of ssGSEA scores between IM3 and other tumor samples. We have detailed this approach in the figure legend now.

Correlating pathway ssGSEA scores and T-effector cell type abundancies (figure 3D) seems to be a reasonable approach. T effector cell abundancies describe values that are obtained independently for each sample, i.e. these values are not normalized with regard to the abundancies in other samples, as Z-scores and differential pathway activities are. Consequently, independent pathway enrichment scores need to be utilized as well, therefore use of the ssGSEA scores. If so, the authors should describe this line of reasoning for the used pathway enrichment score in the method section and maybe briefly mention it in the results section.

Response: We agree with the reviewer and have described this line of reasoning for this correlation analysis in the methods (Line 1203-1205). It has also been mentioned in the results (Line 293-295). The figure caption has also been modified to make it easier to be understood.

The same accounts for the cross-comparison of pathway activities across all IM subtypes in figure 3F. If the documented pathway activities are ssGSEA scores, it is more reasonable to use the ssGSEA scores instead of the derived Z-score transformed values for statistical tests.

Response: We agree with the reviewer that it is more reasonable to use the ssGSEA scores instead of the derived Z-score in Figure 3F. The initial label was set as pathway activity and maybe a bit confusing to the readers. We have changed it to ssGSEA score now.

Regarding metabolite enrichment analysis, the reader might also wonder, why were differentially expressed metabolites between tumor and NAT subjected to DA analysis

and differentially expressed metabolites between IM4 and all other subtypes or NAT subjected to MetaboAnalyst pathway enrichment analysis.

Response: The reviewer asked “why were differentially expressed metabolites between tumor and NAT subjected to DA analysis and differentially expressed metabolites between IM4 and all other subtypes or NAT subjected to MetaboAnalyst pathway enrichment analysis”. When comparing tumors to NATs, we obtained a large list of differentially expressed metabolites and then subjected them to DA analysis to obtain a global view of metabolism disorder in ccRCC. The DA score took the total number of metabolites involved in a pathway into account, which amplifies differences related to small pathways. However, for the analysis in Figure 4D-4E, we aimed to directly identify the pathways with the greatest changes between subtypes. Therefore, we used MetaboAnalyst to perform enrichment analysis directly on differentially expressed metabolites. We provided additional details on this approach in our methods section (Line 1188-1189). Furthermore, validating MetaboAnalyst results using the DA method revealed similar enrichment tendencies but did not prioritize the same pathways (**Response Figure 3**).

Figure 3: DA scores of selected pathways between IM4 and IM1-3 (left) or IM1-3 and NATs (right). Size of each dot represents total number of metabolites in the pathway.

Generally, it seems important to this reviewer to introduce the different metrics to describe pathway activities and restrict them to a minimum of different approaches. If the ‘pathway activity’ score is the ssGSEA, it would make sense to describe that

the basic measure of all pathway enrichment results are the ssGSEA scores. Similar to gene expression values, these were Z-score transformed to allow clustering and visualization of pathways with similar trends over all samples, independently of the scale of change. Otherwise, the reader might get the impression that the different pathway enrichment scores are selected based on favorable results. Axis labels in the figures should also describe the underlying scores using the same terms (e.g. replace pathway activities by ssGSEA or add ssGSEA in brackets, if this is the used algorithm).

Response: We thank the reviewer for this suggestion. Pathway activity in this paper referred to ssGSEA score. In order to avoid unnecessary misunderstanding, we have changed them to ‘ssGSEA score’ or ‘pathway scores’ as suggested by the reviewer now.

Test for differential pathway activities

Any time the authors conclude from their pathway enrichment analysis that a particular pathway is more or less active in an IM subtype, they should state a P/T-value for their conclusion, using the linear model from limma (e.g. lines 167–182 or line 262, lines 288–290).

Differential expression analysis of the transporter genes (figure 4H) should be added as a documentation of significance as well

Response: The reviewer’s suggestion has been well taken. T value from the linear algorithm of limma has been visualized into heatmaps and added to the supplementary figure (**Extended Data Fig. 2F**) as suggested by the reviewer. Result of differential expression analysis of transporter genes has also been added (Figure 4H).

Methods section

It is difficult to align the methods section to the results and the figures. It should give a general overview of which datasets were analyzed by which method and with which gene library to obtain which type of values. The method section would profit from flow charts that summarize the analysis steps starting with each raw dataset (e.g., raw read counts) and how one value (e.g. Z-scores) derives from another (e.g. ssGSEA scores). Please add such flow charts to the supplemental information.

Response: We agree with the reviewer's opinion and flow charts have been added to the supplementary figures now (**Extended Data Fig. 2D, 3E & 7A**). We have also added part of the method details to the figure captions to help readers understand the methods to generate these figures.

Figure captions for main and supplemental figures

The figure captions of all figures are insufficient. They should contain enough information, so that the reader can get a sufficient understanding of the figures without reading any other part of the manuscript. The supplemental figures are missing figure captions. Please add, following the same principal.

Response: We thank the reviewer for pointing out this important thing. We are sorry for the loss of supplementary figure legend in the original manuscript and it has been added now. Legends of main figures and supplementary figures have also been modified as suggested by the reviewer.

Minor comments

Single nucleus RNAseq

The authors show the overall results of single nucleus RNAseq in figures 2E, 2F, 2G, 2H and focus on malignant cells in figures 6A, 6B, 6C, 6D, 6E and 6F. While figure 2E shows one island that contains the malignant cell types, figure 6D shows multiple islands of malignant cell types, one for each analyzed samples. What is the difference between the analyses used to generate both figures, besides the focus on only one cell type in figure 6D? Normally, resubmission of a cluster to the clustering pipeline to identify subclusters does not separated cells that were initially close neighbors from each other.

Response: We thank the reviewer for pointing out this issue. Due to the nature of droplet-based snRNA-seq technology, batch effect between samples is inevitable. Since then, we performed a harmony algorithm to regress out such batch effect, and got the UMAP plot shown in Figure 2E. However, algorithms for batch effect curation available now are all not suitable for analysis of tumor cells. Since solid tumor cells have strong patient-specific heterogeneity, over-estimation of batch effect is inevitable¹. Harmony-curated tumor cells could only maintain commonality between tumor samples, but will lose a considerable portion of heterogeneity. Since then, we used ‘pca’ instead of ‘harmony’ to construct the UMAP coordinate in **Figure 6D**. We have added this approach details to the methods (Line 854-856).

The reviewer also asked why we focus on only one cell type in **Figure 6D**. As shown in **Response Figure 4** below, both IM2 signature and IM4 signature are epithelial specific signatures. Apart from epithelial cells, IM4 signature could also be detected in fibroblast, which

is of relatively low abundance no matter in which IM subgroups. Since then, when analyzing IM2 and IM4 features, we only considered epithelial cells.

Response Figure 4: UMAP showing IM2 score and IM4 score per cell. Each dot was colored by ssGSEA score.

Reference:

1. Chen, W., *et al.* A multicenter study benchmarking single-cell RNA sequencing technologies using reference samples. *Nature biotechnology* **39**, 1103-1114 (2021)

Please also clearly state that the malignant single cells were only mapped to IM2-like and IM4-like cells, but not to IM2 and IM4 subtypes. In case of the own and TCGA bulk transcriptomic datasets only IM1 and IM3 samples were mapped to IM2-like and IM4-like samples, while at the same time the initial IM2 and IM4 subtype classification was kept. Assuming the same approach for the single malignant cells,

the reader might think that the IM2 and IM4 cells were excluded from the analysis shown in figures 7A/B/C, which is inconsistent with their consideration for the TCGA data in figures 7F/G.

Response: We thank the reviewer for helping us improve the clarity of our manuscript. We have clearly stated that malignant single cells were only mapped to IM2-like and IM4-like cells in the methods (Line 1331-1333) and mentioned in the figure legend.

Arrangement of malignant cells along pseudotime

The authors document that malignant cells can be arranged along one trajectory that correlates with the IM2-like and IM4-like scores (Figure 7A). It should be clearly stated that the IM2 and IM4 scores shown in figure 7B were calculated based on the gene expression in the malignant cells (and not based on the samples they map to). Consequently, the arrangement of the malignant cells along the pseudotime is dominated by those genes that contribute to the IM2/4 scores.

Response: We thank the reviewer for the suggestions. We have clearly stated that the IM2 and IM4 scores shown in **Figure 7B** were calculated based on the gene expression in the malignant cells and the arrangement of the malignant cells along the pseudotime is dominated by those genes that contribute to the IM2/4 scores in the figure legend.

Pseudotimeline and transcription factor profiling

To identify potential transcription factors that determine the progression from IM2 to IM4 subtypes the authors calculate differentially expressed genes between IM4-like

and IM2-like malignant cells and samples. The authors should add that only the malignant cells were used from the single cell RNAseq dataset, as this reviewer assumes from the previous analyses. Please explain in the methods section and explain in the results section why only malignant cells (or all cells) were used for the analysis (e.g. add ‘to identify the TF motifs in malignant cells’).

Response: The reviewer suggested that reasons for why only malignant cells (or all cells) were used for the analysis should be explained in the methods and results section. Since DCCD process defined in this paper described a transformation process happened in tumor cells, we only considered malignant cells in pseudotime and transcription factor analysis. As suggested by the reviewer, we have added the statement to the results section (Line 519-522). We have also explained why we only use malignant cells in the methods (Line 893-894).

Discussion

Since the IM4 subtypes depend on increased nutrient import, could the changed TME support the tumors?

Response: We thank the reviewer for raising this interesting question. We believe that TME could provide support for tumors. Compared to other IM subtypes, there are significantly more complex fibroblast components in the IM4 microenvironment. It is generally believed that CAFs can support the growth and proliferation of tumor cells through multiple pathways. However, at the snRNA level, we cannot determine whether CAFs can directly provide nutrients needed by tumor cells because this information is at the mRNA level within cell nuclei rather than at the single-cell metabolite level. Spatial metabolomics may provide some support, but current spatial metabolomic techniques rely on first-level mass spectrometry (MS1) for metabolite analysis with

limited throughput and individual spots far exceeding single-cell diameters; therefore, it is still difficult to give a clear localization of metabolites. Using flow cytometry technology to classify microenvironmental cells followed by metabolic sequencing or higher-precision spatial metabolomics techniques may help us identify this phenomenon. We have already begun trying these two approaches.

Generally, figure 2A suggests upregulation of angiogenesis in IM1 and IM4 subtypes, although endothelial cells were primarily associated with IM1 and IM2.

Response: We thank the reviewer for pointing out this issue. Angiogenesis pathway score in **Figure 2A** is derived from ssGSEA score of ‘Hallmark angiogenesis’, which also includes a series of CAF-derived signatures, such as *VCAN*, *POSTN*, *COL5A2* et al. Since then, activity of this pathway cannot represent the abundance of endothelial cells directly.

Origin of the initial IM gene expression signature

The authors re-analyzed published single cell RNAseq data to identify an initial signature of 911 protein-coding genes for initial subtype classification. Here, the authors are missing any description why the initial approach of using these datasets should succeed in the identification of a suitable signature. How many tumor samples were analyzed in this published dataset? Are the analyzed published samples representative of the existing ccRCC spectrum, i.e. do they span a great variety of known tumor stages, molecular and clinical characteristics (i.e. survival rates)? The methods section offers a sufficient description of the underlying methodology that lead to the identification of the signature and suppl. figure 2A visualizes it graphically. Please state in the results section that these 911 genes are the merged

top 60 significant cell type specific marker genes with the highest fold changes. Please also add which cell types were used for this analysis in the results section (i.e. using all cells of the TME, excluding the malignant cells), when referencing suppl. figure 2A.

Response: The reviewer asked “why the initial approach of using these datasets should succeed in the identification of a suitable signature.”. Currently, most of the publicly available deconvolution reference matrices are based on expression matrices provided by large-scale cell atlas projects, which do not involve disease-specific cell types. For example, the reference matrix of xcell is extracted from ENCODE and the matrix of cibersortx is from microarray profiling of cells collected by FACS sorting. Moreover, recent pan-cancer single-cell atlases have demonstrated the existence of numerous tissue-specific or cancer-specific cell types in different tissues and cancer types, indicating that common signal matrices are insufficient to describe the complete microenvironmental features of tumors. Therefore, it is necessary to develop microenvironmental signal matrices specific to certain tumors. We have supplemented this information in the methods section (Line 1230-1232).

The published scRNA-seq dataset includes 19 samples derived from 10 patients. Although the cohort is not large enough, these 10 patients included ccRCC of different stages and grades, which could give the overall characteristics of ccRCC. Meanwhile, in order to avoid excessive reliance on a single dataset and improve the accuracy of bulk-level analysis, we only classified cells into broad categories rather than subtypes, and removed marker genes shared by different cell types. According to the reviewer's suggestion, we have stated in the results section that these 911 genes are the merged top 60 significant cell type specific marker genes with the highest fold changes (Line 171). We have also added details about the specific cellular components included in the signal matrix to the results section (Line 172-173).

Clear labeling of analyzed datasets

The authors separate their own samples into the four IM subtypes and characterize cell-type composition, pathway activities and potential cell type interactions of the different subtypes (Figure 2A). Here, the authors should state, that predicted cell type abundancies arise from enrichment analysis of bulk gene expression profiles, to prevent misunderstanding that the described results are based on single cell/nucleus datasets.

Response: We thank the reviewer for helping us improve the clarity of our manuscript. We have added this statement to the figure legend.

Sample numbers

Please always mention sample numbers subjected to each omics assay in the method, results and figure captions.

Response: We thank the reviewer for this suggestion. We have added sample numbers in the method, results and figure captions now.

xCell

How was xCell applied to the data? This reviewer could not find any description in the methods section or a reference.

Response: We appreciate the reviewer for informing us of this missing method details. To compare the result generated by our pipeline to that of xcell, we obtained a cell composition matrix from the online tool of xcell. We have added this statement to the methods (Line 1210-1213).

Abbreviations

Please double check that all abbreviations are introduced by their full terms in the main text (e.g. TCGA KIRC). A supplemental table with general abbreviations, on top of the one with metabolite abbreviations, would be very helpful too.

Response: We agree with the reviewer that a full list of abbreviations would be very helpful. As suggested by the reviewer, abbreviations apart from those of metabolites have been added to Table S5.

Line 191: For an easier understanding, it would be helpful, if the survival analysis of the four subtypes in the TCGA data would form a new paragraph.

Response: We thank the reviewer for this suggestion and have made the survival analysis a separate paragraph.

Line 225: Please replace ontology by pathway or gene list.

Response: Ontology has been replaced by pathway now.

Line 251: Using the full term ‘Tumor Microenvironment’ in the headline might be better, if in agreement with the journal’s style

Response: We have taken the reviewer’s suggestion. TME has been replaced by full term “Tumor Microenvironment”.

Line 259: ‘Cluster 2 was derived from interstitial cells and endothelial cells, but not from malignant cells.’ Please explain how the single nucleus data allowed this conclusion.

Response: As shown in **Extended Data Fig. 5A**, Pathways belonging to cluster 2 show the highest ssGSEA score in fibroblast, pericyte, smooth muscle cells and endothelial cells, but were expressed at low levels in other cell types. The figure legend was missing in the initial manuscript and has been added now. To make the statement more accurate, we have modified it to ‘mainly derived from interstitial cells’.

Line 274: Please specify how many samples were subjected to spatial transcriptomic and metabolomics analysis (figure 1A) and which subtypes these samples belonged to. In particular, how many IM3 samples were analyzed to support the infiltration of colocalization of CD8+ T-cells and gyanine and hypoxanthine metabolism.

Response: We appreciate the reviewer for the suggestions. We have now specified how many samples were used in spatial transcriptomics and metabolomics (Line 302-304). It has also been

made clear that R29_T, R51_T, and Y7_T were used to visualize the localization of guanine and hypoxanthine.

Line 352: Here a brief sentence could be added that the downregulation of metabolic energy generation pathways implies alternative energy sources for IM4 tumors. Consequently, the expression of membrane transporters was investigated.

Response: We thank the reviewer for this suggestion and have added such a sentence to make the document more coherent.

Line 367: Please state that IM4 is renamed into DCCD. Line 368: Please introduce abbreviation and state that IM4 is DCCD.

Response: The reviewer's suggestion has been well taken. We have added the statement that "IM4 tumors were named as De-clear cell differentiated (DCCD) ccRCC since these tumors have finished this transformation process".

Line 380: If a drugs mechanism of action can be linked to identified pathway of a particular IM subtype, it would be helpful to briefly mention it (e.g. axitinib inhibits angiogenesis in line 395).

Response: We agree with the reviewer and have taken the advice. We have documented that ccRCC with higher angiogenesis activity may benefit more from TKIs. Additionally, IM3 tumors may be candidates for ICB therapy due to more tumor-infiltrating lymphocytes in the TME.

Line 382: Please state what kind of sequencing data was used (i. e. bulk transcriptomic, zscore)

Response: We thank the reviewer for showing us where we can be clearer and have modified the statement.

Line 412: Adding the expression low-risk IM2 and high-risk IM4 might better illustrate why IM1 and IM3 subtypes are reclassified.

Response: The expression has been modified as the reviewer suggested.

Line 512: Should it not be AR negative tumor to indicate that AR is downregulated in H46_T relative to the other tumors?

Response: We agree with the reviewer that AR negative may be inaccurate here. It has been replaced by AR^{low} now.

Line 482: Since the clusters are not annotated, it might be better to use the expressions top and bottom clusters instead of cluster 1 and 2.

Response: The statement has been modified as the reviewer suggested.

Line 550: The HIF2A-dependend LD accumulation occurs only in one molecular subtype of ccRCC. How is this conclusion made?

Response: We thank the reviewer for pointing out this. The initial statement is not accurate. Accumulation of these LDs occurs in both IM2 subgroup and IM2-like region of IM1 and IM3. It could be observed in both Oil red O staining (Figure 4C) and spatial metabolomics (Figure 6I). The statement has been already modified.

Line 583: Why does the observation that “loss of AR is a continuous, importat process of DCCD” indicate that “lymph nodes are the potential colonization site for these micro metastatic niches” ?

Response: We thank the reviewer for pointing out this issue. In this paper, we found that expression level of AR decreased alongside the DCCD process, which means AR^{low} ccRCC should located at the end of this transformation axis. Previous study has demonstrated that loss of AR in ccRCC may be related to a higher lymph node metastasis rate, but not lung metastasis rate (Line 648-650). Apart from that, we also find that early stage IM4 tumors that finished DCCD have a much higher recurrence rate than other groups. Current first-line treatment strategy for these patients is radical nephrectomy. If these tumors are strictly limited in the kidney, it could

not be explained why these tumors could recur outside the kidney. Taken together, we hypothesized that DCCD ccRCC could transfer to tissues, outside the kidney and form micrometastasis niches at an early stage. According to higher lymph node metastasis rate, these tumors may prefer lymph nodes to other tissues to form these micrometastasis niches. We admit that this theory is just a hypothesis now.

Line 600: Mentioning of drug targets can facilitate an easier understanding of the underlying biology.

Response: We thank the reviewer for this suggestion. Drug targets have been added following the drug names.

Line 711: Please add that the 50 NAT samples are from the same specimen as 50 of the tumor samples, as this reviewer assumes.

Response: We have added this statement as the reviewer suggested.

Line 720: Does it mean “enrich target mRNA” ?

Response: We thank the reviewer for informing us of this mistake and now made appropriate changes to the manuscript.

Line 741: Please write “centrifuged” .

Response: We thank the reviewer for pointing out this mistake and have now made appropriate change to the manuscript.

Line 802: Were p-values and adjusted p-values considered in the manuscript?

Response: We thank the reviewer for pointing out this issue. We have checked the code and found that p-values of ORR analysis were calculated by Fisher’s exact test and then adjusted by BH method. It was not labeled to the Figure in the initial manuscript and has been added now (**Figure 2G**). We have also modified the methods (Line 858-866) and mentioned it in the figure legend.

Line 807: Please also describe that bulk transcriptomic data was subjected to ssgsea based pathway enrichment analysis.

Response: We thank the reviewer for this suggestion. Here we described application of ssGSEA in snRNA-seq data. Application of ssGSEA in bulk transcriptome and proteome has been described separately (Line 1194-1196).

Line 812: Please state that ssgsea was also used with other gene libraries.

Response: The reviewer's suggestion has been well taken. We have added the statement that ssGSEA was also performed on gene list defined in this paper (Line 873-874).

Line 814: Does the description of the correlation refer to figure 2H looks? That looks more like Pearson correlation.

Response: We thank the reviewer for pointing out our mistake. The description referred to **Extended Data Fig. 3L** and is also calculated by Pearson's correlation strategy. We have modified this manuscript appropriately now.

Line 824: Please add transcription factor profiling to the headline.

Response: We thank the reviewer for this suggestion. The headline has been modified now.

Line 827: Please indicate that the IM2-like and IM4-like samples contain all four initial IM subtype classifications, as this reviewer assumes. IM2-like and IM4-like subgroupings were only generated based on the IM1 and IM3 subtypes of the bulk transcriptomic datasets. This might cause confusion, if the IM2 and IM4 subtypes contributed to the identification of transcription factors as well.

Response: The reviewer expressed their concern that the initial expression of the manuscript might cause confusion. As the reviewer assumed, IM2 and IM4 were also involved in this

analysis. The differential analysis was performed between IM2/IM2-like and IM4/IM4-like groups. We have now made appropriate changes to the manuscript

Line 916: Please use lower case letters for ‘parent’ or replace comma by dot.

Response: We thank the reviewer for pointing out the writing mistake and have corrected it now.

Line 1089: Maybe, add ‘between these regions within the same sample’ for more clarity.

Response: We thank the reviewer for helping us improve the clarity of our manuscript. The statement has been added now.

Line 1140: Were three cutoffs applied, one p-value, one q-value and one adjusted p-value cutoff?

Response: We thank the reviewer for this comment. Clusterprofiler 4 returns three cutoff values. When performing this analysis, p-value cutoff and q-value cutoff was set as parameter of enrichGO function. And adjusted p-value was used to choose candidate pathways.

Figure 1

Figure 1A: Is this figure missing the generation of 100 + 50 bulk transcriptomic and 100 + 50 bulk proteomic datasets?

Response: We thank the reviewer for this comment. At the top of this figure, we aimed to show that 100 tumors and 50 NATs were subjected to 4 different bulk omics profiling. WTS is the abbreviation of whole transcriptomic sequencing. We have modified this figure to make it clearer.

Figure 1B: The arrangement of the figure legends that explain the colors is initially misleading, since the top figure is separated from its legend by the bottom figure.

Response: We have taken the reviewer's suggestion. The legend for the colors has been moved to the top of the figure.

Figure 1B: What is the difference of this figure and suppl. figure 1A? Why do the values differ?

Response: We appreciate the reviewer for informing us of this missing explanation. **Extended Data Fig. 1A** shows the mutation rate of selected genes in another Asian cohort (PKU). We have added the supplementary figure legends now.

Figure 1C: What do the abbreviations FUSCC, PHU and TJRCC stand for?

Response: We thank the reviewer for informing us of this missing information. FUSCC referred to the RCC cohort profiled by a team from Fudan University Shanghai Cancer Center (FUSCC).

PKU referred to the RCC cohort profiled by a team from Peking University (PKU). TJRCC referred to the cohort profiled in this paper (RCC cohort of Tongji Hospital). We have added full names of these abbreviations to the manuscript (Line1370-1372 & Figure legend 1C).

Figure 1D: Please reference in the figure caption that the colors are the same as in figure 1B.

Response: We thank the reviewer for the suggestion. This statement has been added to the figure legend now.

Figure 1F: The flow chart in figure 1A documents generation of two transcriptomic datasets, i.e. single nucleus RNAseq and Spatial transcriptomics. Figure 1F compares gene expression profiles of each sample. Please add that each dot represents the bulk transcriptomic, proteomic and metabolomics profiles of 100 tumor and 50 NAT samples?

Response: We thank the reviewer for helping us improve the clarity of our manuscript and we have added this statement to the figure legend now.

Figure 1G: How were the T-values calculated? The method section describes in the chapter ‘Gene set enrichment analysis’ that a linear model implemented in limma was used to calculate the difference between ssGSEA pathway scores between tumor and NAT samples. Nevertheless, it does not mention that is used Gene Ontology for this. Were GO biological processes used in this figure? Were redundant pathways removed as a functionality offered by clusterProfiler that is referenced in the next method

section ‘Pathway enrichment analysis’ ? How were the shown biological processes selected?

Response: We thank the reviewer for these questions. T-values were calculated by linear model in limma that returned along with p-value. This analysis was performed using a pooled geneset including hallmark, c2 and c5 downloaded from MSigDB as we described in the methods (Line 1194-1197). ClusterProfiler was used to analyze gene modules of Figure 4A but not here. The reviewer also asked ‘How were the shown biological processes selected’. Since there is a huge difference between tumors and NATs, thousands of pathways show significant differences between them. Hence, in this figure, we only visualized well-known cancer-associated alterations in ccRCC to investigate consistence between different omics. We have added a supplementary table to show the complete result of this analysis (Table S6).

Figure 1H: What do the dashed lines show? What do the numbers show (i.e. total number of up- and down-regulated metabolites in pathway)? Please specify in the figure caption.

Response: We thank the reviewer for pointing out these issues. The dashed lines don’t contain any information. They were only used to beautify the bubble plot. The numbers show the total number of metabolites involved in the pathway. This information has been added to the figure legend.

Figure 1H: Which metabolite library was used for the analysis?

Response: KEGG pathway database was used in this analysis. We have added this detail in methods (Line 1183-1184) and mentioned in the figure legend now.

Figure 1H: Please write differential abundance score or explain the abbreviation DA in the figure caption.

Response: We thank the reviewer for this suggestion. Full name of differential abundance has been added to the figure caption now.

Figure 2

Figure 2A: Adding the four IM subtypes as a headline over the four blocks would enhance faster understanding, in particular since the text in the legend is very small.

Response: A headline of IM subtypes has been added now.

Figure 2A: The top block of this figure also contains too much information. Please remove all information from this figure that is not discussed in the text. Detailed information can be shown separately as a supplemental figure.

Response: We thank the reviewer for this suggestion and have modified the top block as the reviewer suggested. SNA and arm-level CNV have been removed from the top block and shown in a supplementary figure instead (**Extended Data Fig. 2E**).

Figure 2A: How were the DEGs calculated? What is the denominator (all reference samples)? Are the DEGs also visualized as Z-scores, as the figure legend implies?

Response: We appreciate the reviewer for informing us of the missing information. DEGs in Figure 2A were defined as genes differentially expressed in one IM subtype versus all other IM subtypes. The visualization strategy has also been described in the figure caption.

Figure 2A: How were the Z-scores from the single sample GSEA values calculated?

Response: The reviewer asked “How were the Z-scores from the single sample GSEA values calculated?”. Single sample GSEA scores were calculated in each sample and only tumor samples were used in Figure 2A. Matrix of 100 tumor samples was transformed into z-score and visualized here. We have added this information now.

Figure 2A: Which algorithm was used to calculate pathway Z-scores? Are these Z-scores also based on single sample GSEA?

Response: We thank the reviewer for helping us improve the clarity of our manuscript. All the pathway score indicates ssGSEA score. Pathway score has been replaced by ssGSEA score now to make it clear to the reader.

Figure 2A: A description of how xCell was used is missing in the method section.

Response: We appreciate the reviewer for informing us of this missing information. We have added details of the approach that xCell was performed using the website tool (Line 1210-1213).

Figure 2B: Please reference CNA as Copy number alteration.

Response: CNA has been replaced by Copy number alteration now.

Figure 2C: Please reference TMB as Tumor mutational burden.

Response: TMB has been replaced by tumor mutation burden now.

Figure 2G: What do the Odds ratios compare? Cell type counts in one immune subtype versus all others? Please frame all ORs discussed in the results section.

Response: We thank the reviewer for informing us of the missing information. Odds ratio was used to evaluate whether these cell subclusters have IM subgroup preference by comparing proportion of one cell type in one IM2 subtype to the others. Immune cells and stromal cells are analyzed separately here (Line 858-866). Odds ratio of endothelial described in results was shown in **Extended Data Fig. 3G**. The supplementary figure legend has been added now

Figure 2H: The number of dots suggests that this is bulk transcriptomic data. Which values were correlated? Are these Z-scores? If so, why does the figure not correlate ssgsea scores, to be consistent with previous figures. Please specify used value types in the figure caption and/or add to axis labels. Which correlation metrics was used, ‘pearson’ as implied by the red interpolation line or ‘spearman’ as mentioned in the methods section (line 814). Please briefly describe what the red line shows.

Response: We thank the reviewer for these suggestions. **Figure 2H** show correlation between specific cell subgroups in the cohort profiled in this manuscript. Cell abundance was estimated using ssGSEA score (Line 874-877). To make it clear, we have added “ssGSEA” to the figure caption. We have also added ‘pearson’s correlation’ to the figure legend as suggested by the reviewer. We also added a statement that the red line shows the linear interpolation of the data

Figure 3:

Figure 3A: Please remove all information from the figure top that is not discussed in the results section.

Response: We thank the reviewer for the suggestion and have removed the irrelevant information.

Figure 3A: Please label the row axis of the heatmap using the same label as used in figure 2A, if, as this reviewer assumes, the same algorithm was used.

Response: We thank the reviewer for the suggestions. IM subgroups have been labeled in the same color as that of Figure 2A. Since unsupervised clustering was used here, it was difficult to label the subtypes directly on the figure. We enlarged the label on the right to make it conspicuous.

Figure 3A: Please add that this analysis is based on the bulk transcriptomic profiles, as this reviewer assumes.

Response: The reviewer's suggestion has been well taken. We have added the statement that ssGSEA scores were calculated using the TPM matrix of bulk transcriptome.

Figure 3A: Though it seems to be obvious, but the explanation that Z-scores were calculated based on mean and standard deviation of the ssGSEA scores across all shown samples should be added to the figure caption for proper documentation.

Response: We thank the reviewer for the suggestion. This statement has been added to the figure legend.

Figure 3B: "T-values were calculated by a linear model of Limma." Please explain in detail. Which values were used as input, ssGSEA values that were probably also used for Figure 3A? Please also add a section to the methods.

Response: The reviewer requested that the figure caption should be explained in detail. Figure legend of Figure 3B has been detailed now. In Figure 3B, SsGSEA scores of bulk transcriptomic

and proteomic data were calculated, respectively. T-values are calculated by a linear model of limma using ssGSEA score matrix. Details of this approach have also been added to the methods (Line 1198-1200).

Figure 3B: How do the pathways ‘cytidine’, ‘nucleoside’ and ‘pyrimidine’ relate to the GO pathways shown in A? Please at least refer to the full GO names in the figure caption.

Response: We thank the reviewer for this comment. Pathways associated with cytidine, pyrimidine or nucleoside were labeled in specific colors in Figure 3B. Pathways visualized in Figure 3A are part of these GO terms. Since a considerable number of pathways were labeled, it’s too long to be listed in the figure caption. We have added them as a supplementary table instead (Table S7).

Figure 3C: Please specify which data was used to distinguish between high and low pyrimidine metabolism (e.g., bulk transcriptomic data).

Response: We thank the reviewer for the suggestion. We have made it clear that bulk RNA-seq data of TCGA KIRC was used here.

Figure 3C: Please specify which values were used to separate samples into high and low pyrimidine metabolic activity. ssGSEA scores/Z-scores from figure 3A or T-values from figure 3B?

Response: We thank the reviewer for informing us of this missing information. Bulk RNA-seq data of TCGA KIRC was used here. We first calculated the ssGSEA score in each sample and then stratified these samples by higher or lower pyrimidine metabolism activity. This approach has been added now.

Figure 3C: Please use the full expression ‘Kaplan–Meier Plot’ in the figure legend.

Response: The figure legend has been modified as the reviewer suggested.

Figure 3D: Please specify which data (e.g., bulk transcriptomic) and which value types were used for this analysis? Please add the score (e.g. ssGSEA) to the axis.

Response: We thank the reviewer for this suggestion. TPM matrix of bulk RNA-seq was used to calculate ssGSEA score. It has been added to the figure legend.

Figure 3D: Please specify in which unit the T effector abundance is shown. Are these relative cell type counts?

Response: We thank the reviewer for pointing out this missing information. T effector abundance was obtained from xcell results provided by the xcell team. We have added this information now.

Figure 3D: Please specify which correlation was used ('pearson' ?).

Response: The missing information has been added. We used pearson's correlation and have made it clear now.

Figure 3D: Please briefly describe that the red line shows the linear interpolation of the data in the figure caption.

Response: We thank the reviewer for this suggestion. This statement has been added to the figure caption now.

Figure 3E: Please specify that this figure shows the results from bulk metabolomics analysis, as this reviewer assumes.

Response: This information has been added as the reviewer suggested.

Figure 3E: Why does this figure not show the metabolite expression in the individual samples, as figure 3A does? How were the Z-scores calculated?

Response: We thank the reviewer for this comment. Here we aimed to use a space-saving pattern to show the heatmap. It does not influence the information shown in this figure. To make the statistical significance clear, we have added adjusted p-value of differential expression analysis to this Figure now. In this heatmap, mean value of each metabolite in bulk

metabolomics was calculated in each IM subgroup and then transformed into z-score. Z-scores were calculated based on mean and standard deviation across 4 IM subgroups. This detail had been added to the figure caption now.

Figure 3F: Which score is shown for the pathway activity? Does the figure show ssGSEA scores?

Response: We thank the reviewer for informing us of this missing information. Pathway activity was evaluated by ssGSEA algorithm. We have changed the axis label to make it clear.

Figure 3G: Please add the dataset shown in this figure. Is it spatial transcriptomics, as this reviewer assumes?

Response: The reviewer's suggestion has been well taken. ST in this figure caption is the abbreviation of spatial transcriptomics. It has been replaced by spatial transcriptomics now.

Figure 3G: Please add the subtype of the sample to this figure.

Response: We thank the reviewer for this suggestion. IM subtype has been added to Figure 3G

Figure 3G: To which figure does the scale bar below the second image belong to? It does not seem to represent the colors in this figure.

Response: We thank the reviewer for pointing out this issue. The scale bar below the bottom image of **Figure 3G** belongs to **Figure 3G**. To make the HE image visible, we set alpha value of this figure to (0.1,1), which means spot with low level of CD8A will be nearly completely transparent.

Figure 3G: It would be helpful to add “Cytotoxic T-cell” to the label CD8A as well.

Response: The reviewer’s suggestion has been well taken. “Cytotoxic T-cell” has been added to the label.

Figure 3H: Which sample is shown for this slide (probably R29_T)? Please add sample information to the figure.

Response: We thank the reviewer for pointing out this missing information. The top line of Figure 3H showed the spatial metabolomics profiling of R29_T. It has been added to the label now.

Figure 3H: What do the two figures in the lower row show? Is this the background noise?

Response: We thank the reviewer for helping us improve the clarity of our manuscript. The lower row of Figure 3H show metabolites level in R51_T. Both two metabolites showed low expression levels. We have added label to make it clear.

Figure 3I: Here, it would also be helpful to add the cell types that are labeled by both marker proteins as well.

Response: We thank the reviewer for this suggestion. “T cell” and “Cytotoxic T-cell” have been added to the figure now.

Figure 3I: Please write Immunohistochemistry staining in the figure caption.

Response: IHC has been replaced by the full name as the reviewer suggested.

Figure 3I: Please add the subtype of the sample to this figure.

Response: The IM subtype has been added to Figure 3I now.

Figure 3J-L: Please write spatial transcriptomics.

Response: We thank the reviewer for this suggestion. We have documented that Figure 3J-L are based on spatial transcriptome.

Figure 3M: How are the pathway activities calculated? Which data is used? Do the densities show the ssGSEA score calculated for the gene expression in each each

hexagon on the spatial transcriptomics slide?

Response: We thank the reviewer for these comments. The pathway activity was calculated with ssGSEA algorithm (Line 1116-1117). As the reviewer described, the density plot shows ssGSEA score in each cluster on the spatial transcriptomics slide. We have added this statement to the figure caption now.

Figure 4

Figure 4A: Please remove all information that is not mentioned in the results section from the top figure block. Increasing the height of the bars that show the tumor subtypes can help as well.

Response: The reviewer's suggestions have been well taken. Information not mentioned in the results section has been removed. We have also increased the height of the bars to make it clear to the reader.

Figure 4B: A more common visualization of the enrichment results is to show the $-\log_{10}(\text{FDR})$ values as bars and color the bars based on the number of genes (or simply write the number of genes into each bar).

Response: We thank the reviewer for this suggestion. **Figure 4B** has been modified as the reviewer suggested. Row axis has been replaced by $-\log_{10}(\text{FDR})$ and the number of genes is labeled in each bar.

Figure 4C: It would be helpful for the reader, if the figure contains the information that Oil Red O stains lipid droplets or lipids.

Response: This statement has been added to the figure legend.

Figure 4D: It would be advantageous to either show $-\log_{10}(\text{p-values})$ or $-\log_{10}(\text{FDR})$ in both figures 4B and 4D.

Response: We thank the reviewer for this comment. The label of **Figure 4D** has been modified to be consistent with that of **Figure 4B**.

Figure 4F: Why does the figure only contain one value for each metabolite in each IM subgroup? Same comment as in figure 3F.

Response: We thank the reviewer for this comment. Similar to **Figure 3E**, this heatmap was visualized in a space saving pattern. We have added the result of wilcox test to the Figure to make the statistical significance clear.

Figure 4H: Why does the figure not show gene expression values in all samples as all previous figures? Can the overexpression of transporters in IM4 be supported by a statistical test?

Response: We thank the reviewer for this comment. Similar to Figure 3E and 4F, we used the mean expression value of transporters in each IM subtype to reduce the size of the heatmap. Adjusted p-value of each transporter has been added to the Figure to make the statistical significance clear.

Figure 4I: Which data was subjected to pathway enrichment analysis? What do the circle sizes visualize? Does a negative T-value imply downregulation? Were all samples mapping to one tumor subtype or NAT compared with all others?

Response: We thank the reviewer for informing us of the missing information. This analysis is also based on ssGSEA score. T-value shows the results from limma algorithm. Here we aimed to show differential pathways between each IM subgroup and NATs. It was performed by comparing one subtype to all other samples. A negative T-value implies downregulated pathway score. We have added this information to the figure caption now.

Figure 4K: Does this figure show TCGA KIRC data? Please add to figure legend and results section. Which comparisons do the p-values refer to? This could be visualized using curly brackets connecting the compared subtypes.

Response: We thank the reviewer for these suggestions. Figure 4K is based on TCGA KIRC dataset and it has been added to the figure caption now. The p-value in the initial version shows p-value of log-rank test while multi-comparison was not performed. Curly brackets have been added to this figure as the reviewer suggested now.

Figure 5

Figures 5A/D/G: The figure caption says ‘Expression were normalized into row z-score’, as this reviewer assumed for figures 2A and 4A as well (although the legend to figure 2A says that DEGs are shown).

Response: We thank the reviewer for this comment. We applied the same approach to transform expression values into z-scores in Figure 2A, 4A and 5A. The approach detail has been added now.

Figure 5B/E/H: Which comparisons do the p-values refer to?

Response: We thank the reviewer for this comment. The initial p values shown in Figure 5B/E/H were log-rank p that indicates overall difference between four IM subgroups. Results of pairwise comparison corrected by BH algorithm have been added now. The approach detail has also been added to the methods (Line 1278-1279).

Figure 6

Figure 6A: What is the IM4 score? How were the genes for the score selected?

Response: We thank the reviewer for this comment. IM4 score was calculated by ssGSEA algorithm using a self-defined gene list. We used genes significantly up-regulated in IM4 to calculate IM4 score in each sample. This approach has been described in the methods (Line 1324-1327).

Figure 6B: It should be stated that the DCCD score and IM4 score are two different labels for the same values, as this reviewer assumes.

Response: We thank the reviewer for helping us improve the clarity of our manuscript. DCCD score is calculated using IM4 score but not equal to IM4 score. In order to balance the influence of IM2 feature and IM4 feature in this index, we scale centered them and used (IM4 score)-(IM2 score) as the final DCCD score. This approached detail has been shown in methods (Line 1323-1333).

Figure 6B: Adding the IM2-like and IM4-like labels as headlines to the figure would be helpful.

Response: The label has been added as the reviewer suggested.

Figure 6C: It should be explicitly stated, that IM2- and IM4-like are defined by a positive or negative IM4-IM2 difference, as figure 6B implies.

Response: We thank the reviewer for this suggestion. We have documented that IM1 and IM3 samples were classified into IM4-like or IM2-like according to a positive or negative DCCD score now.

Figure 6C: Which comparison does the p-value refer to? Are all other comparisons not significant, as this reviewer assumes?

Response: We thank the reviewer for pointing out this issue. The initial p-value indicates overall difference between four IM subgroups. Results of pairwise comparison corrected by BH algorithm have been added now.

Figure 6D: Which data was used for this UMAP, the same single nucleus RNAseq data shown in figure 2E? Why do the cells cluster by sample and not by cell type? Does the figure show only one cell type, i.e. only the malignant cells shown in figure 2E, as figure 6F implies? Was the data reanalyzed under different parameters, using only the malignant cells?

Response: We thank the reviewer for pointing out these issues. Only malignant cells in Figure 2E were used here. Since malignant cells of solid tumors have high heterogeneity between different individuals, we used ‘pca’ instead of ‘harmony’ to reconstruct the UMAP in Figure 6D. That’s why the cells were clustered by sample. This strategy aimed to fully determine whether the DCCD process only happens in part of these tumors but could be obscured in ‘harmony’ corrected UMAP.

Figure 6M: What do the numbers how?

Response: We thank the reviewer for this comment. The number is the molecular weight of the metabolite. Since the spatial metabolome is based on MS1, we cannot know the accurate identity

of these metabolites. Instead, we can only know this molecule belongs to which metabolite class in this database. Hence, we colored part of these dots according to the class and labeled the molecular weight of Fatty acyls. It has been added to the figure legend.

Figure 6N: It would be good to show the same spectra in figure 6M and 6N as well as to round the m/z spectra to the same number of decimals as in figure 6M.

Response: We thank the reviewer for this suggestion. We have colored the metabolites in Figure 6N in red in Figure 6M to make it conspicuous now.

Figure 7

Figure 7B: Please explain the colors in the figure legend.

Response: This statement has been added and we have also added a label to the figure.

Figure 7C: Please summarize in the figure legend how the TFs were identified (e.g. differentially expressed TFs between IM4-like and IM2-like malignant cells and bulk transcriptomic samples).

Response: As suggested by the reviewer, we have updated the figure legend to describe how these TFs were identified.

Figure 7C: Which TF-target gene database was used for the analysis?

Response: We thank the reviewer for this comment. No TF-target gene database was used in this analysis. Figure 7C only shows the expression level of these TFs in snRNA-seq data. We have added this information to the figure legend now.

Figure 7D: Which sample is shown? Please also state that this is based on ssGSEA scores calculated from spatial transcriptomics data.

Response: We appreciate the reviewer for informing us of this missing information. Y7_T was used in this analysis. This figure shows the trajectory constructed by SPATA2 from the pre-defined IM2-like region to IM4-like region. The plot is colored by Seurat clusters and the arrow shows the transformation direction. We have added this statement to the figure legend now.

Figure 7E: Are these values also Z-scores? Why does the legend distinguish between high and low instead of showing Z-scores?

Response: We thank the reviewer for this comment. This heatmap was created automatically by plotTrajectoryHeatmap function of SPATA2. The expression values were also scaled into row z-score automatically. We have modified the figure legend now.

Figure 7F: What are DC1, 2 and 3?

Response: DC is the abbreviation of diffusion component. We have added the full name to the figure legend now.

Figure 7F: What do the arrows show? Why are they colored in blue (IM2) and red (IM1)?

Response: We thank the reviewer for this comment. Figure 7F shows the result of diffusion map of TCGA KIRC data. Based on that DCCD process from IM2 to IM4 is a continuous process, we hypothesized that a trajectory similar to that of scRNA-seq data could be constructed in TCGA KIRC data and then performed this analysis. The arrows were used to show the direction of transformation while the color may be confusing. We have modified the figure to make it clear.

Figure 7C/E/G: Since these figures all show Z-score normalized expression values of selected transcription factors (assumed by this reviewer), they should all be colored using the same color scheme, i.e. the one used in figure 7G that also agrees with the color scheme's used for the gene expression values in previous figures.

Response: Color of these heatmaps has been modified as the reviewer suggested. Re-drawing of Figure 7C resulted in a similar but not completely the same heatmap, which may be caused by the update of these packages.

Figure 7I: Please increase the text size. There is still space in the figure.

Response: The text has been enlarged as the reviewer suggested.

Figure 7J: Please add the sample names instead of the numbers to the figure.

Response: The figure has been modified as the reviewer suggested.

Reviewer #2:

Remarks to the Author:

The manuscript Intergrated Multi-omics Sequencing Reveals Metabolic Reprograms in the Progression of ccRCC by Hu et al provides a detailed multiomic analysis clear cell renal cell carcinoma. Although the entire study population is large (100 patients), only a subset of these went on to have a true multiomic examination. Having said that, this is the first paper that has combined all of these approaches. As a consequence, the dataset is unique and in itself valuable. The authors should be commended on this.

The analysis also appears sound in general, with the conclusions plausible and important. As with all large scale projects, it is impossible for reviewers to fully check the veracity of the conclusions. However, there are no obvious flaws in the approach.

The paper is in general well written and presented. There are numerous typographical errors which I shall not comprehensively discuss.

I have some specific questions which need addressing prior to publication, but otherwise feel that this paper is a welcome and significant addition to the field.

We thank the reviewer and appreciate the overall positive comments. We also appreciate the reviewer for providing this guidance to help us revise our manuscript. We have carefully

analyzed all of the reviewer's suggestions and indicated the changes we have made in the following point-to-point response.

1) Line 37 - define TJ-RCC

Response: TJ-RCC has been defined as 'RCC cohort of Tongji Hospital' now.

2) Line 42 - 'extremely' how is this defined?

Response: We thank the reviewer for this comment. The adjective may be not accurate. It is replaced by 'more' now.

3) Line 77 - 'mutation hot spots' - these are no hot spots per se, as they are all in tumour suppressor genes. Perhaps it would be more accurate to say enriched in driver genes.

Response: We agree with the reviewer that 'mutation hot spots' may be inaccurate here and have modified it to 'mutations enriched in driver genes'.

4) Line 79 - 'finding indicates that data obtained from clinical cohorts from different countries are necessary for understanding the heterogeneity of ccRCC'. Heterogeneity can mean many things, but for this I presume you are meaning geographically related intertumoural heterogeneity. Most RCC researchers use

heterogeneity to mean intra-tumoural het.

Response: We thank the reviewer for pointing out this issue. Here we aimed to describe the difference of AA-associated mutations between cohorts from different countries. We have added ‘geographically related’ to make this statement clear.

5) Line 92 - ‘hot spot’ again, another use of hot spot which is confusing.

Response: We thank the reviewer for this comment and have changed it to ‘focus’ now.

6) Line 108 - Table S1. It would be valuable to include the precise mutation in this table, eg genomic location, base change, etc.

Response: We thank the reviewer for this suggestion. The precise mutation in each sample has been added to the supplementary table now. Since some tumors contain multiple mutations in one gene, we added an independent table containing the information (Table S8).

7) Line 117 - COSMIC needs referencing

Response: We appreciate the reviewer for informing us of this missing reference. A reference has been added where we first referred to COSMIC.

8) Line 121 - 'In contrast, TTN, MUC16, MTOR, XIRP2, etc., were more frequently mutated in the AA group'. This doesn't mean anything in the context of an enhanced mutation rate due to AA exposure. Can you run dN/dS to see whether there might be additional driver events in this cohort?

Response: We thank the reviewer for this comment. Here we aimed to show that increased mutation rate caused by AA exposure only influence genes except commonly most frequently mutated genes since they didn't show a significant difference between AA and non-AA groups (Figure 1D). We agree with the reviewer that a higher mutation rate in these genes has only a little significance in the context of an enhanced mutation rate in the AA group. Therefore, we analyzed the percent of T>A signature in AA and non-AA groups to determine the effect of exposure to AA on causing these mutations (Figure 1E-1F). This analysis revealed that the additional mutation in the AA group mainly happens in genes except overall most frequently mutated genes in ccRCC. As the reviewer suggested, we have also tried to run dN/dS to confirm whether there are additional driver events in this cohort. We utilized a dN/dS method that considered the context of the mutation sequence, the gene sequence, and the variation of the mutation rate across genes¹. Only VHL, PBRM1, SETD2, BAP1, and TPTE show positive selection effects in this cohort (**Response Figure 5**). None of these mutations that occurred in the TPTE region were caused by T>A shift.

Response Figure 5: dS/dN analysis revealed 5 genes show positive selection effects in TJ-RCC.

Reference

1. Martincorena, I., *et al.* Universal Patterns of Selection in Cancer and Somatic Tissues. *Cell* **171**, 1029-1041.e1021 (2017).

9) Line 124 - can you define which 'mutation hub genes'. I presume this is established drivers, but you need to specify which you have included, rather than list a handful.

Response: The reviewer's suggestion has been well taken. We have clearly defined that the mutation hub genes are 5 most frequently mutated genes in ccRCC, including VHL, PBRM1, SETD2, BAP1 and KDM5C in the results (Line 126-127).

10) Line 125 - Please apply some statistics to test differences in these cohorts and consider using something other than a pie chart. For one, the values have been normalised which means that we lose data on the absolute values.

Response: We thank the reviewer for this suggestion and have added an additional boxplot to show the statistical significance (Figure 1F).

11) Line 127 - This is interesting and perplexing:

1) Many of the drivers "hub genes" are indels, can the authors reassure that these have not inadvertently been ascribed a mutational context.

Response: We thank the reviewer for the suggestion. We have rechecked the analysis pipeline. Since AA signature defined in COSMIC only considered SNA, indels were not considered in Figure 1E.

2) One interpretation of the data is that AA mutagenesis occurs once the tumours have formed with their respective genetic drivers. However, once tumours have formed, they would no longer act as a tubule or filtration unit of the kidney, and they would therefore presumably no longer be exposed to high levels of AA. How do the authors

explain this dichotomy?

Response: We appreciate the reviewer's thoughtful analysis of our data and the intriguing question they have brought up. The reviewer assumed that once tumors have formed, they would no longer act as a tubule or filtration unit of the kidney, and they would therefore presumably no longer be exposed to high levels of AA. We thought it is a reasonable hypothesis. The work of Zhao-Ning Lu et al.¹ has confirmed that the proportion of T>A signals in subclones is reduced in AA exposure-induced tumors, which may be due to lower levels of AA exposure in the central region of the tumor. However, even if this assumption is true, these mutations induced by AA that appear at an early stage of tumor formation should still be clonal rather than subclonal. Hence, we should still be able to find traces of AA signals in these tumors. Therefore, the low frequency of AA in driver mutations suggests that this exposure factor has a relatively low degree of involvement in the process of tumor formation. AA signature-enriched ccRCC may be discovered simply because AA-related mutation signals are widely distributed in the epithelial cells of East Asian populations. For example, Ruoyan Li et al. found a high frequency of T>A mutations in the normal urothelium of Chinese individuals². Meanwhile, considering the possibility raised by the reviewer, we think that discussing the progression of ccRCC here may be unreasonable as physical isolation leads to AA not being involved in progression. Therefore, we have made modifications to the text.

1. Lu, Z.N., *et al.* The Mutational Features of Aristolochic Acid-Induced Mouse and Human Liver Cancers. *Hepatology (Baltimore, Md.)* **71**, 929-942 (2020).
2. Li, R., *et al.* Macroscopic somatic clonal expansion in morphologically normal human urothelium. *Science (New York, N.Y.)* **370**, 82-89 (2020).

12) Line 148 LD needs defining

Response: We thank the reviewer for informing us of this missing issue. The full name of LD has been added now.

13) Line 161 - There is only a single dataset referenced here. Which are the others that were used - how did you select them?

Response: We thank the reviewer for informing us of the writing mistake. We only used one single-cell dataset and the 'datasets' has been modified to 'dataset' now. The single-cell sequencing dataset profiled by Aleksandar et al. contains ccRCC at different pathological stages and grades. We thought it could describe the overall feature of ccRCC. Therefore, we used this dataset to conduct this analysis.

14) Paragraph beginning line 167: This entire section needs careful rewording as often cells are described as being rare or enriched. As there is no spatial RNA/protein or morphology described here, the language needs toning down. One could say that RNA signals are present that might infer these cell types.

Response: We thank the reviewer for this suggestion and have carefully modified the statement to make it accurate. Considering the cell abundance used here is offered by ssGSEA algorithm, we replaced enrichment of cell types with increased cell score or signature to make it appropriate.

15) Line 203: How did you chose these 20 samples?

Response: The reviewer asked how we chose these 20 samples subjected to single nucleic profiling. We chose these 20 samples using randomized strategy. Firstly, we randomly selected 5 samples in each IM subtype. Since only 4 IM4 samples are available after multi-omics sequencing, we finally used 6 IM3 samples and 4 IM4 samples for subsequent profiling. Next, these 20 samples were randomized 1:1 into snRNA-seq group or 10X multiome group. This statement has now been added to the methods (Line 789-793).

16) Line 209: 'Subcluster analysis revealed that the distribution of immune cell clusters strongly discriminated across immune subtypes'. Evidence? stats?

Response: We thank the reviewer for this comment. Figure 2G visualized the odds ratio of immune cell subclusters between 4 IM subgroups. Fisher's exact test was performed and then corrected by the BH algorithm. $OR > 1.5$ commonly has an adjusted p-value < 0.05 . We have now added this information to Figure 2G and detailed in the methods (Line 859-866), and mentioned it in the figure legend now. We have also added the adjusted p-value to the heatmap now.

17) Line 227: Where is the evidence and stats that fibroblasts are enriched in this cluster?

Response: The reviewer requested the evidence and stats that fibroblasts are enriched in this cluster. As shown in Figure 2A, we could observe that the IM4 group has the highest fibroblast

score among 4 IM subtypes. We have added a boxplot now to make it clear (**Extended Data Fig. 3K**).

18_ Line 229: Did you look at other associations, for instance ACKR1 expressing endothelial cells and macrophage subsets for instance?

Response: We appreciate the suggestion provided by the reviewer. In the initial manuscript, we considered the endothelial cells as a whole in this analysis. According to the reviewer's suggestion, we separated vein (ACKR1+) and artery (GJA5+) endothelial cells to calculate the correlation between cell subgroups. Similar to SMCs, vein, and artery endothelial both show a positive correlation with *IGF1*+ macrophages and fibroblasts, while the correlation was a bit lower than that between *IGF1*+ macrophages and fibroblasts. We applied mFISH staining to confirm whether *IGF1*+ macrophages could be colocalized with veins and arteries. The staining show that veins are localized at the SMC-enriched region. It also explained why vein, artery, and SMC show a higher correlation in **Extended Data Fig. 3L**. These observations have been added to the results section (Line248-251) now.

19) Line 260 'In addition, that difference is dependent on stromal cell abundance and is weakly affected by molecular subtypes'. I don't see the evidence for this in the figures.

Response: We thank the reviewer for this comment. Via ssGSEA analysis we found that both Cluster 1 and Cluster 2 in **Extended Data Fig. 5A** were mainly derived from interstitial cells and endothelial cells. And the same cell group showed a similar expression pattern between 4 IM

subgroups (**Extended Data Fig. 5B**). The interpretation may be too assertive as that difference could still be observed between IM subgroups. We have now deleted this statement.

20) Line 434 I agree that the plot supports this, but Is there not a nicer way to plot this information? In this case, the location in the UMAP means nothing as most variance can be attributed to batch variance.

Response: We agree with the reviewer's opinion that Figure 6D&6E contains considerable batch effect. According to distinctive mutation and CNV patterns, solid tumor cells commonly show almost patient-specific features, which means they have strong collinearity with the batch effect. Since then, an accurate algorithm for removing batch effect between solid tumor cells is still a challenge. Based on the suggestions from the reviewer, we re-clustered the data after removing batch effects. The new UMAP clearly reflects the transition from IM2 to IM4, but it failed to show the proportion of tumor cells belonging to either IM2-like or IM4-like in each sample. We kept the original UMAP plot to demonstrate inter-sample differences and added a new UMAP plot (**Extended Data Fig. 9C-9D**) to illustrate the universality of this differentiation in ccRCC.

21) Line 444: 75% seems high to me rather than 'only'

Response: We agree with the reviewer's viewpoint and have modified this statement.

22) Line 447: where is the subclonal SCNA - it is not clear on this figure? If it is subtle, how would you know that the expression differences are not a consequence of

the IM2 to IM4 gradient rather than SCNA?

Response: We appreciate the reviewer's comment. Figure 6H shows the SCNA situation in sample X98_T. Compared with IM2-like cancer cells, IM4-like cells have a more obvious chr 9 loss, which was confirmed in the spatial transcriptome of the same sample (Figure 6K). In **Extended Data Fig. 9G**, IM4-like cells in sample Z43_T had higher chr12 gain and chr19 gain signature than IM2-like cells. Therefore, we described in the text that subclonal SCNA only occurred in two samples. We added this information to the figure legend to help readers quickly locate it.

The reviewer also asked how we would know that the expression differences are not a consequence of the IM2 to IM4 gradient rather than SCNA. According to our results, in large DNA sequencing datasets including TJ-RCC generated in this paper and TCGA-KIRC, IM4-like ccRCC showed a higher frequency of SCNA compared to IM2-like tumors, especially on chr 9 and chr 12. This suggests the possibility of SCNA involvement in this transformation process. However, this proportion does not exceed 50% in the IM4-like group. Through snRNA-seq results, we observed subclonal-level SCNA only in two partially DCCD-transformed samples. Among them, the spatial transcriptome of the X98_T slice clearly indicated that chr 9 loss occurred only in the region of IM4-like. These data suggest that there may be some connection between subclonal SCNA and DCCD transformation but SCNA is not necessary for DCCD. Therefore, in the text, we mentioned that "there is no absolute correlation between SCNA events and DCCD. DCCD may be induced by the processes involved in local TME remodeling, such as hypoxia." (Line 629-631).

Reviewer #3:

Remarks to the Author:

In this paper, the authors generated a multi-omic dataset from samples of a cohort of 100 ccRCC patients. They profiled the multi-omic samples, focusing on potential exposure to aristolochic acid and comparing tumor vs normal adjacent tissue. Then they defined a cell composition gene signature of ccRCC from a public single cell ccRCC dataset and used it to cluster the patients of their cohort into 4 groups. They further characterized the defining features of these 4 groups, at a pathway and clinical level. They determined that 2 of these subgroups, IM2 and IM4, were particularly associated with a cellular phenotype they coined De-clear cell differentiated cells (DCCD). They evaluated the relevance of this cellular phenotype with respect to various treatment strategies, and characterized some of the patient samples in more depth with single nuc RNA and visium slices. Finally, they studied the pseudotime progression of this cellular phenotype across patient samples.

Overall, this study provides an analysis of a large amount of multi-omic data on ccRCC samples, especially regarding Asian ccRCC patients. These can be important for the understanding of ccRCC and ultimately lead to treatment decisions that would rely more on molecular profiles of patients. It is however not our expertise, so we cannot make authoritative assessments of the relevance of the findings.

Further, we believe that there are several major points that are important to address.

We appreciate the reviewer for the overall positive comments on our work. We have carefully considered the suggestions of the reviewer and revised our manuscript. The point-to-point response has been addressed as follows:

Major comments

1) While extracting a gene signature representative of cell composition is a very interesting idea to cluster ccRCC patient samples, it is held back by the potential quality of this signature. Since almost the entire study relies on the definition of this signature, we would expect to see more evidence regarding the quality of this signature (rely heavily on Harmony performance and the subsequent cell type definition).

Response: We thank the reviewer for this comment. We agree with the reviewer's opinion that our analysis relied heavily on Harmony performance and the subsequent cell type definition. UMAP of single cells used in this analysis before and post harmony correction have been shown below (**Response Figure 6**). Harmony successfully removed the batch effect in this dataset. To avoid bias brought by the cell identification and make it easier to identify cell type-specific signatures in bulk-seq data, we only use cell identities at the major functional group level. For example, CD8+ T cells were only classified into CD8+ MAIT and CD8+ T cells due to innate and adaptive immunity nature. Commonly used markers confirmed that this identification is reliable (**Extended Data Fig. 2B-2C**).

Response Figure 6: UMAP of single cells using pca (left) and harmony (right).

2) It would be interesting to see how these findings would compare to results obtained with a gene signature obtained not by a union of the top marker genes of each cell type, but instead the top DEG (healthy vs tumor) for each cell type.

Response: We thank the reviewer for raising this interesting analysis strategy and have tried it as the reviewer described. Surprisingly, this strategy resulted in similar results as the approach of our paper (**Response Figure 7A-7B**). We thought it might be caused by that DEGs calculated in each cell type are still cell type-specific genes (**Response Figure 7C**). Since then, it still resulted in similar subgroups found in this work.

Response Figure 7: A: relationship between IM subgroups and subgroups identified by DEG approach; B: heatmap showing expression of 687 genes in DEG subgroups; C: heatmap showing the mean expression level of 687 genes in each cell subgroup in tumor or NATs. Expression levels were transformed into row z-scores.

3) The findings of this study are also heavily dependent on the division of the patient cohort into 4 subgroups. While the authors show that the choice of this number isn't entirely arbitrary, it still isn't clear how this number was the only obvious choice. If it was not, then the authors should provide insights into the robustness of their conclusion if this number would change.

Response: The reviewer expressed the concern that the choice of the number of IM clusters isn't entirely arbitrary. The number of subgroups was chosen based on the delta area under CDF plot as shown below. With $k > 4$, the delta area only showed a slight change. Therefore, we finally classified 100 ccRCC samples into 4 subgroups (Line 1256-1257).

Response Figure 8: Scree plot showing the delta area of each k-value from ConsensusClusterPlus 2.

4) Same this as 4), but regarding the choice of 911 genes for the signature, based on the top 60 for each cell type. What happens if a number other than top 60 is chosen ?

Response: We thank the reviewer for this comment. To examine whether the number of the top markers used would influence the result of sub-clustering of tumors, we tried to use top 40 and 100 markers to perform the same analysis. The results show that the number of the top markers

only slightly influenced the results. When we use top 40 markers, IM1 and IM4 could not be separated with $k=4$. The pipeline using top 40 genes needs $k=8$ to distinguish IM1 from IM4. This result indicates that classification using this pipeline is not sensitive to the number of top markers selected from scRNA-seq data.

Response Figure 9: Relationship between subtypes classified by different top gene cutoff or k value.

5) Instead of defining the IM2-like and IM4 like signatures (which are supposed to represent low and high DCCD), it would improve the clarity of the study a lot to instead define a single signature directly representative of the DCCD phenotype that can be used to score samples.

Response: We thank the reviewer for the suggestion. We agree with the reviewer that a single signature to identify the DCCD subtype could be clearer and more user-friendly. We defined a DCCD score as the D-value of the IM4 score and IM2 score (Line 1323-1333). We also visualized this score in the updated Figure 6B. The DCCD score is highly positively correlated to the IM4 score, and it could help score single samples rapidly.

6) The integrated metabolic analysis feels very arbitrary with respect to the enzymatic pathways that were highlighted. For example, why isn't branched chain amino acid metabolism considered, even though it is among the most deregulated metabolic pathways highlighted in this study and many others on ccRCC?.

Response: We thank the reviewer for these comments. In previous clinical metabolomics profiling of ccRCC, Hakimi et al.¹ demonstrated that expression levels of enzymes were weakly correlated with the levels of metabolites in ccRCC. In Figure 1J, we aimed to show the overall change of key metabolic processes (glycolysis, TCA cycle, and fatty acid metabolism) at transcriptomics, proteomics, and metabolomics level. Since arginine biosynthesis was deregulated in ccRCC and was found to be associated with DCCD in this work, we also added urea cycle to Figure 1J. We agree with the reviewer that additional visualization of changes in top deregulated metabolic pathway is necessary and has added a visualization of galactose metabolism pathway in **Extended Data Fig. 1H**.

7) Make the code available for reviewers to check for clarity and reproducibility. The provided repository only contains a small subset of scripts.

Response: We thank the reviewer for helping us improve the clarity and reproducibility of our manuscript. The code uploaded to GitHub has been modified and completed now.

8) Make the data easier accessible for reviewers and eventually readers. the Mendeley link does not work, even after registering into Mendely with an account. We suggest to use rather a repository like Zenodo.

Response: We thank the reviewer for this suggestion. We have now uploaded the processed data to Zenodo (<https://zenodo.org/record/8063124>) and the link to Mendely has also been modified and made publicly available now.

9) There are other large multi-omic characterizations of ccRCC cohorts, and this study should be place in the context of those, finding compared, etc.

Response: The reviewer requested "There are other large multi-omic characterizations of RCC cohorts, and this study should be placed in the context of those, finding compared, etc.". we thank and agree with the reviewer for this comment. According to the reviewer's suggestion, we added this information to the context and compared them.

1. Various large clinical sequencing cohorts revealed that VHL, PBRM1, SETD2, BAP1 and KDM5C are the most commonly mutated genes in ccRCC and a similar result was found in our cohort (Line 111-112). Via NMF clustering of mutation signatures, we discovered an AA-enriched ccRCC subgroup, which was also found in other Asian cohorts. Hence, we pooled them with previous Asian cohorts to investigate the features of these tumors (Line 114-131).

2. Previous studies have characterized ccRCC tumors based on their specific genomic alterations, leading to their classification as a metabolic disease¹. Profiling their alteration at transcriptomic and proteomic levels lead to distinct result from that of metabolomics²⁻³. Therefore, we performed RNA-seq, global proteomics and non-targeted metabolomics in this cohort to investigate these changes and the relationship between them (Line 149-152).

3. Previous multi-omics study has identified distinct microenvironmental subtypes in clear cell renal cell carcinoma (ccRCC) that are associated with different clinical outcomes². It was defined using xCell deconvolution of bulk-seq data based on a reference matrix of cell-type-specific gene expression signatures provided by the ENCODE project, which may lose some ccRCC-specific

signals. Here, we classified the immune microenvironment of ccRCC using a signal matrix derived from single-cell RNA sequencing and identified four distinct IM subtypes (Line 163-177). We found that IM3, previously characterized as having the highest CD8+ T-cell infiltration, did not exhibit the worst clinical outcome (Line 211-214). Furthermore, we identified a subtype of ccRCC with poor prognosis and distinctive metabolic features.

4. Recent multi-omic study (CPTAC, phase 2) performed integrative histopathologic, proteogenomic, and metabolomic analyses on large ccRCC cohorts⁴. By combining histologic and molecular profiles, they discovered that 90% of ccRCCs exhibit intratumor heterogeneity (ITH), with 50% showing ITH in immune signatures. They also discussed the metabolomic alterations between low-grade tumors and high-grade tumors. In this work, we discussed the metabolomic differences between IM subgroups and identified a distinct subgroup named as DCCD-ccRCC. We also performed spatial metabolomics profiling on part of these samples and revealed that partial metabolic reprogramming serves as part of the ITH in ccRCC (Line 597-605).

Reference

1. Wettersten, H.I., Aboud, O.A., Lara, P.N., Jr. & Weiss, R.H. Metabolic reprogramming in clear cell renal cell carcinoma. *Nature reviews. Nephrology* **13**, 410-419 (2017).
2. Clark, D.J., *et al.* Integrated Proteogenomic Characterization of Clear Cell Renal Cell Carcinoma. *Cell* **179**, 964-983.e931 (2019).
3. Hakimi, A.A., *et al.* An Integrated Metabolic Atlas of Clear Cell Renal Cell Carcinoma. *Cancer cell* **29**, 104-116 (2016).
4. Li, Y., *et al.* Histopathologic and proteogenomic heterogeneity reveals features of clear cell renal cell carcinoma aggressiveness. *Cancer cell* **41**, 139-163.e117 (2023).

Minor comments

1) An enrichment score does not represent activity per se. The authors often refer to a metabolic pathway enrichment score as a metabolic pathway activity. This is a semantic shortcut that changes the meaning and can be misleading.

Response: We appreciate the reviewer for informing us of this misleading statement. We have now replaced them with ‘pathway score’ or ‘signature’.

2) Figure S2B: why were those markers chosen, provide more insights into which markers validate which previous claim.

Response: We thank the reviewer for helping us improve the clarity of our manuscript. These markers are commonly used to mark specific cell types in previous studies. We have now added the statement to the figure legend and added missing reference to the methods (Line 1290-1292).

3) It isn't clear why the authors claim that the urea cycle disorder is not dependent on the dysregulation of the enzymes.

Response: The reviewer asked why we claimed that the urea cycle disorder is not dependent on the dysregulation of the enzymes. When comparing tumor samples to NATs, we found that metabolites in the urea cycle are decreased along with the enzymes. However, when comparing IM4 to other tumors, we found that although IM4 contains fewer metabolites of the urea cycle, the expression of the enzymes was not changed at both transcriptomics and proteomics levels.

The initial statement may be confusing and we have added ‘enhanced’ to make it clear what we aimed to express (Line 366).

4) The comparison between the different survivability curves for different treatment combinations should be made clearer, regarding the figures. It’s hard to compare them by looking at them side by side. A statistical test should be made to formally assess the differences.

Response: The reviewer’s suggestion has been well taken. We have added the results of the pairwise comparison of each Kaplan-Meier test to the plot (**Figure 4K, 5B,5E,5H & 6C; Extended Data Figure 3D, 7C, 7E, 8A, 8C & 8E**).

5) Figure 6H: the piece with a high IM2 score is very small on the slice.

Response: We thank the reviewer for pointing out this issue. **Figure 6H** showed the results from snRNA seq data from another piece adjacent to the slice subjected to the spatial transcriptome profiling. The result of SCNA analysis of the spatial transcriptomics was shown in **Figure 6K & 6L**. We have now pointed out which omic is used in each figure in the figure legend to make it clear.

6) Line 469: Please clarify in which section it is detailed.

Response: We thank the reviewer for this suggestion and have clarified it now (Line 527).

7) Line 475: I don' t understand what the authors mean by motifs here.

Response: We thank the reviewer for this comment and agree with the reviewer that 'motif' is not an accurate statement here. We have now replaced it with TFs

8) "Cell type identification was performed based on markers from single cell atlas of human kidney" Shouldn' t authors instead use human cancer kidney markers? from cohort authors poicked 20 for snRNA and 10 for snATAC - why those number,s and how were those chosen?

Response: We thank the reviewer for this comment. The reviewer mentioned that we documented that cell type identification was performed based on markers from single cell. It described the process to identify cell types in 2 NATs sequenced in this work and displayed in **Extended Data Fig. 6**. The process for tumor samples was based on single cell atlas of ccRCC and was detailed in Line 846-847.

The reviewer also asked why and how we chose 20 tumors to perform snRNA and snATAC sequencing. We chose to sequence 20 samples based on researches with a similar design and it is also limited by the sample available¹ after 4 bulk omics sequencing. These 20 samples were chosen following a randomized strategy. Firstly, we aimed to randomly choose 5 samples from each IM subtype. Since only 4 IM4 samples are available after multi-omics sequencing, we finally used 6 IM3 samples and 4 IM4 samples for subsequent profiling. Next, these 20 samples were randomized 1:1 into snRNA-seq group or 10X multiome group. This statement has now been added to the methods (Line 789-793).

Reference

1. Wang, L.B., *et al.* Proteogenomic and metabolomic characterization of human glioblastoma. *Cancer cell* **39**, 509-528.e520 (2021).

10) SCENIC regulons are normally built in a bespoke manner for each context, so using the ones in the server are not the best (“downloaded a TF list containing all the already known human TFs from github 807 (<https://github.com/aertslab/SCENIC>). ”

Furthermore, now there is the chance to use ATAC + RNA to build regulons with SCENIC+ and other tools of other groups.

Response: We thank the reviewer for this comment. We agree with the reviewer that building regulons with snATAC data is a better choice in this work. TFs listed by SCENIC were only used to extract TFs from snRNA-seq, spatial transcriptome, and bulk-seq data to perform the trajectory analysis (Figure 7C,7E&7G). When processing snATAC data, we performed chromVAR analysis to calculate the motif activity (Line 562-564). We also performed footprinting analysis on snATAC data to identify differential regulons between IM2-like and IM4-like tumor cells (Line 566).

Decision Letter, first revision:

11th Aug 2023

Dear Professor Chen,

Your Article, "Integrated Multi-omics Sequencing Reveals Metabolic Reprograms in the Progression of ccRCC" has now been seen by your 3 original referees. You will see from their comments below that while they find your work of interest, some important points are raised. We are interested in the possibility of publishing your study in Nature Genetics, but would like to consider your response to these concerns in the form of a revised manuscript before we make a final decision on publication.

We therefore invite you to revise your manuscript taking into account all reviewer and editor comments. Please highlight all changes in the manuscript text file. At this stage we will need you to upload a copy of the manuscript in MS Word .docx or similar editable format.

*2) If you have not done so already please begin to revise your manuscript so that it conforms to our Article format instructions, available [here](http://www.nature.com/ng/authors/article_types/index.html). Refer also to any guidelines provided in this letter.

[redacted]

We hope to receive your revised manuscript within four to eight weeks. If you cannot send it within this time, please let us know.

Nature Genetics is committed to improving transparency in authorship. As part of our efforts in this direction, we are now requesting that all authors identified as 'corresponding author' on published papers create and link their Open Researcher and Contributor Identifier (ORCID) with their account on the Manuscript Tracking System (MTS), prior to acceptance. ORCID helps the scientific community achieve unambiguous attribution of all scholarly contributions. You can create and link your ORCID

from the home page of the MTS by clicking on 'Modify my Springer Nature account'. For more information please visit www.springernature.com/orcid.

Sincerely,

Safia Danovi
Editor
Nature Genetics

Reviewers' Comments:

Reviewer #1:

Remarks to the Author:

I am still convinced that this study will have a high impact on the field. The authors addressed most of my suggestions. I have only a few suggestions left.

Major comments

Contradictory observations for risk associated with pyrimidine metabolism and T-cell infiltration As describe in my initial review, the authors document that IM3 samples can be sufficiently distinguished from the other subtypes by their increased nucleotide metabolism activity. As figure 3A suggests and extended figure 5C documents, samples of the IM3 subtypes show a much higher expression of pyrimidine metabolism than all other samples. The authors document that there is a significant difference in survival rates for patients within the top and bottom 50% of pyrimidine activity (3C). One of the main statements in the study is that the prognosis is subtype dependent. Simplified, the prognosis is the best for subtype IM2, is worse for subtypes IM1 and IM3 and the worst for subtype IM4. As explained in my first review, these results seem to contradict each other. The authors have addressed my concern, but more discussion is needed.

I wonder if the correlation between poor prognosis and high pyrimidine metabolism activity simply reflects the effect of comparing IM2 vs IM4 patients as part of a low pyrimidine metabolism group IM2+IM1 and a high pyrimidine metabolism group IM3+IM4. In the IM4 group, the higher pyrimidine metabolism might even be remaining pyrimidine metabolism that will decline even further with further disease progression.

The authors discuss a similar observation at a later stage in their manuscript. Loss of metabolic activity including loss of fatty acid biosynthesis is observed in high-risk IM4 subtypes, although previous studies document worse prognosis for ccRCC with upregulated fatty acid synthesis (Yong et al 2020, Nat Rev Nephrol.).

Similarly, the authors challenge the risk evaluation via tumor stages. Treatment depends on the stage that correlates with the prognosis. Based on the author's finding the correlation between stage and prognosis could be influenced by more DCCD tumors in late than in early stages, so that prognosis prediction would profit from a more accurate classification.

In their response to the editor, the authors point at a similar explanation for the prognostic value of CD8 T-cell infiltration. T-cells infiltrate the tumor to restrict its growth, in agreement with the observation that the prognosis in IM3 subtypes with more T-cells is better than in IM4 subtype with

less T-cells. Nevertheless, the need for immunological restriction of tumor growth is only given at a particular stage, so that the low risk subtype IM2 does not lead to the recruitment of 'protective' T-cells. From this point of view, T-cell infiltration in IM3 subtype might mitigate the poor prognosis to the lower level also observed in IM1 subtypes. IM4 subtypes might simply not have enough or enough effective T-cells for a sufficient restriction of tumor growth and risk mitigation. Simplified, this explanation also states that the correlation of T-cell infiltration and poor prognosis arises from an unfortunate grouping of low risk IM2 and medium risk IM1 in one group and medium risk IM3 and high risk IM4 into the other group.

The authors should clearly address these thoughts in their manuscript, either when presenting the results in the results section or later in the discussion. The stated positive correlation between pyrimidine or T-cell infiltration and poor prognosis implies that a lower/higher pyrimidine metabolic activity of T-cell infiltration is always associated with a better/worse prognosis, which is not true based on the authors findings.

Figure 3B: Why are there only a few colored dots for the selected pathways, although there are ~25 samples of the IM3 subtype?

Minor comments

General comment: Please see below some suggestions that could be added to the figure legends for an easier understanding. Though some of suggested additions might explain standards that are self-evident for researchers in the field, they can still help researchers of different disciplines to faster understand what is visualized.

Figure 1A: Please add the abbreviations to the figure legend.

Figure 1B: Please add that the entries in each column show the genomic profiles for each of the 100 TJ-RCC tumors.

Figure 1C: Please add the full name for SNA.

Figure 1D: Please explain that the p-values show the enrichment of a mutation within indicated genes in samples of the AA tumor type, if compared to Non-AA tumor type.

Figure 1F: It is not clear for which p-values the Wilcox test and the student's t-test were used.

Figure 2A: Does the figure show the z-scores of the absolute expression of those genes (i.e., in the TPA matrix) identified to be DEGs in at least one IM subtype, if compared to all others?

Figure 2G; Please use 'Odd's Ratio (OR)' in the figure legend.

Figure 3A: Please indicate that "TPM matrix" contained the normalized read counts.

Figure 3E: Maybe, writing that "P values were calculated via comparing all expression values of each metabolite in a subgroup to all other values of that metabolite" would be more accurate.

Figure 4A: The legend says that Z-scores were calculated based on mean value and standard deviation of the ssGSEA scores across 100 tumors and NATs. Is this correct? I would guess that mean values and standard deviations were calculated based on the TPM matrix considering only those genes that were differentially expressed in DCCD tumors.

Figure 4D/E: Please explain what the x-axis shows.

Figure 4F: Please also describe how z-scores were calculated (similar to Figure 3E).

Figure 4H: Please see comment to Figure 3E about the p-value calculation.

Figure 4K: Please use Kaplan-Meier (KM) plot.

Figure 4K: Please add that the p-value was calculated by log-rank algorithm.

Figure 4K/5B/5E/5H: Please explain the meaning of the p-values calculated by log-rank algorithm that are visualized in the figures?

Figure 5: Please add that the numbers next to the forrest plot indicate number of participants in each group.

Figure 6L: Please add that this figure shows the chromosome 9 score in the visum slide shown in J.

Figure 6M: Please shift the M higher, next to the legend of the figure.

Figure 3A/D: It would be nicer, if the IM subtype colors are the same over all figures.

Figure 6I: Please describe the experimental method that was used to generate the data shown in this figure. Adding TN5 transposase to the figure helps understanding for readers who are not familiar with ATAC-seq.

Figure 6M: Please briefly describe what the predictive score is.

Extended data figure 1A: Please move the legend to the top of the figure here as well.

Extended data figure 2A: Please explain that the signature matrix shows the marker genes identified in the snRNAseq data. What does the consensus matrix show?

Extended data figure 2F: Please add that ssGSEA scores of the samples of one IM subtype versus all other subtypes were compared.

Extended data figure 3D: Please add that the p-values within the plots were calculated by log-rank test and briefly describe their meanings.

Extended data figure 3G: Please write Odd's Ratio.

Extended data figures 3H-M: Please check, if the legends were added to the correct subfigures (e.g. legend J seems to describe subfigure K).

Extended data figure 5A: If each row shows the ssGSEA values for one metabolic GO term, what are the enriched GO terms? Are these those terms with the highest T-values between a subtype of interest and all other subtypes?

Extended figure 7C/7E/8C/8E: Please add that the shown p-value was calculated by log-rank test and describe what it means.

Extended figure 8B/8D/8F/8G: Please add that the numbers document the number of samples in each group, for researchers who are not familiar with the Forrest plot visualization.

Extended figure 5L: What does LC stand for?

Extended data figure 10C: What are peak-to-gene links?

Extended data figure 10D: Please add that the gene scores were predicted from the snATAC data.

Extended data figure 10F: What is scMEGA?

Methods – line 796: Please replace dyeing by dying.

Reviewer #2:

Remarks to the Author:

Many thanks to the authors for their response. In general, all of my comments have been addressed, apart from the issue below. There remains multiple typographical errors.

Line 121-122: "In contrast, TTN, MUC16, MTOR, XIRP2, etc., were more frequently mutated in the AA group" I still don't understand why this is worthy of comment. What is the mechanism that makes this important? Given the lack of evidence that these are driver events (as per dN/dS in your response), they are likely to be passenger events, which as we know, affect the ENTIRE genome equally. These genes are likely to have higher absolute numbers of mutations because they are longer in size, rather than any interesting biological explanation. I suggest you remove the statement as it is misleading to readers.

Reviewer #3:

Remarks to the Author:

The authors addressed our comments fairly convincingly, except for one of the most crucial one:.

The code of the analysis absolutely not on par with modern code reproducibility standards (https://github.com/AndersonHu85/ccRCC_multiomics)

All the scripts are dumped in a single folder without any instructions. The scripts even lack to proper file extension (.R). The scripts content themselves rely on source files that are still not provided. None of them are executable.

Reproducibility is at the core of science and this works falls very short on that.

Author Rebuttal, first revision:

Reviewer #1:

Remarks to the Author:

I am still convinced that this study will have a high impact on the field. The authors addressed most of my suggestions. I have only a few suggestions left.

Major comments

Contradictory observations for risk associated with pyrimidine metabolism and T-cell infiltration As describe in my initial review, the authors Adocument that IM3 samples can be sufficiently distinguished from the other subtypes by their increased nucleotide metabolism activity. As figure 3A suggests and extended figure 5C documents, samples of the IM3 subtypes show a much higher expression of pyrimidine metabolism than all other samples. The authors document that there is a significant difference in survival rates for patients within the top and bottom 50% of pyrimidine activity (3C). One of the main statements in the study is that the prognosis is subtype dependent. Simplified, the prognosis is the best for subtype IM2, is worse for subtypes IM1 and IM3 and the worst for subtype IM4. As explained in my first review, these results seem to contradict each other. The authors have addressed my concern, but more discussion is needed.

I wonder if the correlation between poor prognosis and high pyrimidine metabolism activity simply reflects the effect of comparing IM2 vs IM4 patients as part of a low pyrimidine metabolism group IM2+IM1 and a high pyrimidine metabolism group IM3+IM4. In the IM4 group, the higher pyrimidine metabolism might even be remaining pyrimidine metabolism that will decline even further with further disease progression.

The authors discuss a similar observation at a later stage in their manuscript. Loss of metabolic activity including loss of fatty acid biosynthesis is observed in high-risk IM4 subtypes, although

previous studies document worse prognosis for ccRCC with upregulated fatty acid synthesis (Yong et al 2020, Nat Rev Nephrol.).

Similarly, the authors challenge the risk evaluation via tumor stages. Treatment depends on the stage that correlates with the prognosis. Based on the author's finding the correlation between stage and prognosis could be influenced by more DCCD tumors in late than in early stages, so that prognosis prediction would profit from a more accurate classification. In their response to the editor, the authors point at a similar explanation for the prognostic value of CD8 T-cell infiltration. T-cells infiltrate the tumor to restrict its growth, in agreement with the observation that the prognosis in IM3 subtypes with more T-cells is better than in IM4 subtype with less T-cells. Nevertheless, the need for immunological restriction of tumor growth is only given at a particular stage, so that the low risk subtype IM2 does not lead to the recruitment of 'protective' T-cells. From this point of view, T-cell infiltration in IM3 subtype might mitigate the poor prognosis to the lower level also observed in IM1 subtypes. IM4 subtypes might simply not have enough or enough effective T-cells for a sufficient restriction of tumor growth and risk mitigation. Simplified, this explanation also states that the correlation of T-cell infiltration and poor prognosis arises from an unfortunate grouping of low risk IM2 and medium risk IM1 in one group and medium risk IM3 and high risk IM4 into the other group. The authors should clearly address these thoughts in their manuscript, either when presenting the results in the results section or later in the discussion. The stated positive correlation between pyrimidine or T-cell infiltration and poor prognosis implies that a lower/higher pyrimidine metabolic activity of T-cell infiltration is always associated with a better/worse prognosis, which is not true based on the authors findings.

Response: We thank the reviewer for the thoughtful comments and observations. We agree that the apparent contradictions you've highlighted warrant a more nuanced discussion in our manuscript.

The reviewer has provided a highly reasonable and comprehensive explanation for the relationship between pyrimidine metabolism and T-cell infiltration. The observation that the association of enhanced pyrimidine metabolic activity with unfavorable prognosis is mainly due to the absence of T-cell infiltration in better prognosis subtypes such as IM2. As the reviewer

suggested, this could be due to the inability of ccRCC at this stage to recruit T-cells. The high-risk IM4 subtype, which lacks protective T-cell infiltration, exhibits a worse prognosis than the IM3 subtype. Therefore, it is not appropriate to simply associate high pyrimidine metabolic activity with poor prognosis. We fully agree with the reviewer's perspective. In accordance with the reviewer's recommendation, we have added a section to the discussion in our manuscript to thoroughly explore the relationships between these phenomena.

Figure 3B: Why are there only a few colored dots for the selected pathways, although there are ~25 samples of the IM3 subtype?

Response: We thank the reviewer for pointing out this issue. **Figure 3B** visualized the relationship between transcriptome and proteome. Firstly, we performed differential analysis on two datasets respectively to obtain differentially expressed pathway scores between IM3 and all other tumor samples. Then, we used T-value from the differential analysis of each pathway to investigate relationship between different omics. Since then, each dot in **Figure 3B** represents a single pathway, not a tumor sample. Pathways related to pyrimidine, cytidine and nucleoside metabolism was colored. We have included the relevant data in **Table S7**. We have also added a statement to the Figure caption to make it clearer to the reader.

Minor comments

General comment: Please see below some suggestions that could be added to the figure legends for an easier understanding. Though some of suggested additions might explain standards that are self-evident for researchers in the field, they can still help researchers of different disciplines to faster understand what is visualized.

Figure 1A: Please add the abbreviations to the figure legend.

Response: Abbreviations have been added to the figure legend now.

Figure 1B: Please add that the entries in each column show the genomic profiles for each of the 100 TJ-RCC tumors.

Response: We thank the reviewer for this suggestion. The statement has been added now.

Figure 1C: Please add the full name for SNA.

Response: Full name of SNA has been added to the ylab of **Figure 1C**.

Figure 1D: Please explain that the p-values show the enrichment of a mutation within indicated genes in samples of the AA tumor type, if compared to Non-AA tumor type.

Response: We have taken the reviewer's suggestion and added this statement to the figure legend.

Figure 1F: It is not clear for which p-values the Wilcox test and the student's t-test were used.

Response: We thank the reviewer for pointing out this issue. P values were calculated by two side student's t test here. We have modified the writing mistake now.

Figure 2A: Does the figure show the z-scores of the absolute expression of those genes (i.e., in the TPA matrix) identified to be DEGs in at least one IM subtype, if compared to all others?

Response: This Figure showed the z-score transformed absolute expression of those DEGs as the reviewer described. We have modified the statement to make it clear now.

Figure 2G; Please use ‘Odd’ s Ratio (OR)’ in the figure legend.

Response: We thank the reviewer for this suggestion. Full name of OR has been added now.

Figure 3A: Please indicate that “TPM matrix” contained the normalized read counts.

Response: The statement has been added now.

Figure 3E: Maybe, writing that “P values were calculated via comparing all expression values of each metabolite in a subgroup to all other values of that metabolite” would be more accurate.

Response: The expression has been modified as the reviewer suggested.

Figure 4A: The legend says that Z-scores were calculated based on mean value and standard deviation of the ssGSEA scores across 100 tumors and NATs. Is this correct? I would guess that mean values and standard deviations were calculated based on the TPM matrix considering only those genes that were differentially expressed in DCCD tumors.

Response: We thank the reviewer for pointing out this writing mistake. Z-score was transformed from TPM matrix in this figure. It has been corrected now.

Figure 4D/E: Please explain what the x-axis shows.

Response: X-axis represented the pathway impact calculated by MetaboAnalyst. It has been added to the figure legend now

Figure 4F: Please also describe how z-scores were calculated (similar to Figure 3E).

Response: The reviewer's suggestion has been well taken. This description has been added now.

Figure 4H: Please see comment to Figure 3E about the p-value calculation.

Response: We thank the reviewer for this suggestion and have modified the statement.

Figure 4K: Please use Kaplan-Meier (KM) plot.

Response: Full name of KM has been added now.

Figure 4K: Please add that the p-value was calculated by log-rank algorithm.

Response: Method detail has been added as the reviewer suggested now.

Figure 4K/5B/5E/5H: Please explain the meaning of the p-values calculated by log-rank algorithm that are visualized in the figures?

Response: We thank the reviewer for this suggestion. The p-values calculated by log-rank algorithm represented the global difference between tumor subgroups. The statement has been added to these figures, respectively.

Figure 5: Please add that the numbers next to the forrest plot indicate number of participants in each group.

Response: We have added this statement as the reviewer suggested.

Figure 6L: Please add that this figure shows the chromosome 9 score in the visum slide shown in J.

Response: The figure caption has been modified as the reviewer suggested.

Figure 6M: Please shift the M higher, next to the legend of the figure.

Response: We have adjusted the figure following the reviewer's recommendation.

Figure 3A/D: It would be nicer, if the IM subtype colors are the same over all figures.

Response: We thank the reviewer for this suggestion. The IM subtype colors have been made the same over all figures now.

Figure 6I: Please describe the experimental method that was used to generate the data shown in this figure. Adding TN5 transposase to the figure helps understanding for readers who are not familiar with ATAC-seq.

Response: Method used to generate this figure has been mentioned in the figure caption and Line 919-921. TN5 transposase has also been added as the reviewer suggested.

Figure 6M: Please briefly describe what the predictive score is.

Response: The meaning of predicted score has been added to the figure caption now.

Extended data figure 1A: Please move the legend to the top of the figure here as well.

Response: The legend has been moved to the top now.

Extended data figure 2A: Please explain that the signature matrix shows the marker genes identified in the snRNAseq data. What does the consensus matrix show?

Response: The explanation has been added as the reviewer suggested. The consensus matrix showed result of ConsensusClusterPlus when $k=4$. We have also added this statement here.

Extended data figure 2F: Please add that ssGSEA scores of the samples of one IM subtype versus all other subtypes were compared.

Response: We thank the reviewer for this suggestion. The statement has been modified now.

Extended data figure 3D: Please add that the p-values within the plots were calculated by log-rank test and briefly describe their meanings.

Response: We have added that the global p-value was calculated by log-rank algorithm in the figure legend now.

Extended data figure 3G: Please write Odds Ratio.

Response: Full name of OR has been added now.

Extended data figures 3H-M: Please check, if the legends were added to the correct subfigures (e.g. legend J seems to describe subfigure K).

Response: We thank the reviewer for informing us this mistake. The figure legends for 3J and 3K were previously in the wrong order. This has now been corrected.

Extended data figure 5A: If each row shows the ssGSEA values for one metabolic GO term, what are the enriched GO terms? Are these those terms with the highest T-values between a subtype of interest and all other subtypes?

Response: We are grateful for the issues identified by the reviewer. The description of the “Enriched GO terms” might have been imprecise. The labels on the right side of the heatmap represented high-frequency keywords in each module, and we have made a correction accordingly. The heatmap displayed GO terms with significant differences among the IM subgroups, and relevant explanations have now been added to the figure legend.

Extended figure 7C/7E/8C/8E: Please add that the shown p-value was calculated by log-rank test and describe what it means.

Response: We thank the reviewer for this suggestion. This statement has been added to the figure legends now.

Extended figure 8B/8D/8F/8G: Please add that the numbers document the number of samples in each group, for researchers who are not familiar with the Forrest plot visualization.

Response: The description has been added as the reviewer suggested.

Extended figure 5L: What does LC stand for?

Response: We thank the reviewer for this comment. LC-pos represents positive mode LC-MS and LC-neg represents negative mode LC-MS. The full names have been added now.

Extended data figure 10C: What are peak-to-gene links?

Response: We thank the reviewer for pointing out this issue. Peak-to-gene links represented correlations between peak accessibility in snATAC data and gene expression in paired snRNA data. It has been added to the figure legend now.

Extended data figure 10D: Please add that the gene scores were predicted from the snATAC data.

Response: The statement has been added as the reviewer suggested.

Extended data figure 10F: What is scMEGA?

Response: We thank the reviewer for this comment. ScMEGA is the abbreviation of Single-cell multiomic enhancer-based gene regulatory network inference. It is a pipeline designed to infer gene regulatory network by using single cell multi-omics data. We have added the full name to the figure legend and briefly introduced this pipeline in the methods (Line 922-923) now.

Methods - line 796: Please replace dyeing by dying.

Response: We thank the reviewer for informing us this writing mistake. It has been corrected now.

Reviewer #2:

Remarks to the Author:

Many thanks to the authors for their response. In general, all of my comments have been addressed, apart from the issue below. There remains multiple typographical errors.

Line 121-122: "In contrast, TTN, MUC16, MTOR, XIRP2, etc., were more frequently mutated in the AA group" I still don't understand why this is worthy of comment. What is the mechanism that makes this important? Given the lack of evidence that these are driver events (as per dN/dS in your response), they are likely to be passenger events, which as we know, affect the ENTIRE genome equally. These genes are likely to have higher absolute numbers of mutations because they are longer in size, rather than any interesting biological explanation. I suggest you remove the statement as it is misleading to readers.

Response: We thank the reviewer for the effort on helping us improving our manuscript and for pointing out the potential misunderstanding in lines 121-122. We agree that, in the absence of further evidence, the mention of these specific genes can be misleading and may incorrectly suggest a biologically significant pattern where one may not exist.

We understand your point about the size bias in mutation counts, where larger genes naturally tend to accumulate more mutations simply because of their length. In the context of our study, the significance of these mutations is not yet clear and, as you noted, they might be passenger events rather than driver events. Therefore, we have decided to remove this statement from the manuscript.

We appreciate your feedback on this matter, as it has helped us to improve the clarity and accuracy of our manuscript.

Reviewer #3:

Remarks to the Author:

The authors addressed our comments fairly convincingly, except for one of the most crucial one:

The code of the analysis absolutely not on par with modern code reproducibility standards (https://github.com/AndersonHu85/ccRCC_multiomics)

All the scripts are dumped in a single folder without any instructions. The scripts even lack to proper file extension (.R). The scripts content themselves rely on source files that are still not provided. None of them are executable.

Reproducibility is at the core of science and this works falls very short on that.

Response: Firstly, we would like to express our gratitude towards the reviewer for bringing our attention to the issue of code reproducibility. We recognized the importance of this issue and have taken actions to rectify it. We have made the following revisions to our code:

File Organization: We have reorganized the various scripts according to their functionality and have added detailed comments at the beginning or the title of each script to explain its purpose

and function. All the scripts have been organized following the figure structure of this paper now.

File Extensions: We have added appropriate file extensions to all of our scripts (for example, all R scripts now have a .R extension).

Source Files: We have uploaded all source files to the repository for easy access and usage by other researchers.

Instructions: We have written a detailed README.md file that includes instructions on how to use these scripts.

We hope that these improvements address the concerns raised by the reviewer and make our research more easily replicable and extendable by other researchers. We greatly appreciate the feedback from the reviewer as it has helped us to improve our work.

Decision Letter, second revision:

26th Sep 2023

Dear Dr Chen,

Thank you for submitting your revised manuscript "Integrated Multi-omics Sequencing Reveals Metabolic Reprograms in the Progression of ccRCC" (NG-A62012R1). It has now been seen by Reviewers #1 and #3, and their comments are below. The reviewers find that the paper has improved in revision, and therefore we'll be happy in principle to publish it in Nature Genetics, pending minor revisions to satisfy the referees' final requests and to comply with our editorial and formatting guidelines.

Sincerely,

Safia Danovi
Editor
Nature Genetics

Reviewer #1 (Remarks to the Author):

All of my comments have been addressed in the resubmission. From my side, there is no need for another review, so that the manuscript can be accepted for publication.

Please double check that all abbreviations were introduced in the main text (e.g., line 342: lipid droplet (LD)). Please check the sentence in line 208-2010. I also found a typo in line 178 'signature'.

Jens Hansen
Icahn School of Medicine at Mount Sinai

Reviewer #3 (Remarks to the Author):

Authors have made significant improvements to the code. The structure is much clearer, and the data seems to be available and the code can be run.

We would suggest to add a readme in each of the figure folders of the GitHub with a summary of what each script is doing, and which files it depended on.

Author Rebuttal, second revision:

Reviewer #1:

Remarks to the Author:

All of my comments have been addressed in the resubmission. From my side, there is no need for another review, so that the manuscript can be accepted for publication.

Please double check that all abbreviations were introduced in the main text (e.g., line 342: lipid droplet (LD)). Please check the sentence in line 208-2010. I also found a typo in line 178 'signature'.

Response: We appreciate your thoughtful review and feedback on our manuscript. We are pleased to know that all your comments have been satisfactorily addressed.

As suggested, we have double-checked the manuscript to ensure all abbreviations have been introduced in the main text. The abbreviation "LD" for "lipid droplet" has been properly defined in line 342. We have also reviewed the sentence in lines 208-210 and made necessary corrections for clarity and coherence. Additionally, the typographical error in line 178 has been corrected from 'signiture' to 'signature'.

Reviewer #3:

Remarks to the Author:

Authors have made significant improvements to the code. The structure is much clearer, and the data seems to be available and the code can be run.

We would suggest to add a readme in each of the figure folders of the GitHub with a summary of what each script is doing, and which files it depended on.

Response: Thank you for your positive comments on the improvements we've made to the code and its structure. We appreciate your time and efforts in reviewing our work.

As suggested by the reviewer, we have added detailed README files that provide a summary of what each script does and its dependencies. These README files will guide users in understanding the functionality of each script and its relationship with other files.

Final Decision Letter:

In reply please quote: NG-A62012R2 Chen

10th Jan 2024

Dear Dr Chen,

I am delighted to say that your manuscript "Multi-omic profiling of clear cell renal cell carcinoma identifies metabolic reprogramming associated with disease progression" has been accepted for publication in an upcoming issue of Nature Genetics.

Over the next few weeks, your paper will be copyedited to ensure that it conforms to Nature Genetics style. Once your paper is typeset, you will receive an email with a link to choose the appropriate

publishing options for your paper and our Author Services team will be in touch regarding any additional information that may be required.

Your paper will be published online after we receive your corrections and will appear in print in the next available issue. You can find out your date of online publication by contacting the Nature Press Office (press@nature.com) after sending your e-proof corrections.

Please note that *Nature Genetics* is a Transformative Journal (TJ). Authors may publish their research with us through the traditional subscription access route or make their paper immediately open access through payment of an article-processing charge (APC). Authors will not be required to make a final decision about access to their article until it has been accepted. [Find out more about Transformative Journals](https://www.springernature.com/gp/open-research/transformative-journals)

Authors may need to take specific actions to achieve [compliance with funder and institutional open access mandates](https://www.springernature.com/gp/open-research/funding/policy-compliance-faqs). If your research is supported by a funder that requires immediate open access (e.g. according to FAIR

[Plan S principles](https://www.springernature.com/gp/open-research/plan-s-compliance)) then you should select the gold OA route, and we will direct you to the compliant route where possible. For authors selecting the subscription publication route, the journal's standard licensing terms will need to be accepted, including <https://www.nature.com/nature-portfolio/editorial-policies/self-archiving-and-license-to-publish>. Those licensing terms will supersede any other terms that the author or any third party may assert apply to any version of the manuscript.

If you have not already done so, we invite you to upload the step-by-step protocols used in this manuscript to the Protocols Exchange, part of our on-line web resource, natureprotocols.com. If you complete the upload by the time you receive your manuscript proofs, we can insert links in your article that lead directly to the protocol details. Your protocol will be made freely available upon publication of your paper. By participating in natureprotocols.com, you are enabling researchers to more readily reproduce or adapt the methodology you use. [Natureprotocols.com](http://natureprotocols.com) is fully searchable, providing your protocols and paper with increased utility and visibility. Please submit your protocol to <https://protocolexchange.researchsquare.com/>. After entering your [nature.com](http://www.nature.com) username and password you will need to enter your manuscript number (NG-A62012R2). Further information can be found at <https://www.nature.com/nature-portfolio/editorial-policies/reporting-standards#protocols>

Sincerely,

Safia Danovi
Editor